# TRANSFORMERS ARE DEEP OPTIMIZERS: PROVABLE IN-CONTEXT LEARNING FOR DEEP MODEL TRAINING

## ABSTRACT

We investigate the transformer's capability for in-context learning (ICL) to simulate the training process of deep models. Our key contribution is providing a positive example of using a transformer to train a deep neural network by gradient descent in an implicit fashion via ICL. Specifically, we provide an explicit construction of a $(2N + 4)L$-layer transformer capable of simulating $L$ gradient descent steps of an $N$-layer ReLU network through ICL. We also give the theoretical guarantees for the approximation within any given error and the convergence of the ICL gradient descent. Additionally, we extend our analysis to the more practical setting using $\mathrm{Softmax}$-based transformers. We validate our findings on synthetic datasets for 3-layer, 4-layer, and 6-layer neural networks. The results show that ICL performance matches that of direct training.

## 1 INTRODUCTION

We study transformers' ability to simulate the training process of deep models. This analysis is not only practical but also timely. On one hand, transformers and deep models (Brown, 2020; Radford et al., 2019) are so powerful, popular and form a new machine learning paradigm — foundation models. These large-scale machine learning models, trained on vast data, provide a general-purpose foundation for various tasks with minimal supervision (Team et al., 2023; Touvron et al., 2023; Zhang et al., 2022). On the other hand, the high cost of pretraining these models often makes them prohibitive outside certain industrial labs (Jiang et al., 2024; Bi et al., 2024; Achiam et al., 2023). In this work, we aim to advance the "one-for-many" modeling philosophy of foundation model paradigm (Bommasani et al., 2021) by considering the following research problem:

**Question 1.** Is it possible to train one deep model with the ICL of another foundation model?

The implication of Question 1 is profound: if true, one foundation model could lead to many others without pertaining. In this work, we provide an affirmative example for Question 1. Specifically, we show that transformer models are capable of simulating the training of a deep ReLU-based feed-forward neural network with provable guarantees through **I**n-**C**ontext **L**earning (**ICL**). Our analysis assumes that we have well-pretrained the transformer using the data generated by the deep network. We require the deep network to maintain consistent hyperparameters (e.g., model width and depth) during the pretraining and testing. However, during the testing, we vary the parameter distribution and input data distribution of the deep network to generate data for the transformer.

In ICL, the models learn to solve new tasks during inference by using task-specific examples provided as part of the input prompt, rather than through parameter updates (Wei et al., 2023; Bubeck et al., 2023; Achiam et al., 2023; Bai et al., 2023; Min et al., 2022; Garg et al., 2022; Brown, 2020). Unlike standard supervised learning, ICL enables models to adapt to new tasks during inference using only the provided examples. In this work, the new task of our interest is algorithmic approximation via ICL (Bai et al., 2023; Zhang et al., 2023; Wang et al.). Specifically, we aim to use transformer's ICL capability to replace/simulate the standard supervised training algorithms for $N$-layer networks. To be concrete, we formalize the learning problem of how transformers learn (i) a given function and (ii) a machine learning algorithm (e.g., gradient descent) via ICL, following (Bai et al., 2023).

**(i) ICL for Function $f$.** Let $f : \mathbb{R}^d \to \mathbb{R}$ be the function of our interest. Suppose we have a dataset $\mathcal{D}_n := \{(x_i, y_i)\}_{i \in [n]}$, where $\{x_i\}_{i \in [n]} \subseteq \mathbb{R}^d$ and $\{y_i\}_{i \in [n]} \subseteq \mathbb{R}$ are the input and output of $f$, respectively. Let $x_{n+1}$ be the test input. The goal of ICL is to use a transformer, denoted by $\mathcal{T}$, to predict $y_{n+1}$ based on the test input and the in-context dataset autoregresively: $\widehat{y}_{n+1} \sim \mathcal{T}(\mathcal{D}_n, x_{n+1})$. The goal is for the prediction $\widehat{y}_{n+1}$ to be close to $y_{n+1} = f(x)$.

**(ii) ICL for Gradient Descent of a Parametrized Model** $f(w, \cdot)$**.** Bai et al. (2023) generalize **(i)** to include algorithmic approximations of Gradient Descent (GD) training algorithms and explore how transformers simulate gradient descent during inference without parameter updates. They term the simulated GD algorithm "In-Context Gradient Descent (ICGD)." In essence, ICGD enables transformers to approximate gradient descent on a loss function $L_n(w)$ for a parameterized model $f(w, \cdot)$ based on a dataset $\mathcal{D}_n$. Traditional gradient descent updates $w$ iteratively as $w_{t+1} = w_t - \eta \nabla L_n(w_t)$. In contrast, ICGD uses a transformer $\mathcal{T}$ to simulate these updates within a forward pass. Given example data $\mathcal{D}_n$ and test input $x_{n+1}$, the transformer performs gradient steps in an implicit fashion by inferring parameter updates through its internal representations, using input context without explicit weight changes. Please see Section 2 for explicit formulation.

In this work, we investigate the case where $f(w, \cdot)$ is a deep feed-forward neural network. We defer the detailed problem setting to Section 2. In comparison to standard ICGD (Bai et al., 2023), ICGD for deep feed-forward networks is not trivial. This is due to two technical challenges:

(C1) Analytical feasibility of gradient computation for these thick networks.

(C2) Explicit construction capable of approximating ICGD for such layers and their gradients.

A work similar to ours is (Wang et al.). It demonstrates that the transformer implements multiple steps of ICGD on deep neural networks. However, it requires more layers in the transformer model and fails to consider the Softmax-transformer. We provide a more detailed comparison in Appendix B.1. To this end, we present the first explicit expression for gradient computation of $N$-layer feed-forward network (Lemma 1). Importantly, its term-by-term tractability provides key insights for the detailed construction of a specific transformer to train this network via ICGD (Theorem 1).

**Contributions.** We offer a positive early investigation of Question 1. Our contributions are threefold:

- **Approximation by ReLU-Transformer.** For simplicity, we begin with the ReLU-based transformer. For a broad class of smooth empirical risks, we construct a $(2N + 4)L$-layer transformer to approximate $L$ steps of in-context gradient descent on the $N$-layer feed-forward networks with the same input and output dimensions (Theorem 1). We then extend this to accommodate varying dimensions (Theorem 5). We also provide the theoretical guarantees for the approximation within any given error (Corollary 1.1) and the convergence of the ICL gradient descent (Lemma 14).

- **Approximation by Softmax-Transformer.** We extend our analysis to the Softmax-transformer to better reflect realistic applications. The key technique is to ensure a qualified approximation error at each point to achieve universal approximation capabilities of the Softmax-based Transformer (Lemma 16). We give a construction of a $4L$-layer Softmax transformer to approximate $L$ steps of gradient descent, and guarantee the approximation and the convergence (Theorem 6).

- **Experimental Validation.** We validate our theory with ReLU- and Softmax-transformers, specifically, ICGD for the $N$-layer networks (Theorem 1, Theorem 5, and Theorem 6). We assess the ICL capabilities of transformers by training 3-, 4-, and 6-layer networks in Appendix G. The numerical results show that the performance of ICL matches that of training $N$-layer networks.

**Organization.** We show our main results in Theorem 1 and the informal version in Appendix A. Section 2 presents preliminaries. Section 3 presents the problem setup and ICL approximation to GD steps of $N$-layer feed-forward network based on ReLU-transformer. The appendix includes the related works (Appendix B.1), the detailed proofs of the main text (Appendix D), ICL approximation to GD steps of $N$-layer network based on Softmax-transformer (Appendix F), the experimental results (Appendix G), and the application to diffusion models (Appendix H).

**Notations.** We use lower case letters to denote vectors and upper case letters to denote matrices. The index set $\{1, ..., I\}$ is denoted by $[I]$, where $I \in \mathbb{N}^+$. For any matrices $A \in \mathbb{R}^{n \times n}$, let $\ell_p$ norm of $A$ be induced by vector $\ell_p$-norm, defined as $\|A\|_p := \sup\{\|Ax\|_p : x \in \mathbb{R}^n \text{ with } \|x\|_p = 1\}$. For any function $f$ and distribution $P$, we denote $L^2(P)$ norm of $f$ as $\|f\|_{L^2(P)} = \mathbb{E}_P^{1/2}[\||f\||_2^2]$. We use $A[i, j]$ to denote the element in $i$-th row and $j$-th column of matrix $A$. For any matrices $A \in \mathbb{R}^{m \times n}$ and $B \in \mathbb{R}^{m \times n}$, let $\odot$ denotes the Hadamard product: $(A \odot B)[i, j] := A[i, j] \cdot B[i, j]$. For any matrices $A \in \mathbb{R}^{m \times n}$ and $B \in \mathbb{R}^{p \times q}$, let $\otimes$ denote the Kronecker product:

$$A \otimes B := \begin{bmatrix} A[1, 1]B & \cdots & A[1, n]B \\ \vdots & \ddots & \vdots \\ A[m, 1]B & \cdots & A[m, n]B \end{bmatrix}.$$

## 2 PRELIMINARIES: ICL AND ICGD

We present the ideas we built upon: In-Context Gradient Descent (ICGD).

**(i) ICL for Function $f$.** Let $f : \mathbb{R}^d \to \mathbb{R}$ be the function of our interest. Suppose we have a dataset $\mathcal{D}_n := \{(x_i, y_i)\}_{i \in [n]}$, where $\{x_i\}_{i \in [n]} \subseteq \mathbb{R}^d$ and $\{y_i\}_{i \in [n]} \subseteq \mathbb{R}$ are the input and output of $f$, respectively. Let $x_{n+1}$ be the test input. The goal of ICL is to use a transformer, denoted by $\mathcal{T}$, to predict $y_{n+1}$ based on the test input and the in-context dataset autoregresively: $\widehat{y}_{n+1} \sim \mathcal{T}(\mathcal{D}_n, x_{n+1})$. For convenience in our analysis, we adopt the ICL notation from (Bai et al., 2023). Specifically, we shorthand $(\mathcal{D}_n, x_{n+1})$ into an input sequence (i.e., prompt) of length $n + 1$ and represent it as a compact matrix $H \in \mathbb{R}^{D \times (n+1)} := [h_1, \ldots, h_{n+1}]$ in the form:

$$H := \begin{bmatrix} x_1 & x_2 & \cdots & x_n & x_{n+1} \\ y_1 & y_2 & \cdots & y_n & 0 \\ q_1 & q_2 & \cdots & q_n & q_{n+1} \end{bmatrix} \in \mathbb{R}^{D \times (n+1)}, \quad q_i := \begin{bmatrix} 0_{D-(d+3)} \\ 1 \\ t_i \end{bmatrix} \in \mathbb{R}^{D-(d+1)}. \quad (2.1)$$

Here, we choose $D := \dim x_i + \dim y_i + \dim q_i = \Theta(d)$. We use $q_i$ to fill in the remain $D - (d+1)$ entries in addition to $x_i \in \mathbb{R}^d$ and $y_i \in \mathbb{R}$. The last entry $t_i := \mathbb{1}(i < n+1)$ of $q_i$ is the position indicator to distinguish the $n$ in-context examples and the test data. The **problem of "ICL for $f$"** is to show the existence of a transformer $\mathcal{T}$ that, when given $H$, outputs $\mathcal{T}(H) \in \mathbb{R}^{D \times (n+1)}$ of the same shape, and the "$(d+1, n+1)$ entry of $\mathcal{T}(H)$" provides the prediction $\widehat{y}_{n+1}$. The goal is for the prediction $\widehat{y}_{n+1}$ to be close to $y_{n+1} = f(x)$ measured by some proper loss.

**(ii) ICL for Gradient Descent of a Parametrized Model $f(w, \cdot)$.** In this work, we aim to use ICL to replace/simulate the standard supervised training procedure for $N$-layer neural networks. To achieve this, we introduce the concept of In-Context Gradient Descent (ICGD) for a parameterized model.

Consider a machine learning model $f(w, \cdot) : \mathbb{R}^{D_w} \times \mathbb{R}^d \to \mathbb{R}^d$, parametrized by $w \in \mathbb{R}^{D_w}$. Given a dataset $\mathcal{D}_n := \{(x_i, y_i)\}_{i \in [n]} \overset{\text{iid}}{\sim} \mathbb{P}$, a typical learning task is to find parameters $w^\star$ such that $f(w^\star, \cdot)$ becomes closest to the true data distribution $\mathbb{P}$. Then, for any test input $x_{n+1}$, we predict: $\widehat{y}_{n+1} = f(w^\star, x_{n+1})$. To find $w^\star$, Bai et al. (2023) configure a transformer to implement gradient descent on $f(w, \cdot)$ through ICL, simulating optimization algorithms during inference without explicit parameter updates. We formalize this **In-Context Gradient Descent (ICGD)** problem: using a pretrained model to simulate gradient descent on $f(w, \cdot)$ w.r.t. the provided context $(\mathcal{D}_n, x_{n+1})$.

**Problem 1** (In-Context Gradient Descent (ICGD) on Model $f(w, \cdot)$ (Bai et al., 2023))**.** Let $\epsilon > 0$ and $L \geq 1$. Consider a machine learning model $f(w, x) : \mathbb{R}^{D_w} \times \mathbb{R}^d \to \mathbb{R}^d$ parameterized by $w \in \mathbb{R}^{D_w}$. Given a dataset $\mathcal{D}_n := \{(x_i, y_i)\}_{i \in [n]} \overset{\text{iid}}{\sim} \mathbb{P}$ with $(x_i, y_i) \in \mathbb{R}^d \times \mathbb{R}^d$, define the empirical risk function:

$$\mathcal{L}_n(w) := \frac{1}{2n} \sum_{i=1}^n \ell(f(w, x_i), y_i), \quad \text{where } \ell : \mathbb{R}^d \times \mathbb{R}^d \to \mathbb{R} \text{ is a loss function.} \quad (2.2)$$

Let $\mathcal{W} \subseteq \mathbb{R}^{D_w}$ be a closed domain, and $\text{Proj}_{\mathcal{W}}$ denote the projection onto $\mathcal{W}$. The problem of "ICGD on model $f(w, \cdot)$" is to find a transformer $\mathcal{T}$ with $L$ blocks, each approximating one step of gradient descent using $T$ layers. For any input $H^{(0)} \in \mathbb{R}^{D \times (n+1)}$ in the form of (2.1), the transformer $\mathcal{T}(H^{(0)})$ approximates $L$ steps of gradient descent. Specifically, for $l \in [L]$ and $i \in [n+1]$, the output at layer $Tl$ is: $h_i^{(Tl)} = [x_i; y_i; \overline{w}^{(l)}; \mathbf{0}; 1; t_i]$, where, with $\overline{w}^{(0)} = \mathbf{0}$,

$$\overline{w}^{(l)} = \text{Proj}_{\mathcal{W}} \left( \overline{w}^{(l-1)} - \eta \left( \nabla \mathcal{L}_n(\overline{w}^{(l-1)}) + \epsilon^{(l-1)} \right) \right) \text{ is updated recursively,} \quad (2.3)$$

and $\|\epsilon^{(l-1)}\|_2 \leq \epsilon$ represents the approximation error at step $l - 1$.

Problem 1 aims to find a transformers $\mathcal{T}$ to perform $L$ steps gradient descent on loss $\mathcal{L}_n(w)$ in an implicit fashion (i.e., no explicit parameter update). More precisely, Bai et al. (2023) configure $\mathcal{T}$ with $L$ identical blocks, each approximating one gradient descent step using $T$ layers. In this work, we investigate the case where $f(w, \cdot)$ is an "$N$-layer neural network."

**Transformer.** We defer standard definition of transformer to Appendix C.1 due to the page limit.

## 3 IN-CONTEXT GRADIENT DESCENT ON $N$-LAYER NEURAL NETWORKS

We now show that transformers is capable of implementing gradient descent on $N$-layer neural networks through ICL. In Section 3.1, we define the $N$-layer ReLU neural network and state its ICGD problem. In Section 3.2, we derive explicit gradient descent expression for $N$-layer NN. In Section 3.3, we show how transformers execute gradient descent on $N$-layer NN via ICL.

### 3.1 PROBLEM SETUP: ICGD FOR $N$-LAYER NEURAL NETWORK

To begin, we introduce the construction of our $N$-Layer Neural Network which we aims to implement gradient descent on its empirical loss function.

**Definition 1** ($N$-Layer Neural Network). An $N$-Layer Neural Network comprises $N - 1$ hidden layers and 1 output layer, all constructed similarly. Let $r : \mathbb{R} \to \mathbb{R}$ be the activation function. For the hidden layers: for any $i \in [n + 1], j \in [N - 1]$, and $k \in [K]$, the output for the first $j$ layers w.r.t. input $x_i \in \mathbb{R}^d$, denoted by $\mathrm{pred}_h(x_i; j) \in \mathbb{R}^K$, is defined as recursive form:

$$\mathrm{pred}_h(x_i; 1)[k] := r(v_{1_k}^\top x_i), \quad \text{and} \quad \mathrm{pred}_h(x_i; j)[k] := r(v_{j_k}^\top \mathrm{pred}_h(x_i; j - 1)),$$

where $v_{1_k} \in \mathbb{R}^d$ and $v_{j_k} \in \mathbb{R}^K$ for $j \in \{2, \ldots, N - 1\}$ are the $k$-th parameter vectors in the first layer and the $j$-th layer, respectively. For the output layer ($N$-th layer), the output for the first $N$ layers (i.e the entire neural network) w.r.t. input $x_i \in \mathbb{R}^d$, denoted by $\mathrm{pred}_o(x_i; w, N) \in \mathbb{R}^d$, is defined for any $k \in [d]$ as follows:

$$\mathrm{pred}_o(x_i; w, N)[k] := r(v_{N_k}^\top \mathrm{pred}_h(x_i; N - 1)), \tag{3.1}$$

where $v_{N_k} \in \mathbb{R}^K$ are the $k$-th parameter vectors in the $N$-th layer and $w \in \mathbb{R}^{2dK+(N-2)K^2}$ denotes the vector containing all parameters in the neural network,

$$w := \left[v_{1_1}^\top, \ldots, v_{1_K}^\top, \ldots, v_{j_k}^\top, \ldots v_{N-1_1}^\top, \ldots, v_{N-1_K}^\top, v_{N_1}^\top, \ldots, v_{N_d}^\top\right]^\top. \tag{3.2}$$

**Remark 1** (Prediction Function for $j$-th layer on $i$-th Data: $p_i(j)$). For simplicity, we abbreviate the output from the first $j$-th layer of the $N$-layer neural networks NN with input $x_i$ as $p_i(j)$,

$$p_i(j) := \begin{cases} x_i \in \mathbb{R}^d, & \text{for } j = 0, \\ \mathrm{pred}_h(x_i; j) \in \mathbb{R}^K, & \text{for } j \in [N - 1], \\ \mathrm{pred}_o(x_i; w, N) \in \mathbb{R}^d, & \text{for } j = N. \end{cases} \tag{3.3}$$

Additionally, we define $p_i := [p_i(1); \ldots; p_i(N)] \in \mathbb{R}^{(N-1)K+d}$.

We formalize the problem of using a transformer to simulate gradient descent algorithms for training the $N$-layer NN defined in Definition 1, by optimizing loss (2.2). Specifically, we consider the ICGD (Problem 1) with the parameterized model $f(w, \cdot) := \mathrm{pred}_o(\cdot; w, N)$.

**Problem 2** (ICGD on $N$-Layer Neural Networks). Let the $N$-layer neural networks, activation function $r$, and prediction function $p_i(j)$ for all layers follow Definition 1 and Remark 1. Assume we under the identical setting as Problem 1, considering model $f(w, \cdot) := \mathrm{pred}_o(\cdot; w, N)$ and specifying $\mathcal{W}$ is a closed domain such that for any $j \in [N - 1]$ and $k \in [K]$,

$$\mathcal{W} \subset \left\{w = [v_{j_k}] \in \mathbb{R}^{D_N} : \|v_{j_k}\|_2 \le B_v\right\}. \tag{3.4}$$

The problem of "ICGD on $N$-layer neural networks" is to find a $TL$ layers transformer $\mathcal{T}$, capable of implementing $L$ steps gradient descent as in Problem 1. Specifically, for any $L \in [L], i \in [n + 1]$, the $Tl$-th layer outputs $h_i^{(Tl)} = [x_i; y_i; \overline{w}^{(l)}; \mathbf{0}; 1; t_i]$, where:

$$\overline{w}^{(l)} = \mathrm{Proj}_{\mathcal{W}} \left(\overline{w}^{(l-1)} - \eta \left(\nabla \mathcal{L}_n(\overline{w}^{(l-1)}) + \epsilon^{(l-1)}\right)\right), \quad \overline{w}^{(0)} = \mathbf{0}, \tag{3.5}$$

and $\|\epsilon^{(l-1)}\|_2 \le \epsilon$ is the error term generated from the approximation in the $l$-th step.

**Remark 2** (Necessary for bounded domain $\mathcal{W}$). For using a sum of ReLU to approximate functions like $r$, which illustrated in the consequent section, we need to avoid gradient exploding. Therefore, we require $\mathcal{W}$ to be a bounded domain, and utilize $\mathrm{Proj}_{\mathcal{W}}$ to project $w$ into bounded domain $\mathcal{W}$.

### 3.2 EXPLICIT GRADIENT DESCENT OF $N$-LAYER NEURAL NETWORK

Intuitively, Problem 2 asks whether there exists a transformer capable of simulating the gradient descent algorithm on the loss function of an $N$-layer neural network. We answer Problem 2 by providing an explicit construction for such a transformer $\mathcal{T}$ in Theorem 1. To facilitate our proof, we first introduce the necessary notations for explicit expression of the gradient $\nabla_w \mathcal{L}_n(w)$.

**Definition 2** (Abbreviations). Fix $i \in [n + 1]$, and consider an $N$-layer neural network with activation function $r$ and prediction function $p_i(j)$ as defined in Definition 1.

- Let $D_j \in \mathbb{R}$ denote the total number of parameters in the first $j$ layers. By (3.2), we have:

$$D_j = \begin{cases} 0, & j = 0 \\ dK, & j = 1 \\ (j-1)K^2 + dK, & 2 \leq j \leq N-1 \\ (N-2)K^2 + 2dK, & j = N. \end{cases}$$

- The parameter vector $w := \left[ v_{1_1}^\top, \ldots, v_{1_K}^\top, \ldots, v_{N-1_1}^\top, \ldots, v_{N-1_K}^\top, v_{N_1}^\top, \ldots, v_{N_d}^\top \right]^\top$ follows (3.2). Define $\phi_i := \left( \frac{\partial \ell(p_i(N), y_i)}{\partial p_i(N)} \cdot \frac{\partial p_i(N)}{\partial w} \right)^\top \in \mathbb{R}^{D_N}$. For any $j \in [N]$, let $A_i(j)$ denote the derivative of $\ell(p_i(N), y_i)$ with respect to the parameters in the $j$-th layer:$A_i(j) = \phi_i[D_{j-1} : D_j]$, where $\phi_i[a : b]$ selects elements from the $a$-th to $b$-th position in $\phi_i$.

- For activation function $r(t)$, let $r'(t)$ be its derivative. Define $r_i'(j) \in \mathbb{R}^K$ as:

$$r_i'(j)[k] := r'(v_{j+1_k}^\top p_i(j)).$$

- Define $r_i' := [r_i'(0); \ldots; r_i'(N-1)]$ and $R_i(j)$ as:

$$R_i(j) := \begin{cases} \mathrm{diag}\{r'(v_{j+1_1}^\top p_i(j)), \ldots, r'(v_{j+1_K}^\top p_i(j))\} \in \mathbb{R}^{K \times K}, & j \in \{0, 1, \ldots N-2\} \\ \mathrm{diag}\{r'(v_{j+1_1}^\top p_i(j)), \ldots, r'(v_{j+1_d}^\top p_i(j))\} \in \mathbb{R}^{d \times d}, & j = N-1. \end{cases}$$

- For any $j \in [N]$, let $V_j$ denote the parameters in the $j$-th layer as:

$$V_j := \begin{cases} \left[ v_{1_1}, \ldots, v_{1_K} \right]^\top \in \mathbb{R}^{K \times d}, & j = 1 \\ \left[ v_{j_1}, \ldots, v_{j_K} \right]^\top \in \mathbb{R}^{K \times K}, & j \in 2, \ldots, N-1 \\ \left[ v_{N_1}, \ldots, v_{N_d} \right]^\top \in \mathbb{R}^{d \times K}, & j = N. \end{cases}$$

Definition 2 splits the gradient of $\mathcal{L}_n(w)$ into $N$ parts. This makes $\nabla_w \mathcal{L}_n(w)$ more interpretable and tractable, since all parts follows a recursion formula according to chain rule. With above notations, we calculate the gradient descent step (3.5) of $N$-layer neural network as follows:

**Lemma 1** (Decomposition of One Gradient Descent Step). Fix any $B_v, \eta > 0$. Suppose loss function $\mathcal{L}_n(w)$ on $n$ data points $\{(x_i, y_i)\}_{i \in [n]}$ follows (2.2). Suppose closed domain $\mathcal{W}$ and projection function $\mathrm{Proj}_{\mathcal{W}}(w)$ follows (3.4). Let $A_i(j), r_i'(j), R_i(j), V_j$ be as defined in Definition 2. Then the explicit form of gradient $\nabla \mathcal{L}_n(w)$ becomes

$$\nabla \mathcal{L}_n(w) = \frac{1}{2n} \sum_{i=1}^n \begin{bmatrix} A_i(1) \\ \vdots \\ A_i(N) \end{bmatrix}, \tag{3.6}$$

where $A_i(j)$ denote the derivative of $\ell(p_i(N), y_i)$ with respect to the parameters in the $j$-th layer,

$$A_i(j) = \begin{cases} (R_i(N-1) \cdot V_N \cdot \ldots \cdot R_i(j-1) \cdot \left[ \mathbf{I}_{K \times K} \otimes p_i(j-1)^\top \right])^\top \cdot (\frac{\partial \ell(p_i(N), y_i)}{\partial p_i(N)})^\top, & j \neq N \\ (R_i(N-1) \cdot \left[ \mathbf{I}_{d \times d} \otimes p_i(N-1)^\top \right])^\top \cdot (\frac{\partial \ell(p_i(N), y_i)}{\partial p_i(N)})^\top, & j = N. \end{cases}$$

*Proof Sketch.* Using the chain rule and product rule, we decompose the gradient as follows: $\nabla_w \mathcal{L}_n(w) = \frac{1}{2n} \sum_{i=1}^N [\frac{\partial p_i(N)}{\partial w}]^\top \cdot [\frac{\partial \ell(p_i(N), y_i)}{\partial p_i(N)}]^\top$. Thus, we only need to compute $\frac{\partial p_i(N)}{\partial w}$. By Definition 1 and the chain rule, we prove that $\frac{\partial p_i(N)}{\partial w}$ satisfies the recursive formulation (D.4). Combining these, we derive the explicit form of gradient $\nabla_w \mathcal{L}_n(w)$ , and the gradient step follows directly. Please see Appendix D.1 for a detailed proof. $\square$

Since it is hard to calculate the elements in $A_i(j)$ in a straightforward mannar, we calculate each parts of it successively. Specifically, we define the intermediate terms $s_i(j)$ and $u$ as follows

**Definition 3** (Definition of intermediate terms). Let $A_i(j), r_i'(j), R_i(j), V_j$ be as defined in Definition 2. By Lemma 1, the derivative of $\ell(p_i(N), y_i)$ w.r.t the parameters in the $j$-th layer follows,

$$A_i(j) = \begin{cases} (R_i(N-1) \cdot V_N \cdot \ldots \cdot R_i(j-1) \cdot \left[ \mathbf{I}_{K \times K} \otimes p_i(j-1)^\top \right])^\top \cdot (\frac{\partial \ell(p_i(N), y_i)}{\partial p_i(N)})^\top, & j \neq N \\ (R_i(N-1) \cdot \left[ \mathbf{I}_{d \times d} \otimes p_i(N-1)^\top \right])^\top \cdot (\frac{\partial \ell(p_i(N), y_i)}{\partial p_i(N)})^\top, & j = N. \end{cases}$$

For any $t, y \in \mathbb{R}^d$, we define vector function $u(t, y) := (\frac{\partial \ell(t, y)}{\partial t})^\top : \mathbb{R}^d \times \mathbb{R}^d \to \mathbb{R}^d$. Moreover, for any $j \in [N], i \in [n+1]$, we define $s_i(j)$ as

$$
s_i(j) := \begin{cases} R_i(j-1)V_{j+1}^\top \dots R_i(N-2)V_N^\top \cdot R_i(N-1) \cdot u(p_i(N), y_i), & \in \mathbb{R}^K, j \neq N, \\ R_i(N-1) \cdot u(p_i(N), y_i), & \in \mathbb{R}^d, j = N. \end{cases}
$$

Let $\odot$ denotes hadamard product. For any $j \in [N-1], i \in [N+1]$, Definition 3 leads to

$$
s_i(j) = r_i'(j-1) \odot (V_{j+1}^\top \cdot s_i(j+1)), \tag{3.7}
$$

Moreover, by Definition 3, it holds

$$
A_i(j) = \begin{cases} \left[\mathbf{I}_{K \times K} \otimes p_i(j-1)\right] \cdot s_i(j), & j \neq N, \\ \left[\mathbf{I}_{d \times d} \otimes p_i(N-1)\right] \cdot s_i(N), & j = N. \end{cases} \tag{3.8}
$$

## 3.3 TRANSFORMERS APPROXIMATE GRADIENT DESCENT OF $N$-LAYER NEURAL NETWORKS WITH ICL

For using neural networks to approximate (2.2), which contains smooth functions changeable, we need to approximate these smooth functions by simple combination of activation functions. Our key approximation theory is using a sum of ReLUs to approximate any smooth function (Bai et al., 2023).

**Definition 4** (Approximability by Sum of ReLUs, Definition 12 of (Bai et al., 2023)). Let $z \in \mathbb{R}^k$. We say that a function $g : \mathbb{R}^k \to \mathbb{R}$ is $(\epsilon_{\text{approx}}, R, H, C)$-approximable by sum of ReLUs if there exist a "$(H, C)$-sum of ReLUs" function $f_{H,C}(z)$ defined as

$$
f_{H,C}(z) = \sum_{h=1}^H c_h \sigma(a_h^\top [z; 1]) \quad \text{with} \quad \sum_{h=1}^H |c_h| \leq C, \quad \max_{h \in [H]} \|a_h\|_1 \leq 1, \quad a_h \in \mathbb{R}^{k+1}, \quad c_h \in \mathbb{R},
$$

such that $\sup_{z \in [-R,R]^k} |g(z) - f_{H,C}(z)| \leq \epsilon_{\text{approx}}$.

**Overview of Our Proof Strategy.** Lemma 1 and Definition 4 motivate the following strategy: term-by-term approximation for our gradient descent step (3.6).

**Step 1.** Given $(x_i, w)$, we use $N$ attention layers to approximate the output of the first $j$ layers with input $x_i$, $p_i(j) := \text{pred}_h(x_i; j) \in \mathbb{R}^k$ (Definition 1) for any $j \in [N]$. Then we use 1 attention layer to approximate chain-rule intermediate terms $r_i'(j-1)[k] := r'(v_{j_k}^\top p_i(j-1))$ (Definition 2) for any $i \in [n], j \in [N]$ and $k \in [K]$: Lemma 2 and Lemma 3.

**Step 2.** Given $(r_i', p_i, w)$, we use an MLP layer to approximate $u(p_i(N), y_i)$ (Definition 3), for $i \in [n]$, and use $N$ element-wise multiplication layers to approximate $s_i(j)$ (Definition 3), for any $j \in [N]$: Lemma 4 and Lemma 5. Moreover, Lemma 6 shows the closeness result for approximating $s_i(j)$, which leads to the final error accumulation in Theorem 1.

**Step 3.** Given $(p_i, r_i', g_i s_i(j), w)$, we use an attention layer to approximate $w - \eta \nabla \mathcal{L}_n(w)$. Then we use an MLP layer to approximate $\text{Proj}_{\mathcal{W}}(w)$. And implementing $L$ steps gradient descent by a $(2N+4)L$-layer neural network $\text{NN}_\theta$ constructed based on **Step 1 and 2**. Finally, we arrive our main result: Theorem 1. Furthermore, Lemma 14 shows closeness results to the true gradient descent path.

**Step 1.** We start with approximation for $p_i(j)$.

**Lemma 2** (Approximate $p_i(j)$). Let upper bounds $B_v, B_x > 0$ such that for any $k \in [K], j \in [N]$ and $i \in [n]$, $\|v_{j_k}\|_2 \leq B_v$, and $\|x_i\|_2 \leq B_x$. For any $j \in [N], i \in [n]$, define

$$
B_r^j := \max_{|t| \leq B_v B_r^{j-1}} |r(t)|, \quad B_r^0 := B_x, \quad \text{and} \quad B_r := \max_j B_r^j.
$$

Let function $r(t)$ be $(\epsilon_r, R_1, M_1, C_1)$-approximable for $R_1 = \max\{B_v B_r, 1\}$, $M_1 \leq \widetilde{\mathcal{O}}(C_1^2 \epsilon_r^{-2})$, where $C_1$ depends only on $R_1$ and the $C^2$-smoothness of $r$. Then, for any $\epsilon_r > 0$, there exist $N$ attention layers $\text{Attn}_{\theta_1}, \dots, \text{Attn}_{\theta_N}$ such that for any input $h_i \in \mathbb{R}^D$ takes from (2.1), they map

$$
h_i = [x_i; y_i; w; \bar{p}_i(1); \dots; \bar{p}_i(j-1); \mathbf{0}; 1; t_i] \xrightarrow{\text{Attn}_{\theta_j}} \widetilde{h}_i = [x_i; y_i; w; \bar{p}_i(1); \dots; \bar{p}_i(j); \mathbf{0}; 1; t_i],
$$

where $\bar{p}_i(j)$ is approximation for $p_i(j)$ (Definition 1). In the expressions of $h_i$ and $\widetilde{h}_i$, the dimension of $\mathbf{0}$ differs. Specifically, the $\mathbf{0}$ in $h_i$ is larger than in $\widetilde{h}_i$. The dimensional difference between these $\mathbf{0}$

vectors equals the dimension of $\bar{p}_i(j)$. Suppose function $r$ is $L_r$-smooth in bounded domain $\mathcal{W}$, then for any $i \in [n+1], j \in [N], \bar{p}_i(j)$ such that

$$\bar{p}_i(j) = p_i(j) + \epsilon(i,j), \quad \|\epsilon(i,j)\|_2 \leq \begin{cases} (\sum_{l=0}^{j-1} K^{l/2} L_r^l B_v^l)\sqrt{K}\epsilon_r, & 1 \leq j \leq N-1 \\ (\sum_{l=0}^{N-1} K^{l/2} L_r^l B_v^l)\sqrt{d}\epsilon_r, & j = N \end{cases}. \quad (3.9)$$

Additionally, for any $j \in [N]$, the norm of parameters $B_{\theta_j}$ defined as (C.1) such that $B_{\theta_j} \leq 1 + KC_1$.

*Proof Sketch.* By Definition 1, we provide term-by-term approximations for $p_i(j)$ as forward propagation. Specifically, we construct Attention layers to implement forward propagation algorithm. Then we establish upper bounds for the errors $\|\bar{p}_i(j) - p_i(j)\|_2$ inductively. Finally, we present the norms (C.1) of the Transformers constructed. Please see Appendix D.2 for a detailed proof. □

Notice that the form of error accumulation in Lemma 2 is complicated. For the ease of later presentations, we define the upper bound of coefficient in (3.9) as

$$E_r := \max_{j \in [N]} \frac{\|\epsilon(i,j)\|_2}{\epsilon_r} = \max_{j \in [N]} \{(\sum_{l=0}^{j-1} K^{l/2} L_r^l B_v^l)\sqrt{K}, (\sum_{l=0}^{N-1} K^{l/2} L_r^l B_v^l)\sqrt{d}\}, \quad (3.10)$$

such that (3.9) becomes

$$\bar{p}_i(j) = p_i(j) + \epsilon(i,j), \quad \|\epsilon(i,j)\|_2 \leq E_r \epsilon_r. \quad (3.11)$$

Moreover, we abbreviate $\bar{p}_i := [\bar{p}_i(1); \ldots; \bar{p}_i(N)] \in \mathbb{R}^{(N-1)K+d}$, such that the output of $\text{Attn}_{\theta_1} \circ \cdots \circ \text{Attn}_{\theta_N}$ is

$$h_i = [x_i; y_i; w; \bar{p}_i; \mathbf{0}; 1; t_i]. \quad (3.12)$$

Then, the next lemma approximates $r_i'(j)$ base on $\bar{p}_i(j)$ obtained in Lemma 2.

**Lemma 3** (Approximate $r_i'(j)$). Let upper bounds $B_v, B_x > 0$ such that for any $k \in [K], j \in [N]$ and $i \in [n]$, $\|v_{j_k}\|_2 \leq B_v$, and $\|x_i\|_2 \leq B_x$. For any $j \in [N], i \in [n]$, define

$$B_r'^j := \max_{|t| \leq B_v B_{r'}^{j-1}} |r'(t)|, \quad B_{r'}^0 := B_x, \quad \text{and} \quad B_{r'} := \max_j B_{r'}^j.$$

Suppose function $r'(t)$ is $(\epsilon_{r'}, R_2, M_2, C_2)$-approximable for $R_2 = \max\{B_v B_{r'}, 1\}$, $M_2 \leq \widetilde{\mathcal{O}}(C_2^2 \epsilon_r'^{-2})$, where $C_2$ depends only on $R_2$ and the $C^2$-smoothness of $r'$. Then, for any $\epsilon_r > 0$, there exist an attention layer $\text{Attn}_{\theta_{N+1}}$ such that for any input $h_i \in \mathbb{R}^D$ takes from (3.12), it maps

$$h_i = [x_i; y_i; w; \bar{p}_i; \mathbf{0}; 1; t_i] \xrightarrow{\text{Attn}_{\theta_{N+1}}} \widetilde{h}_i = [x_i; y_i; w; \bar{p}_i; \bar{r}_i'; \mathbf{0}; 1; t_i],$$

where $\bar{r}_i'(j)$ is approximation for $r_i'(j)$ (Definition 2) and $\bar{r}_i' := [\bar{r}_i'(0); \ldots; \bar{r}_i'(N-1)] \in \mathbb{R}^{(N-2)K+d}$. Similar to Lemma 2, in the expressions of $h_i$ and $\widetilde{h}_i$, the dimension of $\mathbf{0}$ differs. In addition, let $E_r$ be defined in (3.11), for any $i \in [n+1], j \in [N], k \in [K], \bar{r}_i'(j)$ such that

$$\bar{r}_i'(j-1)[k] = r_i'(j-1)[k] + \epsilon(i,j,k), \quad |\epsilon(i,j,k)| \leq \epsilon_{r'} + L_{r'} B_v E_r \epsilon_r, \quad (3.13)$$

where $\epsilon_r$ denotes the error generated in approximating $r$ by sum of ReLUs $\bar{r}$ follows (D.5). Additionally, the norm of parameters $B_{\theta_{N+1}}$ defined as (C.1) such that $B_{\theta_{N+1}} \leq 1 + K(N-1)C_2$.

*Proof Sketch.* By Lemma 2, we obtain $\bar{p}_i(j)$, the approximation for $p_i(j)$ (3.3). Using $\bar{p}_i(j)$, we construct an Attention layer to approximate $r_i'(j)$. We then establish upper bounds for the errors $|\bar{r}_i'(j)[k] - r_i'(j)[k]|$ by applying Cauchy-Schwarz inequality and Lemma 2. Finally we present the norms (C.1) of the Transformers constructed. Please see Appendix D.3 for a detailed proof. □

Let $\text{Attn}_{\theta_j}(j \in [N])$ be as defined in Lemma 2, then Lemma 3 implies that for the input takes from Problem 2, the output of $\text{Attn}_{\theta_1} \circ \cdots \circ \text{Attn}_{\theta_{N+1}}$ is

$$h_i = [x_i; y_i; w; \bar{p}_i; \bar{r}_i'; \mathbf{0}; 1; t_i]. \quad (3.14)$$

**Step 2.** Now, we construct an approximation for $u(p_i(N), y_i) = (\frac{\partial \ell(p_i(N), y_i)}{\partial p_i(N)})^\top$.

**Lemma 4** (Approximate $u(p_i(N), y_i)$). Let upper bounds $B_v, B_x, > 0$ such that for any $k \in [K], j \in [N]$ and $i \in [n], \|v_{j_k}\|_2 \leq B_v$, and $\|x_i\|_2 \leq B_x$. For any $k \in [d]$, suppose function $u(t, y)[k]$ be $(\epsilon_l, R_3, M_3^k, C_3^k)$-approximable for $R_3 = \max\{B_v B_r, B_y, 1\}$, $M_3 \leq \widetilde{\mathcal{O}}((C_3^k)^2 \epsilon_l^{-2})$,

where $C_3^k$ depends only on $R_3^k$ and the $C^3$-smoothness of $u(t, y)[k]$. Then, there exists an MLP layer $\text{MLP}_{\theta_{N+2}}$ such that for any input sequences $h_i \in \mathbb{R}^D$ takes from (3.14), it maps

$$h_i = [x_i; y_i; w; \bar{p}_i; \bar{r}_i'; \mathbf{0}; 1; t_i] \xrightarrow{\text{MLP}_{\theta_{N+2}}} \widetilde{h}_i = [x_i; y_i; w; \bar{p}_i; \bar{r}_i'; g_i; \mathbf{0}; 1; t_i],$$

where $g_i \in \mathbb{R}^d$ is an approximation for $u(p_i(N), y_i)$. For any $k \in [d]$, assume $u(p_i(N), y_i)$ is $L_l$-Lipschitz continuous. Then the approximation $g_i$ such that,

$$g_i[k] = u(p_i(N), y_i)[k] + \epsilon(i, k), \quad \text{with} \quad |\epsilon(i, k)| \leq \epsilon_l + L_l E_r \epsilon_r. \tag{3.15}$$

Additionally, the parameters $\theta_{N+2}$ such that $B_{\theta_{N+2}} \leq \max\{R_3 + 1, C_3\}$.

*Proof Sketch.* By Lemma 2, we obtain $\bar{p}_i(N)$, the approximation for $p_i(N)$ (3.3). Using $\bar{p}_i(N)$, we construct an MLP layer to approximate $u$. We then establish upper bounds for the errors $|g_i[k] - u[k]|$ and present the norms (C.1) of Transformers constructed. See Appendix D.4 for a detailed proof. $\square$

Let $\text{Attn}_{\theta_j} (j \in [N + 1])$ be as defined in Lemma 2 and Lemma 3, then for any input sequences $h_i \in \mathbb{R}^D$ takes from (2.1), the output of $\text{Attn}_{\theta_1} \circ \cdots \circ \text{Attn}_{\theta_{N+1}} \circ \text{MLP}_{\theta_{N+2}}$ is

$$h_i = [x_i; y_i; w; \bar{p}_i; \bar{r}_i'; g_i; \mathbf{0}; 1; t_i]. \tag{3.16}$$

Before introducing our next approximation lemma, we define an element-wise multiplication layer, since attention mechanisms and MLPs are unable to compute self-products (e.g., output $xy$ from input $[x; y]$). To enable self-multiplication, we introduce a function $\gamma$. This function, for any square matrix, preserves the diagonal elements and sets all others to zero.

**Definition 5** (Operator Function $\gamma$). For any square matrix $A \in \mathbb{R}^{n \times n}$, define $\gamma(A) := \text{diag}(A[1, 1], \ldots A[n, n]) \in \mathbb{R}^{n \times n}$.

By Definition 5, we introduce the following element-wise multiplication layer, capable of performing self-multiplication operations such as the Hadamard product.

**Definition 6** (Element-wise Multiplication Layer). Let $\gamma$ be defined as Definition 5. An element-wise multiplication layer with $m$ heads is denoted as $\text{Attn}_\theta(\cdot)$ with parameters $\theta = \{Q_m, K_m, V_m\}_{m \in [M]}$. On any input sequence $H \in \mathbb{R}^{D \times n}$,

$$\text{EWML}_\theta(H) = H + \sum_{i=1}^{m} (V_m H) \cdot \gamma((Q_m H)^\top (K_m H)). \tag{3.17}$$

where $Q_m, K_m, V_m \in \mathbb{R}^{D \times D}$ and $\gamma(\cdot)$ is operator function follows Definition 5. In vector form, for for each token $h_i \in \mathbb{R}^D$ in $H$, it outputs $[\text{EWML}_\theta(H)]_i = h_i + \sum_{m=1}^{M} \gamma(\langle Q_m h_i, K_m h_i \rangle) \cdot V_m h_i$. In addition, we define $L$-layer neural networks $\text{EWML}_\theta^L := \text{EWML}_{\theta_1} \circ \cdots \circ \text{EWML}_{\theta_L}$.

**Remark 3** (Necessary for Element-Wise Multiplication Layer). As we shall show in subsequent sections, element-wise multiplication layer is capable of implementing multiplication in $h_i$. Specifically, it allows us to multiply some elements in $h_i$ in Lemma 5. By Definition 7, it is impossible for transformer layers to achieve our goal without any other assumptions.

Similar to (C.1), we define the norm for $L$-layer transformer $\text{EWML}_\theta^L$ as:

$$B_\theta := \max_{m \in [M], l \in [L]} \{\|Q_m^l\|_1, \|K_m^l\|_1, \|V_m^l\|_1\}. \tag{3.18}$$

Then, given the approximations for $p_i(j)$ and $r_i'(j)$, we use $N$ element-wise multiplication layer (Definition 6) to approximate $s_i(j)$, the chain-rule intermediate terms defined as Definition 3.

**Lemma 5** (Approximate $\bar{s}_t(j)$). Recall that $s_i(j) = r_i'(j-1) \odot (V_{j+1}^\top \cdot s_i(j+1))$ follows Definition 3. Let the initial input take from (3.16). Then, there exist $N$ element-wise multiplication layers: $\text{EWML}_{\theta_{N+3}}, \ldots, \text{EWML}_{\theta_{2N+2}}$ such that for input sequences, $j \in [N]$,

$$h_i = [x_i; y_i; w; \bar{p}_i; \bar{r}_i'; g_i; \bar{s}_i(N); \ldots; \bar{s}_i(j+1); \mathbf{0}; 1; t_i],$$

they map $\mathrm{EWML}_{\theta_{2N+3-j}}(h_i) = [x_i; y_i; w; \bar{p}_i; \bar{r}'_i; g_i; \bar{s}_i(N); \ldots; \bar{s}_i(j); \mathbf{0}; 1; t_i]$, where the approximation $\bar{s}_i(j)$ is defined as recursive form: for any $i \in [n+1], j \in [N]$,

$$\bar{s}_i(j) := \begin{cases} \bar{r}'_i(j-1) \odot (V_{j+1}^\top \cdot \bar{s}_i(j+1)), & j \in [N-1] \\ \bar{r}'_i(N-1) \odot g_i, & j = N. \end{cases} \tag{3.19}$$

Additionally, for any $j \in [N]$, $B_{\theta_{N+2+j}}$ defined in (C.1) satisfies $B_{\theta_{N+2+j}} \leq 1$.

*Proof Sketch.* By Lemma 2 and Lemma 3, we obtain $\bar{p}_i(j)$ and $\bar{r}'_i(j)$, the approximation for $p_i(j)$ (3.3) and $r'_i(j)$ respectively. Using $\bar{p}_i(j)$ and $\bar{r}'_i(j)$, we construct $N$ element-wise multiplication layers to approximate $s_i(j)$. We then present the norms (3.18) of the EWMLs constructed. Please see Appendix D.5 for a detailed proof. □

Let $\mathrm{Attn}_{\theta_j}(j \in [N+1]), \mathrm{MLP}_{\theta_{N+2}}$ be as defined in Lemma 2, Lemma 3 and Lemma 4 respectively. Define $\bar{s}_i := [\bar{s}_i(N); \ldots; \bar{s}_i(1)] \in \mathbb{R}^{(N-1)K+d}$, then for any input sequences $h_i \in \mathbb{R}^D$ takes from Problem 2, the output of neural network

$$\mathrm{Attn}_{\theta_1} \circ \cdots \circ \mathrm{Attn}_{\theta_{N+1}} \circ \mathrm{MLP}_{\theta_{N+2}} \circ \mathrm{EWML}_{\theta_{N+3}} \circ \cdots \circ \mathrm{EWML}_{\theta_{2N+1}}, \tag{3.20}$$

is

$$h_i = [x_i; y_i; w; \bar{p}_i; \bar{r}'_i; \bar{s}_i; \mathbf{0}; 1; t_i]. \tag{3.21}$$

For the sake of simplicity, we consider ReLU Attention layer and MLP layer are both a special kind of transformer. In this way, by Definition 9, (3.20) becomes

$$\mathrm{TF}_\theta^{N+2} \circ \mathrm{EWML}_\theta^{N-1}.$$

Next we calculate the error accumulation $|\bar{s}_i(j)[k] - s_i(j)[k]|$ based on Lemma 3 and Lemma 4.

**Lemma 6** (Error for $\bar{s}_i(j)$). Suppose the upper bounds $B_v, B_x > 0$ such that for any $k \in [K], j \in [N]$ and $i \in [n]$, $\|v_{j_k}\|_2 \leq B_v$, and $\|x_i\|_2 \leq B_x$. Let $r'_i(j) \in \mathbb{R}^K$ such that $r'_i(j)[k] := r'(v_{j+1_k}^\top p_i(j))$ follows Definition 2. Let $s_i(j) = R_i(j-1)V_{j+1}^\top \ldots R_i(N-2)V_N^\top \cdot R_i(N-1)u$ follows Definition 3. Let $\bar{r}'_i(j), g_i, \bar{s}_i(j)$ be the approximations for $r'_i(j), u(p_i(N), y_i), s_i(j)$ follows Lemma 3, Lemma 4 and Lemma 5 respectively. Let $B_{r'}$ be the upper bound of $\bar{r}'_i(j)[k]$ and $r'_i(j)[k]$ as defined in Lemma 3. Let $B_l$ be the upper bound of $g_i$ and $u(p_i(N), y_i)$ as defined in Lemma 4. Then for any $i \in [n+1], j \in [N], k \in [K]$,

$$\bar{s}_i(j)[k] \leq B_s, \quad \text{and} \quad |\bar{s}_i(j)[k] - s_i(j)[k]| \leq E_s^r \epsilon_r + E_s^{r'} \epsilon_{r'} + E_s^l \epsilon_l, \quad \text{where,}$$

$$P := \max\{\sqrt{K}, \sqrt{d}\} \tag{3.22}$$

$$B_s := \max_{j \in [N]}\{(P \cdot B_{r'}B_v)^{N-j}B_{r'}B_l\}, \tag{3.23}$$

$$E_s^r := \max_{j \in [N]}\{L_{r'}E_r PB_s B_v^2[\sum_{l=0}^{N-j-1}(B_{r'}B_vP)^l] + (B_{r'}B_vP)^{N-j}(B_lL_{r'}B_vE_r + B_{r'}L_lE_r)\},$$

$$E_s^{r'} := \max_{j \in [N]}\{PB_s B_v[\sum_{l=0}^{N-j-1}(B_{r'}B_vP)^l] + (B_{r'}B_vP)^{N-j}B_l\},$$

$$E_s^l := \max_{j \in [N]}\{(B_{r'}B_vP)^{N-j}B_{r'}\}. \tag{3.24}$$

Above, $B_s$ is the upper bound of $\bar{s}_i(j)[k]$ and $E_s^r, E_s^{r'}, E_s^l$ are the coefficients of $\epsilon_r, \epsilon'_r, \epsilon_l$ in the upper bounds of $|\bar{s}_i(j)[k] - s_i(j)[k]|$, respectively.

*Proof Sketch.* By Lemma 5, we manage to approximate $s_i(j)$ by $\bar{s}_i(j)$. By triangle inequality, we have $|\bar{s}_i(j)[k] - s_i(j)[k]| \leq |\bar{r}'_i(n-1)[k] - r'_i(n-1)[k]| \cdot |v_{n+1_k}^\top \bar{s}_i(n+1)| + |r'_i(n-1)[k]| \cdot |(v_{n+1_k}^\top \bar{s}_i(n+1)) - (v_{n+1_k}^\top s_i(n+1))|$. We bound these four terms separately. By Lemma 3, $|\bar{r}'_i(n-1)[k] - r'_i(n-1)[k]|$ is bounded by $\epsilon_{r'} + L_{r'}B_vE_r\epsilon_r$. We then use induction to establish upper bounds for $\bar{s}_i(j)[k]$ and $|\bar{s}_i(j)[k] - s_i(j)[k]|$. See Appendix D.6 for a detailed proof. □

Lemma 6 offers the explicit form of the error $|\bar{s}_i(j)[k] - s_i(j)[k]|$, which is crucial for calculating the error $\|\nabla_w \bar{\mathcal{L}}_n(w) - \nabla_w \mathcal{L}_n(w)\|_2$ in Theorem 1.

**Step 3.** Combining the above, we prove the existence of a neural network, that implements $L$ in-context GD steps on our $N$-layer neural network. And finally we arrive our main result: a neural network $\mathcal{T}$ for Problem 2.

**Theorem 1** (In-Context Gradient Descent on $N$-layer NNs). *Fix any $B_v, \eta, \epsilon > 0, L \geq 1$. For any input sequences takes from (2.1), their exist upper bounds $B_x, B_y$ such that for any $i \in [n]$, $\|y_i\|_2 \leq B_y, \|x_i\|_2 \leq B_x$. Assume functions $r(t)$, $r'(t)$ and $u(t, y)[k]$ are $L_r, L_{r'}, L_l$-Lipschitz continuous. Suppose $\mathcal{W}$ is a closed domain such that for any $j \in [N-1]$ and $k \in [K]$,*

$$\mathcal{W} \subset \left\{ w = [v_{j_k}] \in \mathbb{R}^{D_N} : \|v_{j_k}\|_2 \leq B_v \right\},$$

*and $\mathrm{Proj}_{\mathcal{W}}$ project $w$ into bounded domain $\mathcal{W}$. Assume $\mathrm{Proj}_{\mathcal{W}} = \mathrm{MLP}_\theta$ for some MLP layer with hidden dimension $D_w$ parameters $\|\theta\| \leq C_w$. If functions $r(t)$, $r'(t)$ and $u(t, y)[k]$ are $C^4$-smoothness, then for any $\epsilon > 0$, there exists a transformer model $\mathrm{NN}_\theta$ with $(2N + 4)L$ hidden layers consists of $L$ neural network blocks $\mathrm{TF}_\theta^{N+2} \circ \mathrm{EWML}_\theta^N \circ \mathrm{TF}_\theta^2$,*

$$\mathrm{NN}_\theta := \mathrm{TF}_\theta^{N+2} \circ \mathrm{EWML}_\theta^N \circ \mathrm{TF}_\theta^2 \circ \ldots \circ \mathrm{TF}_\theta^{N+2} \circ \mathrm{EWML}_\theta^N \circ \mathrm{TF}_\theta^2,$$

*such that the heads number $M^l$, parameter dimensions $D^l$, and the parameter norms $B_{\theta^l}$ suffice*

$$\max_{l \in [(2N+4)L]} M^l \leq \widetilde{O}(\epsilon^{-2}), \quad \max_{l \in [(2N+4)L]} D^l \leq O(NK^2) + D_w, \quad \max_{l \in [(2N+4)L]} B_{\theta^l} \leq O(\eta) + C_w + 1,$$

*where $\widetilde{O}(\cdot)$ hides the constants that depend on $d, K, N$, the radius parameters $B_x, B_y, B_v$ and the smoothness of $r$ and $\ell$. And this neural network such that for any input sequences $H^{(0)}$, take from (2.1), $\mathrm{NN}_\theta(H^{(0)})$ implements $L$ steps in-context gradient descent on risk Eqn (2.2): For every $l \in [L]$, the $(2N + 4)l$-th layer outputs $h_i^{((2N+4)l)} = [x_i; y_i; \overline{w}^{(l)}; \mathbf{0}; 1; t_i]$ for every $i \in [n+1]$, and approximation gradients $\overline{w}^{(l)}$ such that*

$$\overline{w}^{(l)} = \mathrm{Proj}_{\mathcal{W}}(\overline{w}^{(l-1)} - \eta \nabla \mathcal{L}_n(\overline{w}^{(l-1)}) + \epsilon^{(l-1)}), \quad \overline{w}^{(0)} = \mathbf{0},$$

*where $\|\epsilon^{(l-1)}\|_2 \leq \eta\epsilon$ is an error term.*

*Proof Sketch.* Let the first $2N + 2$ layers of $\mathrm{NN}_\theta$ are Transformers and EWMLs constructed in Lemma 2, Lemma 3, Lemma 4, and Lemma 5. Explicitly, we design the last two layers to implement the gradient descent step (Lemma 1). We then establish the upper bounds for error $\|\nabla_w \overline{\mathcal{L}}_n(w) - \nabla_w \mathcal{L}_n(w)\|_2$, where $\nabla_w \overline{\mathcal{L}}_n(w)$, derived from the outputs of $\mathrm{NN}_\theta$, approximates $\nabla_w \mathcal{L}_n(w)$. Next, for any $\epsilon > 0$, we select appropriate parameters $\epsilon_l, \epsilon_r$ and $\epsilon_{r'}$ to ensure that $\|\nabla_w \overline{\mathcal{L}}_n(\overline{w}^{(l-1)}) - \nabla_w \mathcal{L}_n(\overline{w}^{(l-1)})\|_2 \leq \epsilon$ holds for any $l \in [L]$. Please see Appendix D.7 for a detailed proof. $\square$

As a direct result, the neural networks $\mathrm{NN}_\theta$ constructed earlier is able to approximate the true gradient descent trajectory $\{w_{\mathrm{GD}}^l\}_{l \geq 0}$, defined by $w_{\mathrm{GD}}^0 = \mathbf{0}$ and $w_{\mathrm{GD}}^{l+1} = w_{\mathrm{GD}}^l - \eta \nabla_w \mathcal{L}_n(w_{\mathrm{GD}})$ for any $l \geq 0$. Consequently, Theorem 1 motivates us to investigate the error accumulation under setting

$$\overline{w}^{(l)} = \mathrm{Proj}_{\mathcal{W}}(\overline{w}^{(l-1)} - \eta \nabla \mathcal{L}_n(\overline{w}^{(l-1)}) + \epsilon^{(l-1)}), \quad \overline{w}^{(0)} = \mathbf{0},$$

where $\|\epsilon^{(l-1)}\|_2 \leq \eta\epsilon$ represents error terms. Moreover, Corollary 1.1 shows $\mathrm{NN}_\theta$ constructed in Theorem 1 implements $L$ steps ICGD with exponential error accumulation to the true GD paths.

**Corollary 1.1** (Error for implementing ICGD on $N$-layer neural network). *Fix $L \geq 1$, under the same setting as Theorem 1, $(2N + 4)L$-layer neural networks $\mathrm{NN}_\theta$ approximates the true gradient descent trajectory $\{w_{\mathrm{GD}}^l\}_{l \geq 0} \in \mathbb{R}^{D_N}$ with the error accumulation $\|\overline{w}^l - w_{\mathrm{GD}}^l\|_2 \leq L_f^{-1}(1 + nL_f)^l\epsilon$, where $L_f$ denotes the Lipschitz constant of $\mathcal{L}_n(w)$ within $\mathcal{W}$.*

*Proof.* Please see Appendix D.8 for a detailed proof. $\square$

## 4 DISCUSSION AND CONCLUSION

We provide an explicit characterization of the ICL capabilities of a transformer model in approximating the gradient descent training process of a $N$-layer feed-forward neural network. Our results include approximation (Theorem 1) and convergence (Corollary 1.1) guarantees. We further extend our analysis in two ways: (i) from $N$-layer networks with the same input and output dimensions to scenarios with arbitrary dimensions (Appendix E); (ii) from ReLU-transformers (aligned with (Bai et al., 2023)) to more practical Softmax-transformers for ICGD of $N$-layer neural network (Appendix F). We support our theory with numerical validations in Appendix G, and apply our results to learn the score function of the diffusion model through ICL in Appendix H. Please see the related works, a detailed comparison with (Wang et al.), broader impact, and limitations in Appendix B.

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

# Appendix

## A  Informal Version of Results

In this section, we show the informal version of our main results.

**Theorem 2** (ICGD on $N$-layer NNs, informal version of Theorem 1). Assume functions $r(t)$, $r'(t)$ and $u(t, y)[k]$ (Definition 1) are $C^4$-smoothness. Let all parameter and data are bounded, then their exist a explicit constructed transformer capable of solving Problem 1 on $N$-layer NNs.

**Theorem 3** (ICGD on General Risk Function, informal version of Theorem 6). Assume all parameters and data are bounded. Let $l(w, x_i, y_i)$ be a loss function with $L$-Lipschitz gradient. Let $\mathcal{L}_n(w) = \frac{1}{n} \sum_{i=1}^{n} \ell(w, x_i, y_i)$ denote the empirical loss function, then there exists a transformer $\text{NN}_\theta$, such that for any input sequences $H^{(0)}$, take from (2.1), $\text{NN}_\theta(H^{(0)})$ solves Problem 1 on $\mathcal{L}_n(w)$.

## B  Related Works, Broader Impact and Limitations

In this section, we show the related works, broader impact and limitations.

### B.1  Related Works

**In-Context Learning.**  Large language models (LLMs) demonstrate the in-context learning (ICL) ability (Brown, 2020), an ability to flexibly adjust their prediction based on additional data given in context. In recent years, a number of studies investigate enhancing ICL capabilities (Chen et al., 2022; Gu et al., 2023; Shi et al., 2023), exploring influencing factors (Shin et al., 2022; Yoo et al., 2022), and interpreting ICL theoretically (Xie et al., 2021; Wies et al., 2024; Panwar et al., 2023; Li et al., 2023; Bai et al., 2023; Dai et al., 2022). The works most relevant to ours are as follows. (Von Oswald et al., 2023) showed that linear attention-only Transformers with manually set parameters closely resemble models trained via gradient descent. (Bai et al., 2023) providing a more efficient construction for in-context gradient descent and established quantitative error bounds for simulating multi-step gradient descent. However, these results focused on simple ICL algorithms or specific tasks like least squares, ridge regression, and gradient descent on two-layer neural networks. These algorithms are inadequate for practical applications. For example: (i) Approximating the diffusion score function requires neural networks with multiple layers (Chen et al., 2023). (ii) Approximating the indicator function requires at least 3-layer networks (Safran and Shamir, 2017). Therefore, the explicit construction of transformers to implement in-context gradient descent (ICGD) on deep models is necessary to better align with real-world in-context settings. Our work achieves this by analyzing the gradient descent on $N$-layer neural networks through the use of ICL. We provide a more efficient construction for in-context gradient descent. Furthermore, we extend our analysis to $\text{Softmax}$-transformer in Appendix F to better align with real-world uses.

**In-Context Gradient Descent on Deep Models (Wang et al.; Panigrahi et al., 2023).**  A work similar to ours is (Wang et al.). It constructs a family of transformers with flexible activation functions to implement multiple steps of ICGD on deep neural networks. This work emphasizes the generality of activation functions and demonstrates the theoretical feasibility of such constructions. Our work adopts a different approach by enhancing the efficiency of transformers and better aligning with practical applications. We introduce the following novelties:

- **More structured and efficient transformer architecture.** While the work (Wang et al.) uses a $O(N^2 L)$-layer transformer to approximate $L$ gradient descent steps on $N$-layer neural networks, our approach achieves more efficient simulation for ICGD. We approximate specific terms in the gradient expression to reduce computational costs, requiring only a $(2N+4)L$-layer transformer for $L$ gradient descent steps. Our method focuses on selecting and approximating the most impactful intermediate terms in the explicit gradient descent expression (Lemmas 3 to 5), optimizing layer complexity to $O(NL)$.

- **Less restrictive input and output dimensions for $N$-layer neural networks.** The work (Wang et al.) simplifies the output of $N$-layer networks to a scalar. Our work expands this by considering cases where output dimensions exceed one, as detailed in Appendix E. This includes scenarios where input and output dimensions differ.

- **More practical transformer model.** The work (Wang et al.) discusses activation functions in the attention layer that meet a general decay condition ((Wang et al., Definition 2.3)) without consid-

ering the Softmax activation function. We extend our analysis to include Softmax-transformers. Our analysis reflects more realistic applications, as detailed in Appendix F.

- **More advanced and complicated applications.** The work (Wang et al.) discusses the applications to functions, including indicators, linear, and smooth functions. We explore more advanced and complicated scenarios, i.e., the score function in diffusion models discussed in Appendix H. The score function (Chen et al., 2023) falls outside the smooth function class. This enhancement broadens the applicability of our results.

Another work similar to ours is (Panigrahi et al., 2023). It proposes a new efficient construction, Transformer in Transformer (TINT), to allow a transformer to simulate and finetune more complex models (e.g., one transformer). The main distinction between ours and (Panigrahi et al., 2023) lies in the different aims: Our approach focuses on using a standard transformer for the simulator (with a minor modification: the "element-wise multiplication layer"), and we provide a theoretical understanding of how a standard transformer can learn the ICGD of an $N$-layer network using ICL. In contrast, the work (Panigrahi et al., 2023) aims to build even stronger transformers by introducing several structural modifications that enable running gradient descent on auxiliary transformers. While it demonstrates in-context gradient descent for a more advanced model, i.e., one transformer, our work offers the following potential advantages:

- **Explicit transformer construction.** We provide an explicit construction of the transformer, whereas the work (Panigrahi et al., 2023) does not detail the explicit construction of model parameters within their transformer.

- **Exact gradient descent.** We compute the exact and explicit gradient descent for an $N$-layer network (Lemma 1). Building on this, we employ the transformer's ICL to perform gradient descent on all parameters. However, the work (Panigrahi et al., 2023) stops the gradient computation through attention scores in the self-attention layer and only updates the value parameter in the self-attention module. Additionally, it uses Taylor expansion to approximate the gradient.

- **Rigorous error and convergence guarantees.** We provide rigorous gradient descent approximation errors (for multiple steps) and convergence guarantees for the ICGD on an $N$-layer network (Corollary 1.1 and Lemma 14). However, the work (Panigrahi et al., 2023) only presents the gradient approximation error for each specific part of the parameters in a single step.

- **Attention layer better aligned with practice.** Our analysis is based on ReLU-attention (Theorem 1) or Softmax-attention (Theorem 6), whereas the work (Panigrahi et al., 2023) utilizes linear attention. Our choice of attention layer better aligns with practical applications.

### B.2 BROADER IMPACT

This theoretical work aims to shed light on the foundations of large transformer-based models and is not expected to have negative social impacts.

### B.3 LIMITATIONS

Our work has the following four limitations:

- Although we provide a theoretical guarantee for the ICL of the Softmax-Transformer to approximate gradient descent in $N$-layer NN, characterizing the weight matrices construction in Softmax-Transformer remains challenging. This motivates us to rethink transformer universality and explore more accurate proof techniques for ICL in Softmax-Transformer, which we leave for future work.

- The hidden dimension and MLP dimension of the transformer in Theorem 1 are both $\widetilde{O}(NK^2) + D_w$, which is very large. The reason for the large dimensions is that if we use ICL to perform ICGD on the $N$-layer network, we need to allow the transformer to realize the $N$-layer network parameters. This means that it is reasonable for the input dimension to be so large. However, it is possible to reduce the hidden dimension and MLP dimension of the transformer through smarter construction. We leave this for future work.

- The generalization capabilities are limited compared with traditional transformers. In our setting, the pretraining task refers to using in-context examples generated by an $N$-layer network for a given $N$. Specifically, during pretraining, the distribution of the $N$-layer network parameters is predetermined (e.g., $N(0, I)$). The input data distribution of $N$-layer network for generating the

in-context examples is also predetermined (e.g., $N(-2, I)$). The generalization capabilities include the following two aspects: (i) Varying the input data distribution for the $N$-layer network to generate the in-context examples. For example, we change the input data distribution from $N(-2, I)$ to $0.9N(-2, I) + 0.1N(2, I)$ during the testing in Appendix G.1. (ii) Varying the distribution of the $N$-layer network parameters. For example, we change the distribution from $N(0, I)$ to $N(0.5, I)$ in Appendix G.2. The above points lead to differences between the distributions of in-context examples during pretraining and testing. However, we must generate the in-context examples by the $N$-layer network with the same hyperparameters, including the network width and depth. We leave the theoretical analysis of broader generalization capabilities for future work.

- In theory, the FLOPs (Hoffmann et al., 2022) required to perform one forward pass of the transformer are greater than those required for the direct training of an $N$-layer network. (i) For the forward pass of the transformer, the FLOPs for in-context learning (ICL) are $O(nLN^3K^5/\epsilon^2)$, where $\epsilon$ is the approximation error in the sum of ReLU. (ii) For direct training of the $N$-layer network, the FLOPs without ICL are $O(nLNK^2)$. Therefore, the FLOPs required for ICL exceed those needed for direct training of the $N$-layer network. However, experimental results in Appendix G demonstrate that the transformer with ICL can achieve the performance of a trained 6-layer network using fewer FLOPs in practice (3.3 billion vs. 7.6 billion FLOPs). This finding encourages further exploration of more efficient architectures. We also leave this topic for future research.

## C  SUPPLEMENTARY THEORETICAL BACKGROUNDS

Here we present some ideas we built on.

### C.1  TRANSFORMERS

Lastly, we introduce key components for constructing a transformer for ICGD: ReLU-Attention, MLP, and element-wise multiplication layers. We begin with the ReLU-Attention layer.

**Definition 7** (ReLU-Attention Layer). For any input sequence $H \in \mathbb{R}^{D \times n}$, an $M$-head ReLU-attention layer with parameters $\theta = \{Q_m, K_m, V_m\}_{m \in [M]}$ outputs

$$\text{Attn}_\theta(H) := H + \frac{1}{n} \sum_{m=1}^{M} (V_m H) \cdot \sigma((Q_m H)^\top (K_m H)),$$

where $Q_m, K_m, V_m \in \mathbb{R}^{D \times D}$ and $\sigma(\cdot)$ is element-wise ReLU activation function. In vector form, for each token $h_i \in \mathbb{R}^D$ in $H$, it outputs $[\text{Attn}_\theta(H)]_i = h_i + \frac{1}{n} \sum_{m=1}^{M} \sum_{s=1}^{n} \sigma(\langle Q_m h_i, K_m h_s \rangle) \cdot V_m h_s$.

Notably, Definition 7 uses normalized ReLU activation $\sigma/n$, instead of the standard Softmax. We adopt this for technical convenience following (Bai et al., 2023). Next we define the MLP layer.

**Definition 8** (MLP Layer). For any input sequence $H \in \mathbb{R}^{D \times n}$, an $d'$-hidden dimensions MLP layer with parameters $\theta = (W_1, W_2)$ outputs $\text{MLP}_\theta(H) := H + W_2 \sigma(W_1 H)$, where $W_1 \in \mathbb{R}^{d' \times D}$, $W_2 \in \mathbb{R}^{D \times d'}$ and $\sigma(\cdot) : \mathbb{R} \to \mathbb{R}$ is element-wise ReLU activation function. In vector form, for each token $h_i \in \mathbb{R}^D$ in $H$, it outputs $\text{MLP}_\theta(H)_i := h_i + W_2 \sigma(W_1 h_i)$.

Then, we consider a transformer architecture with $L \geq 1$ transformer layers, each consisting of a self-attention layer followed by an MLP layer.

**Definition 9** (Transformer). For any input sequence $H \in \mathbb{R}^{D \times n}$, an $L$-layer transformer with parameters $\theta = \{\theta_{\text{Attn}}, \theta_{\text{MLP}}\}$ outputs

$$\text{TF}_\theta^L(H) := \text{MLP}_{\theta_{\text{mlp}}^{(L)}} \circ \text{Attn}_{\theta_{\text{attn}}^{(L)}} \dots \text{MLP}_{\theta_{\text{mlp}}^{(1)}} \circ \text{Attn}_{\theta_{\text{attn}}^{(1)}}(H),$$

where $\theta = \{\theta_{\text{Attn}}, \theta_{\text{MLP}}\}$ consists of Attention layers $\theta_{\text{Attn}} = \{(Q_m^l, K_m^l, V_m^l)\}_{l \in [L], m \in [M^l]}$ and MLP layers $\theta_{\text{MLP}} = \{(W_1^l, W_2^l)\}_{l \in [L]}$. Above, for any $l \in [L], m \in [M^l], Q_m^l, K_m^l, V_m^l \in \mathbb{R}^{D \times D}$ and $(W_1^l, W_2^l) \in \mathbb{R}^{d' \times D} \times \mathbb{R}^{D \times d'}$ In this section, we consider ReLU Attention layer and MLP layer are both a special kind of 1-layer transformer, which is for technical convenience.

For later proof use, we define the norm for $L$-layer transformer $\text{TF}_\theta$ as:

$$B_\theta := \max_{l \in [L]} \left\{ \max_{m \in [M]} \left\{ \|Q_m^l\|_1, \|K_m^l\|_1 \right\} + \sum_{i=1}^{m} \|V_m^l\|_1 + \|W_1\|_1 + \|W_2\|_1 \right\}. \tag{C.1}$$

The choice of operation norm and max/sum operation is for convenience in later proof only, as our result depends only on $B_\theta$.

### C.2  RELU PROVABLY APPROXIMATES SMOOTH $k$-VARIABLE FUNCTIONS

Following lemma expresses that the smoothness enables the approximability of sum of ReLU.

**Lemma 7** (Approximating Smooth $k$-Variable Functions, modified from Proposition A.1 of (Bai et al., 2023)). For any $\epsilon, C_l > 0, R \geq 1$. If function $g : \mathbb{R}^k \to \mathbb{R}$ such that for $s := \lceil (k-1)/2 \rceil + 1$, $g$ is a $C^s$ function on $B_\infty^k(R)$, and for all $i \in \{0, 1, \dots, s\}$,

$$\sup_{z \in B_\infty^k(R)} \|\nabla^i g(z)\|_\infty \leq L_i, \quad \max_{0 \leq i \leq s} L_i R^i \leq C_l,$$

then function $g$ is $(\epsilon, R, H, C)$-approximable by sum of ReLUs (Definition 4) with $H \leq C(k)C_l^2 \log(1 + C_l/\epsilon)/\epsilon^2$ and $C \leq C(k)C_l$ where $C(k)$ is a constant that depends only on $k$.

## D    PROOFS OF MAIN TEXT

### D.1    PROOF OF LEMMA 1

**Lemma 8** (Lemma 1 Restated: Decomposition of One Gradient Descent Step). Fix any $B_v, \eta > 0$. Suppose loss function $\mathcal{L}_n(w)$ on $n$ data points $\{(x_i, y_i)\}_{i \in [n]}$ follows (2.2). Suppose closed domain $\mathcal{W}$ and projection function $\mathrm{Proj}_{\mathcal{W}}(w)$ follows (3.4). Let $A_i(j), r_i'(j), R_i(j), V_j$ be as defined in Definition 2. Then the explicit form of gradient $\nabla \mathcal{L}_n(w)$ becomes

$$\nabla \mathcal{L}_n(w) = \frac{1}{2n} \sum_{i=1}^{n} \begin{bmatrix} A_i(1) \\ \vdots \\ A_i(N) \end{bmatrix},$$

where $A_i(j)$ denote the derivative of $\ell(p_i(N), y_i)$ with respect to the parameters in the $j$-th layer,

$$A_i(j) = \begin{cases} (R_i(N-1) \cdot V_N \cdot \ldots \cdot R_i(j-1) \cdot \left[ \mathbf{I}_{K \times K} \otimes p_i(j-1)^\top \right])^\top \cdot (\frac{\partial \ell(p_i(N), y_i)}{\partial p_i(N)})^\top, & j \neq N \\ (R_i(N-1) \cdot \left[ \mathbf{I}_{d \times d} \otimes p_i(N-1)^\top \right])^\top \cdot (\frac{\partial \ell(p_i(N), y_i)}{\partial p_i(N)})^\top, & j = N. \end{cases}$$

*Proof of Lemma 1.* We start with calculating $\nabla_w \mathcal{L}_n(w)$. By chain rule and (2.2),

$$\underbrace{\nabla_w \mathcal{L}_n(w)}_{\mathbb{R}^{D_N \times 1}} = \frac{1}{2n} \sum_{i=1}^{n} \underbrace{[\frac{\partial}{\partial w} p_i(N)]^\top}_{\mathbb{R}^{D_N \times d}} \cdot \underbrace{[\frac{\partial}{\partial p_i(N)} \ell(p_i(N), y_i)]^\top}_{\mathbb{R}^{d \times 1}} \qquad \text{(By (2.2) and chain rule)}$$

Thus we only need to calculate $\frac{\partial}{\partial w} p_i(N)$. For a vector $x$ and a function $r : \mathbb{R} \to \mathbb{R}$, we use $\mathbf{r}(x)$ to denote the vector that $i$-th coordinate is $r(x_i)$. Let $R_i(j), V_j$ follows Definition 2, then it holds

$$\underbrace{\frac{\partial p_i(N)}{\partial w}}_{\mathbb{R}^{d \times D_N}} = \underbrace{\frac{\partial \mathbf{r}(\overbrace{V_N}^{\mathbb{R}^{d \times K}} \cdot \overbrace{p_i(N-2)}^{\mathbb{R}^K})}{\partial w}}_{\mathbb{R}^{d \times D_N}} \qquad \text{(By Definition 1)}$$

$$= \underbrace{\frac{\partial \mathbf{r}(V_N \cdot p_i(N-1))}{\partial V_N \cdot p_i(N-1)}}_{\mathbb{R}^{d \times d}} \cdot \underbrace{\frac{\partial V_N \cdot p_i(N-1)}{\partial w}}_{\mathbb{R}^{d \times D_N}} \qquad \text{(By chain rule)}$$

$$= \mathrm{diag}\{r'(v_{N_1}^\top p_i(N-1)), \ldots, r'(v_{N_K}^\top p_i(N-1))\} \cdot \frac{\partial V_N \cdot p_i(N-1)}{\partial w}$$
$$\qquad \text{(By Definition 2)}$$

$$= R_i(N-1) \cdot \frac{\partial V_N \cdot p_i(N-1)}{\partial w}. \qquad (D.1)$$

Notice that for any $k \in [d]$, $v_{N_k}$ is a part of $w$, thus

$$\frac{\partial v_{N_k}}{\partial w} = [\overbrace{\mathbf{0}}^{D_{N-1}+(k-1)K} \quad \overbrace{\mathbf{I}}^{K} \quad \overbrace{\mathbf{0}}^{D_N - D_{N-1} - kK}] \in \mathbb{R}^{d \times D_N}. \qquad (D.2)$$

Therefore, letting $\otimes$ denotes Kronecker product, it holds

$$\frac{\partial V_N \cdot p_i(N-1)}{\partial w}$$

$$
= \begin{bmatrix} v_{N_1}^\top \cdot \frac{\partial p_i(N-1)}{\partial w} + p_i(N-1)^\top \cdot \frac{\partial v_{N_1}}{\partial w} \\ \vdots \\ v_{N_d}^\top \cdot \cdot \frac{\partial p_i(N-1)}{\partial w} + p_i(N-1)^\top \cdot \frac{\partial v_{N_d}}{\partial w} \end{bmatrix} \qquad \text{(By chain rule and product rule)}
$$

$$
= V_N \cdot \frac{\partial p_i(N-1)}{\partial w} + \left[ \mathbf{0}_{D_{N-1}}; \mathbf{I}_{K \times K} \otimes p_i(N-1)^\top \right], \tag{D.3}
$$

where the last step follows from the definition of $V_N$ (i.e., Definition 2) and (D.2).

Substituting (D.3) into (D.1), we obtain

$$
\frac{\partial p_i(N)}{\partial w} = R_i(N-1) \cdot (V_N \cdot \frac{\partial p_i(N-1)}{\partial w} + \left[ \mathbf{0}_{D_{N-1}}; \mathbf{I}_{d \times d} \otimes p_i(N-1)^\top \right]).
$$

Similarly, for any $j \in [N]$, we can proof

$$
\frac{\partial p_i(j)}{\partial w} = R_i(j-1) \cdot (V_j \cdot \frac{\partial p_i(j-1)}{\partial w} + \left[ \mathbf{0}_{D_{j-1}}; \mathbf{I}_{K \times K} \otimes p_i(j-1)^\top; \mathbf{0}_{D_N - D_j} \right]). \tag{D.4}
$$

By the recursion formula (D.4), for any $j \in [N-1]$, we calculate $A_i(j)$ as follows,

$$
A_i(j) = \left( \left( \frac{\partial \ell(p_i(N), y_i)}{\partial p_i(N)} \cdot \frac{\partial p_i(N)}{\partial w} \right)^\top \right) [D_{j-1} : D_j] \qquad \text{(By Definition 2)}
$$

$$
= (\frac{\partial p_i(N)}{\partial w})^\top \cdot (\frac{\partial \ell(p_i(N), y_i)}{\partial p_i(N)})^\top [D_{j-1} : D_j] \qquad \text{(By transpose property)}
$$

$$
= (\frac{\partial p_i(N)}{\partial w})^\top [*, D_{j-1} : D_j] \cdot (\frac{\partial \ell(p_i(N), y_i)}{\partial p_i(N)})^\top
$$

$$
= (R_i(N-1) \cdot V_N \cdot \ldots \cdot R_i(j-1) \cdot \left[ \mathbf{I}_{K \times K} \otimes p_i(j-1)^\top \right])^\top \cdot (\frac{\partial \ell(p_i(N), y_i)}{\partial p_i(N)})^\top,
$$
$$
\text{(By (D.4))}
$$

where $M[*, a : b]$ denotes a sub-matrix of $M$, which includes all the columns but only the rows from the $a$-th row to the $b$-th row of $A$. Similarly, for $j = N$, it holds

$$
A_i(N) = (R_i(N-1) \cdot \left[ \mathbf{I}_{d \times d} \otimes p_i(N-1)^\top \right])^\top \cdot (\frac{\partial \ell(p_i(N), y_i)}{\partial p_i(N)})^\top.
$$

Thus we completes the proof. $\qquad \square$

## D.2 Proof of Lemma 2

**Lemma 9** (Lemma 2 Restated: Approximate $p_i(j)$)**.** Let upper bounds $B_v, B_x > 0$ such that for any $k \in [K], j \in [N]$ and $i \in [n]$, $\|v_{j_k}\|_2 \le B_v$, and $\|x_i\|_2 \le B_x$. For any $j \in [N], i \in [n]$, define

$$
B_r^j := \max_{|t| \le B_v B_r^{j-1}} |r(t)|, \quad B_r^0 := B_x, \quad \text{and} \quad B_r := \max_j B_r^j.
$$

Let function $r(t)$ be $(\epsilon_r, R_1, M_1, C_1)$-approximable for $R_1 = \max\{B_v B_r, 1\}$, $M_1 \le \widetilde{\mathcal{O}}(C_1^2 \epsilon_r^{-2})$, where $C_1$ depends only on $R_1$ and the $C^2$-smoothness of $r$. Then, for any $\epsilon_r > 0$, there exist $N$ attention layers $\text{Attn}_{\theta_1}, \ldots, \text{Attn}_{\theta_N}$ such that for any input $h_i \in \mathbb{R}^D$ takes from (2.1), they map

$$
h_i = [x_i; y_i; w; \bar{p}_i(1); \ldots; \bar{p}_i(j-1); \mathbf{0}; 1; t_i] \xrightarrow{\text{Attn}_{\theta_j}} \widetilde{h}_i = [x_i; y_i; w; \bar{p}_i(1); \ldots; \bar{p}_i(j); \mathbf{0}; 1; t_i],
$$

where $\bar{p}_i(j)$ is approximation for $p_i(j)$ (Definition 1). In the expressions of $h_i$ and $\widetilde{h}_i$, the dimension of $\mathbf{0}$ differs. Specifically, the $\mathbf{0}$ in $h_i$ is larger than in $\widetilde{h}_i$. The dimensional difference between these $\mathbf{0}$ vectors equals the dimension of $\bar{p}_i(j)$. Suppose function $r$ is $L_r$-smooth in bounded domain $\mathcal{W}$, then for any $i \in [n+1], j \in [N], \bar{p}_i(j)$ such that

$$\bar{p}_i(j) = p_i(j) + \epsilon(i,j), \quad \|\epsilon(i,j)\|_2 \leq \begin{cases} (\sum_{l=0}^{j-1} K^{l/2} L_r^l B_v^l)\sqrt{K}\epsilon_r, & 1 \leq j \leq N-1 \\ (\sum_{l=0}^{N-1} K^{l/2} L_r^l B_v^l)\sqrt{d}\epsilon_r, & j = N \end{cases}.$$

Additionally, for any $j \in [N]$, the norm of parameters $B_{\theta_j}$ defined as (C.1) such that

$$B_{\theta_j} \leq 1 + KC_1.$$

*Proof of Lemma 2.* First we need to give a approximation for activation function $r(t)$. By our assumption and Definition 4, $r(t)$ is $(\epsilon_r, R_1, M_1, C_1)$-approximable by sum of ReLUs, there exists:

$$\bar{r}(t) = \sum_{m=1}^{M_1} c_m^1 \sigma(\langle a_m^1, [t;1]\rangle) \text{ with } \sum_{m=1}^{M_1} |c_m^1| \leq C_1, \|a_m^1\|_1 \leq 1, \forall m \in [M_1], \quad \text{(D.5)}$$

such that $\sup_{t \in [-R_1, R_1]} |\bar{r}(t) - r(t)| \leq \epsilon_r$. Let $\bar{p}_i(0) := p_i(0) = x_i$. Similar to $p_i(j)$ follows Definition 1, we pick $\bar{p}_i(j)$ such that for any $j \in [N]$,

$$\bar{p}_i(j)[k] := \bar{r}(v_{j_k}^\top \bar{p}_i(j-1)). \quad \text{(D.6)}$$

Fix any $j \in [N]$, suppose the input sequences $h_i = [x_i; y_i; w; \bar{p}_i(1); \ldots; \bar{p}_i(j-1); \mathbf{0}; 1; t_i]$. Then for every $m \in [M_1], k \in [K]$(or $k \in [d]$ if $j = N$), we define matrices $Q_{m,k}^j, K_{m,k}^j, V_{m,k}^j \in \mathbb{R}^{D \times D}$ such that for all $i \in [n+1]$,

$$Q_{m,k}^j h_i = \begin{bmatrix} a_m^1[1] \cdot \bar{p}_i(j-1) \\ a_m^1[2] \\ \mathbf{0} \end{bmatrix}, \quad K_{m,k}^j h_i = \begin{bmatrix} v_{j_k} \\ 1 \\ \mathbf{0} \end{bmatrix}, \quad V_{m,k}^j h_i = c_m^1 e_{j,k}^1, \quad \text{(D.7)}$$

where $e_{j,k}^1$ denotes the position unit vector of element $\bar{p}_i(j)[k]$ because this position only depends on $j, k$. Since input $h_i = [x_i; y_i; w; \bar{p}_i(1); \ldots; \bar{p}_i(j-1); \mathbf{0}; 1; t_i]$, those matrices indeed exist. In fact, it is simple to check that

$$Q_{m,k}^j = \begin{bmatrix} \mathbf{0} & a_m^1[1]\mathbf{I}_K(j) & \mathbf{0} & \mathbf{0} & \mathbf{0} \\ \mathbf{0} & \mathbf{0} & \mathbf{0} & a_m^1[2] & \mathbf{0} \\ \mathbf{0} & \mathbf{0} & \mathbf{0} & \mathbf{0} & \mathbf{0} \end{bmatrix},$$

$$K_{m,k}^j = \begin{bmatrix} \mathbf{0} & \mathbf{I}_K(j,k) & \mathbf{0} & \mathbf{0} & \mathbf{0} \\ \mathbf{0} & \mathbf{0} & \mathbf{0} & 1 & \mathbf{0} \\ \mathbf{0} & \mathbf{0} & \mathbf{0} & \mathbf{0} & \mathbf{0} \end{bmatrix},$$

$$V_{m,k}^j = \begin{bmatrix} \mathbf{0} & \mathbf{0} & \mathbf{0} & \mathbf{0} \\ \mathbf{0} & \mathbf{0} & c_m^1(j,k) & \mathbf{0} \\ \mathbf{0} & \mathbf{0} & \mathbf{0} & \mathbf{0} \end{bmatrix}, \quad \text{(D.8)}$$

are suffice to (D.7). $\mathbf{I}_K(j), \mathbf{I}_K(j,k), c_m^1(j,k)$ represents their positions are related to variables in parentheses. In Addition, by (C.1), notice that they have operator norm bounds

$$\max_{j,m,k} \|Q_{m,k}^j\|_1 \leq 1, \quad \max_{j,m,k} \|K_{m,k}^j\|_1 \leq 1, \quad \max_j \sum_{k,m} \|V_{m,k}^j\|_1 \leq KC_1.$$

Consequently, for any $j \in [N], B_{\theta_j} \leq 1 + C_1$.

By our construction follows (D.7), a simple calculation shows that

$$\sum_{m\in[M_1],k\in[K]} \sigma(\langle Q_{m,k}^j h_i, K_{m,k}^j h_s\rangle)V_{m,k}^j h_s$$

$$= \sum_{k=1}^{K}\sum_{m=1}^{M_1} c_m^1 \sigma(\langle a_m^1, [v_{j_k}^\top \bar{p}_i(j-1);1]\rangle)e_{j,k}^1 \qquad \text{(By our construction (D.7))}$$

$$= \sum_{k=1}^{K} (\bar{r}(v_{j_k}^\top \bar{p}_i(j-1)))e_{j,k}^1 \qquad \text{(By definition of } \bar{r} \text{ follows (D.5))}$$

$$= [\mathbf{0}; \bar{p}_i(j); \mathbf{0}]. \qquad \text{(By definition of } \bar{p}_i(j) \text{ follows (D.6))}$$

Therefore, by definition of ReLU Attention layer follows Definition 7, the output $\widetilde{h}_i$ becomes

$$\widetilde{h}_i = [\text{Attn}_{\theta_j}(h_i)]$$

$$= h_i + \frac{1}{n+1}\sum_{s=1}^{n+1}\sum_{m\in[M_1],k\in[K]} \sigma(\langle Q_{m,k}^j h_i, K_{m,k}^j h_s\rangle)V_{m,k}^j h_s$$

$$= h_i + \frac{1}{n+1}\sum_{s=1}^{n+1}(n+1)[\mathbf{0}; \bar{p}_i(j); \mathbf{0}]$$

$$= [x_i; y_i; w; \bar{p}_i(1); \dots; \bar{p}_i(j-1); \mathbf{0}; 1; t_i] + [\mathbf{0}, \bar{p}_i(j), \mathbf{0}]$$

$$= [x_i; y_i; w; \bar{p}_i(1); \dots; \bar{p}_i(j-1); \bar{p}_i(j); \mathbf{0}; 1; t_i].$$

Therefore, let the attention layer $\theta_j = \{(Q_{m,k}^j, K_{m,k}^j, V_{m,k}^j)\}_{(k,m)}$, we construct $\text{Attn}_{\theta_j}$ such that

$$h_i = [x_i; y_i; w; \bar{p}_i(1); \dots; \bar{p}_i(j-1); \mathbf{0}; 1; t_i] \xrightarrow{\text{Attn}_{\theta_j}} \widetilde{h}_i = [x_i; y_i; w; \bar{p}_i(1); \dots; \bar{p}_i(j); \mathbf{0}; 1; t_i].$$

In addition, by setting $R_1 = \max\{B_v B_r, 1\}$, the lemma then follows directly by induction on $j$. For the base case $j = 1$, it holds

$$|\bar{p}_i(1)[k] - p_i(1)[k]| = |\bar{r}_i(v_{1_k}^\top x_i)[k] - r(v_{1_k}^\top x_i)| \qquad \text{(By Definition 1)}$$

$$\leq \epsilon_r. \qquad \text{(By definition of } \bar{r} \text{ follows (D.5))}$$

Suppose the claim holds for iterate $j-1$ and function $r$ is $L_r$-smooth in bounded domain $\mathcal{W}$. Then for iterate $j$,

$$|\bar{p}_i(j)[k] - p_i(j)[k]|$$

$$\leq |\bar{p}_i(j)[k] - r(v_{j_k}^\top \bar{p}_i(j-1))| + |r(v_{j_k}^\top \bar{p}_i(j-1)) - p_i(j)[k]| \qquad \text{(By triangle inequality)}$$

$$\leq \epsilon_r + L_r\|v_{j_k}^\top\|_2\|\bar{p}_i(j-1) - p_i(j-1)\|_2 \qquad \text{(By (D.5) and Cauchy–Schwarz inequality)}$$

$$\leq \epsilon_r + \sqrt{K}L_r B_v(\epsilon_r \sum_{l=0}^{j-2} K^{l/2}L_r^l B_v^l) \qquad \text{(By inductive hypothesis)}$$

$$\leq \epsilon_r \sum_{l=0}^{j-1} K^{l/2}L_r^l B_v^l,$$

Thus, it holds

$$\|\bar{p}_i(j) - p_i(j)\|_2 = \sqrt{\sum_{k=1}^{K} |\bar{p}_i(j)[k] - p_i(j)[k]|^2}$$

$$\leq \sqrt{K}(\epsilon_r \sum_{l=0}^{j-1} K^{l/2} L_r^l B_v^l).$$

This finish the induction. Then for the output layer $j = N$, it holds

$$\|\bar{p}_i(N) - p_i(N)\|_2 = \sqrt{\sum_{k=1}^{d} |\bar{p}_i(N)[k] - p_i(N)[k]|^2}$$

$$\leq \sqrt{d}(\epsilon_r \sum_{l=0}^{N-1} K^{l/2} L_r^l B_v^l).$$

Thus we complete the proof. $\qquad\square$

## D.3 PROOF OF LEMMA 3

**Lemma 10** (Lemma 3 Restated: Approximate $r_i'(j)$). Let upper bounds $B_v, B_x > 0$ such that for any $k \in [K], j \in [N]$ and $i \in [n]$, $\|v_{j_k}\|_2 \leq B_v$, and $\|x_i\|_2 \leq B_x$. For any $j \in [N], i \in [n]$, define

$$B_r'^j := \max_{|t| \leq B_v B_{r'}^{j-1}} |r'(t)|, \quad B_{r'}^0 := B_x, \quad \text{and} \quad B_{r'} := \max_j B_{r'}^j.$$

Suppose function $r'(t)$ is $(\epsilon_{r'}, R_2, M_2, C_2)$-approximable for $R_2 = \max\{B_v B_{r'}, 1\}$, $M_2 \leq \widetilde{\mathcal{O}}(C_2^2 \epsilon_r'^{-2})$, where $C_2$ depends only on $R_2$ and the $C^2$-smoothness of $r'$. Then, for any $\epsilon_r > 0$, there exist an attention layer $\mathrm{Attn}_{\theta_{N+1}}$ such that for any input $h_i \in \mathbb{R}^D$ takes from (3.12), it maps

$$h_i = [x_i; y_i; w; \bar{p}_i; \mathbf{0}; 1; t_i] \xrightarrow{\mathrm{Attn}_{\theta_{N+1}}} \widetilde{h}_i = [x_i; y_i; w; \bar{p}_i; \bar{r}_i'; \mathbf{0}; 1; t_i],$$

where $\bar{r}_i'(j)$ is approximation for $r_i'(j)$ (Definition 2) and $\bar{r}_i' := [\bar{r}_i'(0); \dots; \bar{r}_i'(N-1)] \in \mathbb{R}^{(N-2)K+d}$. Similar to Lemma 2, in the expressions of $h_i$ and $\widetilde{h}_i$, the dimension of $\mathbf{0}$ differs. In addition, let $E_r$ be defined in (3.11), for any $i \in [n+1], j \in [N], k \in [K]$, $\bar{r}_i'(j)$ such that

$$\bar{r}_i'(j-1)[k] = r_i'(j-1)[k] + \epsilon(i,j,k), \quad |\epsilon(i,j,k)| \leq \epsilon_{r'} + L_{r'} B_v E_r \epsilon_r,$$

where $\epsilon_r$ denotes the error generated in approximating $r$ by sum of ReLUs $\bar{r}$ follows (D.5). Additionally, the norm of parameters $B_{\theta_{N+1}}$ defined as (C.1) such that $B_{\theta_{N+1}} \leq 1 + K(N-1)C_2$.

*Proof of Lemma 3.* By Definition 2, recall that for any $j \in [N], i \in [n+1], k \in [K]$,

$$r_i'(j)[k] = r'(v_{j+1_k}^\top p_i(j)). \tag{D.9}$$

Therefore we need to give a approximation for $r'$. By our assumption and Definition 4, $r'(t)$ is $(\epsilon_{r'}, R_2, M_2, C_2)$-approximable by sum of relus. In other words, there exists:

$$\bar{r}'(t) = \sum_{m=1}^{M_2} c_m^2 \sigma(\langle a_m^2, [t;1] \rangle) \text{ with } \sum_{m=1}^{M_2} |c_m^2| \leq C_2, \|a_m^2\|_2 \leq 1, \forall m \in [M_2], \tag{D.10}$$

such that $\sup_{t \in [-R_2, R_2]} |\bar{r}'(t) - r'(t)| \leq \epsilon_{r'}$. Similar to (D.9), we pick $\bar{r}_i'(j)$ such that

$$\bar{r}_i'(j)[k] := \bar{r}'(v_{j+1_k}^\top \bar{p}_i(j)). \tag{D.11}$$

To ensure (D.11), we construct our attention layer as follows: for every $j \in [N], m \in [M_2], k \in [K]$, we define matrices $Q_{j,m,k}^{N+1}, K_{j,m,k}^{N+1}, V_{j,m,k}^{N+1} \in \mathbb{R}^{D \times D}$ such that

$$
Q_{j,m,k}^{N+1} h_i = \begin{bmatrix} a_m^2[1] \cdot \bar{p}_i(j-1) \\ a_m^2[2] \\ \mathbf{0} \end{bmatrix}, \quad K_{j,m,k}^{N+1} h_i = \begin{bmatrix} v_{j_k} \\ 1 \\ \mathbf{0} \end{bmatrix}, \quad V_{j,m,k}^{N+1} h_i = c_m^2 e_{j,k}^2, \qquad \text{(D.12)}
$$

for all $i \in [n+1]$ and $e_{j,k}^2$ denotes the position unit vector of element $\bar{r}_i'(j)[k]$. Since input $h_i = [x_i; y_i; w; \bar{p}_i; \mathbf{0}; 1; t_i]$, similar to (D.8), those matrices indeed exist. In addition, they have operator norm bounds

$$
\max_{j,m,k} \|Q_{j,m,k}^{N+1}\|_1 \leq 1, \quad \max_{j,m,k} \|K_{j,m,k}^{N+1}\|_1 \leq 1, \quad \sum_{j,m,k} \|V_{j,m,k}^{N+1}\|_1 \leq K(N-1)C_2.
$$

Consequently, by definition of parameter norm follows (C.1), $B_{\theta_{N+1}} \leq 1 + K(N-1)C_2$.

A simple calculation shows that

$$
\sum_{j \in [N], m \in [M_2], k \in [K]} \sigma(\langle Q_{j,m,k}^{N+1} h_i, K_{j,m,k}^{N+1} h_s \rangle) V_{j,m,k}^{N+1} h_s
$$

$$
= \sum_{j=1}^{N} \sum_{k=1}^{K} \sum_{m=1}^{M_2} c_m^2 \sigma(\langle a_m^2, [v_{j_k}^\top \bar{p}_i(j-1); 1] \rangle) e_{j,k}^2 \qquad \left(\text{By our construction follows (D.12)}\right)
$$

$$
= \sum_{j=1}^{N} \sum_{k=1}^{K} (\bar{r}'(v_{j_k}^\top \bar{p}_i(j-1))) e_{j,k}^2 \qquad \left(\text{By definition of } \bar{r}' \text{ follows (D.5)}\right)
$$

$$
= [\mathbf{0}; \bar{r}_i'(0); \ldots; \bar{r}_i'(N-1); \mathbf{0}] \qquad \left(\text{By definition of } \bar{r}_i'(j) \text{ follows (D.11)}\right)
$$

$$
= [\mathbf{0}; \bar{r}_i'; \mathbf{0}], \qquad \left(\text{By definition of } \bar{r}_i'\right)
$$

Therefore, by definition of ReLU Attention layer follows Definition 7, the output $\widetilde{h}_i$ becomes

$$
\widetilde{h}_i = [\text{Attn}_{\theta_N}(h_i)]
$$

$$
= h_i + \frac{1}{n+1} \sum_{s=1}^{n+1} \sum_{j \in [N-1], m \in [M_2], k \in [K]} \sigma(\langle Q_{j,m,k}^N h_i, K_{j,m,k}^N h_s \rangle) V_{j,m,k}^N h_s
$$

$$
= h_i + \frac{1}{n+1} \sum_{s=1}^{n+1} (n+1)[\mathbf{0}; \bar{r}_i'; \mathbf{0}]
$$

$$
= [x_i; y_i; w; \bar{p}_i; \mathbf{0}; 1; t_i] + [\mathbf{0}; \bar{r}_i'; \mathbf{0}]
$$

$$
= [x_i; y_i; w; \bar{p}_i; \bar{r}_i'; \mathbf{0}; 1; t_i].
$$

Next, we calculate the error accumulation in this approximation layer. By our assumption, $R_2 = \max\{B_v B_{r'}, 1\}$. Thus, for any $j \in [N], k \in [K], i \in [n+1]$, it holds

$$
v_{j_k}^\top p_i(j-1) \leq R_2.
$$

As our assumption, we suppose function $r'$ is $L_r$-smooth in bounded domain $\mathcal{W}$. Combining above, the upper bound of error accumulation $|\bar{r}_i'(j)[k] - r_i'(j)[k]|$ becomes

$$
|\bar{r}_i'(j)[k] - r_i'(j)[k]|
$$

$$
\leq \left|\bar{r}_i'(j)[k] - r'(v_{j_k}^\top \bar{p}_i(j-1))\right| + \left|r'(v_{j_k}^\top \bar{p}_i(j-1)) - r_i'(j)[k]\right| \qquad \text{(By triangle inequality)}
$$

$$
\leq \epsilon_{r'} + L_{r'} \|v_{j_k}^\top\|_2 \|\bar{p}_i(j-1) - p_i(j-1)\|_2 \qquad \text{(By (D.10) and Cauchy–Schwarz inequality)}
$$

$$
\leq \epsilon_{r'} + L_{r'} B_v E_r \epsilon_r. \qquad \text{(By definition of } E_r \text{ follows (3.11))}
$$

Thus we complete the proof. □

### D.4 PROOF OF LEMMA 4

**Lemma 11** (Lemma 4 Restated: Approximate $\partial_1 \ell(p_i(N), y_i)$). Let upper bounds $B_v, B_x, > 0$ such that for any $k \in [K], j \in [N]$ and $i \in [n]$, $\|v_{j_k}\|_2 \leq B_v$, and $\|x_i\|_2 \leq B_x$. For any $k \in [d]$, suppose function $u(t, y)[k]$ be $(\epsilon_l, R_3, M_3^k, C_3^k)$-approximable for $R_3 = \max\{B_v B_r, B_y, 1\}$, $M_3 \leq \widetilde{\mathcal{O}}((C_3^k)^2 \epsilon_l^{-2})$, where $C_3^k$ depends only on $R_3^k$ and the $C^3$-smoothness of $u(t, y)[k]$. Then, there exists an MLP layer $\text{MLP}_{\theta_{N+2}}$ such that for any input sequences $h_i \in \mathbb{R}^D$ takes from (3.14), it maps

$$h_i = [x_i; y_i; w; \bar{p}_i; \bar{r}_i'; \mathbf{0}; 1; t_i] \xrightarrow{\text{MLP}_{\theta_{N+2}}} \widetilde{h}_i = [x_i; y_i; w; \bar{p}_i; \bar{r}_i'; g_i; \mathbf{0}; 1; t_i],$$

where $g_i \in \mathbb{R}^d$ is an approximation for $u(p_i(N), y_i)$. For any $k \in [d]$, assume $u(p_i(N), y_i)$ is $L_l$-Lipschitz continuous. Then the approximation $g_i$ such that,

$$g_i[k] = u(p_i(N), y_i)[k] + \epsilon(i, k), \quad \text{with} \quad |\epsilon(i, k)| \leq \epsilon_l + L_l E_r \epsilon_r.$$

Additionally, the parameters $\theta_{N+2}$ such that $B_{\theta_{N+2}} \leq \max\{R_3 + 1, C_3\}$.

*Proof of Lemma 4.* By our assumption and Definition 4, for any $k \in [d]$, function $u[k](t, y)$ is $(\epsilon_l, R_3, M_3^k, C_3^k)$-approximable by sum of relus, there exists :

$$g_k(t, y) = \sum_{m=1}^{M_3^k} c_m^{3,k} \sigma(\langle a_m^{3,k}, [t; y; 1] \rangle) \text{ with } \sum_{m=1}^{M_3^k} \left| c_m^{3,k} \right| \leq C_3, \|a_m^{3,k}\|_2 \leq 1, \forall m \in [M_3^k], \quad \text{(D.13)}$$

such that $\sup_{(t,y) \in [-R_3, R_3]^2} |g_k(t, y) - u[k](t, y)| \leq \epsilon_l$. Then we construct our MLP layer.

Let $M_3 := \sum_{k=1}^d M_3^k$, we pick matrices $W_1^{N+1} \in \mathbb{R}^{M_3 \times D}, W_2^{N+1} \in \mathbb{R}^{D \times M_3}$ such that for any $i \in [n+1], m \in [M_3]$,

$$W_1^{N+1} h_i = \begin{bmatrix} a_1^{3,1}[1] \cdot \bar{p}_i(N) + a_1^{3,1}[2] \cdot y_i + a_1^{3,1}[3] - R_3(1 - t_i) \\ \vdots \\ a_{M_3^1}^{3,1}[1] \cdot \bar{p}_i(N) + a_{M^1}^{3,1}[2] \cdot y_i + a_{M_3^1}^{3,1}[3] - R_3(1 - t_i) \\ \vdots \\ a_1^{3,d}[1] \cdot \bar{p}_i(N) + a_1^{3,d}[2] \cdot y_i + a_1^{3,d}[3] - R_3(1 - t_i) \\ \vdots \\ a_{M_3^d}^{3,d}[1] \cdot \bar{p}_i(N) + a_{M^d}^{3,d}[2] \cdot y_i + a_{M_3^d}^{3,d}[3] - R_3(1 - t_i) \end{bmatrix} \in \mathbb{R}^{M_3},$$

$$W_2^{N+1}[j, m] = c_m^{3,k} \cdot \mathbf{1}\{j = D_g^k, M_3^{k-1} < m \leq M_3^k\}, \quad \text{(D.14)}$$

where $D_g^k$ denotes the position of element $g_i[k]$. Since input $h_i = [x_i; y_i; w; \bar{p}_i; \bar{r}_i'; \mathbf{0}; 1; t_i]$, similar to (D.8), those matrices indeed exist. Furthermore, by (C.1), they have operator norm bounds

$$\|W_1^{N+1}\|_1 \leq R_3 + 1, \quad \|W_2^{N+1}\|_1 \leq C_3$$

Consequently, $B_{\theta_{N+2}} \leq \max\{R_3 + 1, C_3\}$.

By our construction (D.14), a simple calculation shows that

$$W_2^{N+1} \sigma(W_1^{N+1} h_i) = \sum_{k=1}^d \sum_{m=1}^{M_3^k} \sigma(\langle a_m^{3,k}, [\bar{p}_i(N); y_i; 1] \rangle - R_3(1 - t_i)) \cdot c_m^{3,k} e_{D_g^k}$$

$$= 1\{t_j = 1\} \cdot \begin{bmatrix} \mathbf{0} \\ g_1(\bar{p}_i(N), y_i) \\ \vdots \\ g_d(\bar{p}_i(N), y_i) \\ \mathbf{0} \end{bmatrix}.$$

For $k \in [d]$, we let $g_i[k] = 1\{t_j = 1\} \cdot g_k(\bar{p}_i(N), y_i)e_{D_g^k}$ for $i \in [n+1]$. Hence, $\text{MLP}_{\theta_{N+2}}$ maps

$$h_i = [x_i; y_i; w; \bar{p}_i; \bar{r}_i'; \mathbf{0}; 1; t_i] \xrightarrow{\text{MLP}_{\theta_{N+2}}} \widetilde{h}_i = [x_i; y_i; w; \bar{p}_i; \bar{r}_i'; g_i; \mathbf{0}; 1; t_i],$$

Next, we calculate the error generated in this approximation. By setting $R_3 = \max\{B_v B_r, B_y, 1\}$, for any $i \in [n+1]$, it holds

$$p_i(N) \le R_3, \quad y_i \le R_3$$

Moreover, as our assumption, we suppose function $\partial_1 \ell$ is $L_l$-smooth in bounded domain $\mathcal{W}$. Therefore, by the definition of the function $g$, for each $i \in [n]$, the error becomes

$$\begin{aligned}
&|g_i[k] - u(p_i(N), y_i)[k]| \\
&\le |g_i[k] - u(\bar{p}_i(N), y_i)[k]| + |u(\bar{p}_i(N), y_i)[k] - u(p_i(N), y_i)[k]| &&\text{(By triangle inequality)} \\
&\le \epsilon_l + L_l \|\bar{p}_i(N) - p_i(N)\|_2 &&\text{(By the definition of } g_k \text{ follows (D.13) and } L_l\text{-smooth assumption)} \\
&\le \epsilon_l + L_l E_r \epsilon_r,. &&\text{(By the definition of } E_r \text{ follows (3.11))}
\end{aligned}$$

Combining above, we complete the proof. $\qquad\square$

## D.5 PROOF OF LEMMA 5

**Lemma 12** (Lemma 5 Restated: Approximate $\bar{s}_t(j)$). Recall that $s_i(j) = r_i'(j-1) \odot (V_{j+1}^\top \cdot s_i(j+1))$ follows Definition 3. Let the initial input takes from (3.16). Then, there exist $N$ element-wise multiplication layers: $\text{EWML}_{\theta_{N+3}}, \dots, \text{EWML}_{\theta_{2N+2}}$ such that for input sequences, $j \in [N]$,

$$h_i = [x_i; y_i; w; \bar{p}_i; \bar{r}_i'; g_i; \bar{s}_i(N); \dots; \bar{s}_i(j+1); \mathbf{0}; 1; t_i],$$

they map $\text{EWML}_{\theta_{2N+3-j}}(h_i) = [x_i; y_i; w; \bar{p}_i; \bar{r}_i'; g_i; \bar{s}_i(N); \dots; \bar{s}_i(j); \mathbf{0}; 1; t_i]$, where the approximation $\bar{s}_i(j)$ is defined as recursive form: for any $i \in [n+1], j \in [N-1]$,

$$\bar{s}_i(j) := \begin{cases} \bar{r}_i'(j-1) \odot (V_{j+1}^\top \cdot \bar{s}_i(j+1)), & j \in [N-1] \\ \bar{r}_i'(N-1) \odot g_i, & j = N. \end{cases}$$

Additionally, for any $j \in [N]$, $B_{\theta_{N+2+j}}$ defined in (C.1) satisfies $B_{\theta_{N+2+j}} \le 1$.

*Proof of Lemma 5.* We give the construction of parameters directly. For every $j \in [N-1], k \in [K]$, we define matrices $Q_k^{2N+3-j}, K_k^{2N+3-j}, V_k^{2N+3-j} \in \mathbb{R}^{D \times D}$ such that for all $i \in [n+1]$,

$$Q_k^{2N+3-j} h_i = \begin{bmatrix} v_{j+1_1}[k] \\ \vdots \\ v_{j+1_K}[k] \\ \mathbf{0} \end{bmatrix}, \quad K_k^{2N+3-j} h_i = \begin{bmatrix} \bar{s}_i(j+1) \\ \mathbf{0} \end{bmatrix}, \quad V_k^{2N+3-j} h_i = \bar{r}_i'(j-1)[k] \cdot e_{j,k}^3,$$

$$\text{(D.15)}$$

where $e_{j,k}^3$ denotes the position unit vector of element $\bar{s}_i(j)[k]$.

Since input $h_i = [x_i; y_i; w; \bar{p}_i; \bar{r}'_i; g_i; \bar{s}_i(N); \ldots; \bar{s}_i(j+1); \mathbf{0}; 1; t_i]$, similar to (D.8), those matrices indeed exist. Thus, it is straightforward to check that

$$\sum_{k \in [K]} \gamma(\langle Q_k^{2N+3-j} h_i, K_k^{2N+3-j} h_i \rangle) V_k^{2N+3-j} h_i$$

$$= \sum_{k=1}^{K} (V_{j+1}^{\top}[k, *] \cdot \bar{s}_i(j+1)) \bar{r}'_i(j-1)[k] e_{j,k}^3 \quad \text{(By definition of EWML layer follows Definition 6)}$$

$$= \begin{bmatrix} \mathbf{0} \\ r_i(j-1)[1] V_{j+1}^{\top}[1, *] \cdot \bar{s}_i(j+1) \\ \vdots \\ \bar{r}'_i(j-1)[k] V_{j+1}^{\top}[K, *] \cdot \bar{s}_i(j+1) \\ \mathbf{0} \end{bmatrix} \quad \text{(By definition of } e_{j,k}^3)$$

$$= \begin{bmatrix} \mathbf{0} \\ \bar{r}'_i(j-1) \odot (V_{j+1}^{\top} \cdot \bar{s}_i(j+1)) \\ \mathbf{0} \end{bmatrix} \quad \text{(By definition of hadamard product)}$$

$$= [\mathbf{0}; \bar{s}_i(j); \mathbf{0}]. \quad \text{(By definition of } \bar{s}_i(j) \text{ follows (3.19))}$$

Therefore, by the definition of EWML layer follows Definition 6, the output $\widetilde{h}_i$ becomes

$$\widetilde{h}_i = [\text{Attn}_{\theta_{2N+3-j}}(h_i)]$$

$$= h_i + \sum_{m \in [2], k \in [K]} \sigma(\langle Q_{m,k}^{2N+3-j} h_i, K_{m,k}^{2N+3-j} h_s \rangle) V_{m,k}^{2N+3-j} h_s$$

$$= h_i + [\mathbf{0}; s(j); \mathbf{0}]$$

$$= [x_i; y_i; w; \bar{p}_i; \bar{r}'_i; g_i; \bar{s}_i(N-1); \ldots; \bar{s}_i(j+1); \mathbf{0}; 1; t_i] + [\mathbf{0}; \bar{s}_i(j); \mathbf{0}]$$

$$= [x_i; y_i; w; \bar{p}_i; \bar{r}'_i; g_i; \bar{s}_i(N-1); \ldots; \bar{s}_i(j); \mathbf{0}; 1; t_i].$$

Finally we come back to approximate the initial approximation $\bar{s}_i(N) = \bar{r}'_i(N-1) \odot g_i$. Notice that $g_i$ and $r'_i(N-1)$ are already in the input $h_i = [x_i; y_i; w; \bar{p}_i; \bar{r}'_i; g_i; \mathbf{0}; 1; t_i]$, thus it is simple to construct $\text{EWML}_{N+3}$, similar to (D.15), such that it maps,

$$[x_i; y_i; w; \bar{p}_i; \bar{r}'_i; g_i; \mathbf{0}; 1; t_i] \xrightarrow{\text{EWML}_{N+3}} [x_i; y_i; w; \mathbf{0}; 1; \bar{p}_i; \bar{r}'_i; g_i; \bar{s}_i(N); \mathbf{0}; 1; t_i].$$

Since we don't using the sum of ReLU to approximate any variables, these step don't generate extra error. Besides, by (3.18), matrices have operator norm bounds

$$\max_{j,k} \|Q_k^{N+2+j}\|_1 \leq 1, \quad \max_{j,k} \|K_k^{N+2+j}\|_1 \leq 1, \quad \max_{j,k} \|V_k^{N+2+j}\|_1 \leq 1.$$

Consequently, for any $j \in [N]$, $B_{\theta_{N+2+j}} \leq 1$. Thus we complete the proof. $\qquad \square$

## D.6 PROOF OF LEMMA 6

**Lemma 13** (Lemma 6 Restated: Error for $g_i \bar{s}_i(j)$)**.** Suppose the upper bounds $B_v, B_x > 0$ such that for any $k \in [K], j \in [N]$ and $i \in [n]$, $\|v_{j_k}\|_2 \leq B_v$, and $\|x_i\|_2 \leq B_x$. Let $r'_i(j) \in \mathbb{R}^K$ such that $r'_i(j)[k] := r'(v_{j+1_k}^{\top} p_i(j))$ follows Definition 2. Let $s_i(j) = R_i(j-1) V_{j+1}^{\top} \ldots R_i(N-2) V_N^{\top} \cdot R_i(N-1) u$ follows Definition 3. Let $\bar{r}'_i(j), g_i, \bar{s}_i(j)$ be the approximations for $r'_i(j), u(p_i(N), y_i), s_i(j)$ follows Lemma 3, Lemma 4 and Lemma 5 respectively. Let $B_{r'}$ be the upper bound of $\bar{r}'_i(j)[k]$ and $r'_i(j)[k]$ as defined in Lemma 3. Let $B_l$ be the upper bound of $g_i[k]$ and $u(p_i(N), y_i)[k]$ as defined in Lemma 4. Then for any $i \in [n+1], j \in [N], k \in [K]$,

$$\bar{s}_i(j)[k] \leq B_s,$$

$$|\bar{s}_i(j)[k] - s_i(j)[k]| \le E_s^r \epsilon_r + E_s^{r'} \epsilon_{r'} + E_s^l \epsilon_l,$$

where

$$P := \max\{\sqrt{K}, \sqrt{d}\}$$

$$B_s := \max_{j \in [N]}\{(P \cdot B_{r'} B_v)^{N-j} B_{r'} B_l\},$$

$$E_s^r := \max_{j \in [N]}\{L_{r'} E_r P B_s B_v^2 [\sum_{l=0}^{N-j-1}(B_{r'}B_vP)^l] + (B_{r'}B_vP)^{N-j}(B_l L_{r'} B_v E_r + B_{r'} L_l E_r)\},$$

$$E_s^{r'} := \max_{j \in [N]}\{P B_s B_v [\sum_{l=0}^{N-j-1}(B_{r'}B_vP)^l] + (B_{r'}B_vP)^{N-j} B_l\},$$

$$E_s^l := \max_{j \in [N]}\{(B_{r'}B_vP)^{N-j} B_{r'}\}.$$

Above, $B_s$ is the upper bound of $\bar{s}_i(j)[k]$ and $E_s^r, E_s^{r'}, E_s^l$ are the coefficients of $\epsilon_r, \epsilon_r', \epsilon_l$ in the upper bounds of $|\bar{s}_i(j)[k] - s_i(j)[k]|$, respectively.

*Proof of Lemma 6.* We use induction to prove the first two statements. To begin with, we illustrate the recursion formula for $\bar{s}_i(j)$. By (3.19), recall that for any $j \in [N]$,

$$\bar{s}_i(j) := \begin{cases} \bar{r}_i'(j-1) \odot (V_{j+1}^\top \cdot \bar{s}_i(j+1)), & j \in [N-1] \\ \bar{r}_i'(N-1) \odot g_i, & j = N. \end{cases}$$

We consider applying induction to prove the first statement:

$$\bar{s}_i(j)[k] \le (P \cdot B_{r'} B_v)^{N-n} B_{r'} B_l.$$

As for the base case, $j = N$:

$$\bar{s}_i(N)[k] = \bar{r}_i'(N-1)[k] \cdot g_i[k] \le B_{r'} B_l.$$

Therefore, if the statement holds for $j = n+1$, by (3.19) and our assumption, it holds

$$\begin{aligned} \bar{s}_i(n)[k] &= \bar{r}_i'(n-1)[k] \cdot (v_{j+1_k}^\top \bar{s}_i(n+1)) && \text{(By recursion formula (3.19))} \\ &\le \bar{r}_i'(n-1)[k] \cdot \|v_{n+1_k}\|_2 \cdot \|\bar{s}_i(n+1)\|_2 && \text{(By Cauchy-schwarz inequality)} \\ &\le \bar{r}_i'(n-1)[k] \cdot \|v_{n+1_k}\|_2 \cdot \max\{\sqrt{K}, \sqrt{d}\} \cdot \max_k |\bar{s}_i(n+1)[k]| \\ &\le (B_{r'} B_v) \cdot \max\{\sqrt{K}, \sqrt{d}\} \cdot (\max\{\sqrt{K}, \sqrt{d}\} \cdot B_{r'} B_v)^{N-n-1} B_{r'} B_l \\ & && \text{(By inductive hypothesis)} \\ &= (P \cdot B_{r'} B_v)^{N-n} B_{r'} B_l. && \text{(By definition of } P \text{ follows (3.22))} \end{aligned}$$

Thus, by the principle of induction, the first statement is true for all integers $j \in [N]$. Moreover, by the definition of $B_s$ follows (3.23), we know $B_s$ is the upper bound of $\bar{s}_i(j)[k]$. Next we apply induction to prove the second statement:

$$\begin{aligned} |\bar{s}_i(j)[k] - s_i(j)[k]| &\le (\epsilon_{r'} + L_{r'} B_v E_r \epsilon_r) P B_v B_s [\sum_{l=0}^{N-n-1}(B_{r'}B_vP)^l] \\ &\quad + (B_{r'}B_vP)^{N-n}[(B_l L_{r'} B_v E_r + B_{r'} L_l E_r)\epsilon_r + B_l \epsilon_{r'} + B_{r'} \epsilon_l]. \end{aligned}$$

For the base case, $j = N$:

$$\begin{aligned} &|\bar{s}_i(N)[k] - s_i(N)[k]| \\ &= |\bar{r}_i'(N-1)[k] \cdot g_i[k] - r_i'(N-1)[k] \cdot u(p_i(N), y_i)[k]| && \text{(By definition (3.19) and (3.7))} \end{aligned}$$

$$\leq |\bar{r}'_i(N-1)[k] - r'_i(N-1)[k]| \cdot |g_i[k]| + |r'_i(N-1)[k]| \cdot |g_i[k] - u(p_i(N), y_i)[k]|$$
$$\text{(By triangle inequality)}$$

$$\leq (\epsilon_{r'} + L_{r'}B_v E_r \epsilon_r)B_l + B_{r'}(\epsilon_l + L_l E_r \epsilon_r). \qquad \text{(By (3.13) and (3.15))}$$
$$= (B_l L_{r'} B_v E_r + B_{r'} L_l E_r)\epsilon_r + B_l \epsilon_{r'} + B_{r'}\epsilon_l$$

Therefore, if the statement holds for $j = n + 1$, by (3.19) and our assumption, it holds

$$|\bar{s}_i(n)[k] - s_i(n)[k]|$$
$$= \left|\bar{r}'_i(n-1)[k] \cdot (v^\top_{n+1_k}\bar{s}_i(n+1)) - r'_i(n-1)[k] \cdot (v^\top_{n+1_k}s_i(n+1))\right|$$
$$\text{(By the recursion formula (3.7) and (3.19))}$$

$$\leq |\bar{r}'_i(n-1)[k] - r'_i(n-1)[k]| \cdot \left|v^\top_{n+1_k}\bar{s}_i(n+1)\right|$$
$$+ |r'_i(n-1)[k]| \cdot \left|(v^\top_{n+1_k}\bar{s}_i(n+1)) - (v^\top_{n+1_k}s_i(n+1))\right| \qquad \text{(By triangle inequality)}$$

$$\leq (\epsilon_{r'} + L_{r'}B_v E_r \epsilon_r)PB_v B_s + B_{r'}B_v\|\bar{s}_i(n+1) - s_i(n+1)\|_2$$
$$\text{(By error accumulation of approximating } r' \text{ follows (3.13) )}$$

$$\leq (\epsilon_{r'} + L_{r'}B_v E_r \epsilon_r)PB_v B_s + B_{r'}B_v P \max_k |\bar{s}_i(n+1)[k] - s_i(n+1)[k]|$$

$$\leq (\epsilon_{r'} + L_{r'}B_v E_r \epsilon_r)PB_v B_s + B_{r'}B_v P\Big\{(\epsilon_{r'} + L_{r'}B_v E_r \epsilon_r)PB_v B_s [\sum_{l=0}^{N-n-2}(B_{r'}B_v P)^l]$$

$$+ (B_{r'}B_v P)^{N-n-1}[(B_l L_{r'}B_v E_r + B_{r'}L_l E_r)\epsilon_r + B_l \epsilon_{r'} + B_{r'}\epsilon_l]\Big\} \quad \text{(By inductive hypothesis)}$$

$$\leq (\epsilon_{r'} + L_{r'}B_v E_r \epsilon_r)PB_v B_s [\sum_{l=0}^{N-n-1}(B_{r'}B_v P)^l]$$

$$+ (B_{r'}B_v P)^{N-n}[(B_l L_{r'}B_v E_r + B_{r'}L_l E_r)\epsilon_r + B_l \epsilon_{r'} + B_{r'}\epsilon_l].$$

Thus, by the principle of induction, the second statement is true for all integers $j \in [N-1]$. By the definition of $E_s$ follows (3.24), it is simple to check that

$$|\bar{s}_i(j)[k] - s_i(j)[k]| \leq E_s^r \epsilon_r + E_s^{r'} \epsilon_{r'} + E_s^l \epsilon_l.$$

Thus we complete the proof. $\qquad\square$

### D.7 PROOF OF THEOREM 1

**Theorem 4** (Theorem 1 Restated: In-context gradient descent on $N$-layer NNs). Fix any $B_v, \eta, \epsilon > 0, L \geq 1$. For any input sequences takes from (2.1), their exist upper bounds $B_x, B_y$ such that for any $i \in [n]$, $\|y_i\|_2 \leq B_y$, $\|x_i\|_2 \leq B_x$. Assume functions $r(t)$, $r'(t)$ and $u(t,y)[k]$ are $L_r, L_{r'}, L_l$-Lipschitz continuous. Suppose $\mathcal{W}$ is a closed domain such that for any $j \in [N-1]$ and $k \in [K]$,

$$\mathcal{W} \subset \left\{w = [v_{j_k}] \in \mathbb{R}^{D_N} : \|v_{j_k}\|_2 \leq B_v\right\},$$

and $\text{Proj}_{\mathcal{W}}$ project $w$ into bounded domain $\mathcal{W}$. Assume $\text{Proj}_{\mathcal{W}} = \text{MLP}_\theta$ for some MLP layer with hidden dimension $D_w$ parameters $\|\theta\| \leq C_w$. If functions $r(t)$, $r'(t)$ and $u(t,y)[k]$ are $C^4$-smoothness, then for any $\epsilon > 0$, there exists a transformer model $\text{NN}_\theta$ with $(2N+4)L$ hidden layers consists of $L$ neural network blocks $\text{TF}_\theta^{N+2} \circ \text{EWML}_\theta^N \circ \text{TF}_\theta^2$,

$$\text{NN}_\theta := \text{TF}_\theta^{N+2} \circ \text{EWML}_\theta^N \circ \text{TF}_\theta^2 \circ \ldots \circ \text{TF}_\theta^{N+2} \circ \text{EWML}_\theta^N \circ \text{TF}_\theta^2,$$

such that the heads number $M^l$, embedding dimensions $D^l$, and the parameter norms $B_{\theta^l}$ suffice

$$\max_{l \in [(2N+4)L]} M^l \leq \widetilde{O}(\epsilon^{-2}), \quad \max_{l \in [(2N+4)L]} D^l \leq O(NK^2) + D_w, \quad \max_{l \in [(2N+4)L]} B_{\theta^l} \leq O(\eta) + C_w + 1,$$

where $\widetilde{O}(\cdot)$ hides the constants that depend on $d, K, N$, the radius parameters $B_x, B_y, B_v$ and the smoothness of $r$ and $\ell$. And this neural network such that for any input sequences $H^{(0)}$, take from (2.1), $\mathrm{NN}_\theta(H^{(0)})$ implements $L$ steps in-context gradient descent on risk Eqn (2.2): For every $l \in [L]$, the $(2N+4)l$-th layer outputs $h_i^{((2N+4)l)} = [x_i; y_i; \overline{w}^{(l)}; \mathbf{0}; 1; t_i]$ for every $i \in [n+1]$, and approximation gradients $\overline{w}^{(l)}$ such that

$$\overline{w}^{(l)} = \mathrm{Proj}_{\mathcal{W}}(\overline{w}^{(l-1)} - \eta\nabla\mathcal{L}_n(\overline{w}^{(l-1)}) + \epsilon^{(l-1)}), \quad \overline{w}^{(0)} = \mathbf{0},$$

where $\|\epsilon^{(l-1)}\|_2 \leq \eta\epsilon$ is an error term.

*Proof of Theorem 1.* We consider the first $N+2$ transformer layers $\mathrm{TF}_\theta^{N+2}$ are layers in Lemma 2 ,Lemma 3 and Lemma 4. Then we let the middle $N$ element-wise multiplication layers $\mathrm{EWML}_\theta^N$ be layers in Lemma 5. We only need to check approximability conditions. By Lemma 7 and our assumptions, for any $\epsilon_r, \epsilon_{r'}, \epsilon_l$, it holds

- Function $r(t)$ is $(\epsilon_r, R_1, M_1, C_1)$-approximable for $R_1 = \max\{B_v B_r, 1\}$, $M_1 \leq \widetilde{\mathcal{O}}(C_1^2 \epsilon_r^{-2})$, where $C_1$ depends only on $R_1$ and the $C^2$-smoothness of $r(t)$.

- Function $r'(t)$ is $(\epsilon_{r'}, R_2, M_2, C_2)$-approximable for $R_2 = \max\{B_v B_{r'}, 1\}$, $M_2 \leq \widetilde{\mathcal{O}}(C_2^2 \epsilon_r'^{-2})$, where $C_2$ depends only on $R_2$ and the $C^2$-smoothness of $r'(t)$.

- Function $\partial_1 \ell(t, y)$ is $(\epsilon_l, R_3, M_3, C_3)$-approximable for $R_3 = \max\{B_v B_r, 1\}$, $M_3 \leq \widetilde{\mathcal{O}}(C_3^2 \epsilon_l^{-2})$, where $C_3$ depends only on $R_3$ and the $C^3$-smoothness of $u(t, y)[k]$.

which suffice approximability conditions in Lemma 2, Lemma 3 and Lemma 4.

Now we construct the last two layers to implement $w - \eta\nabla\mathcal{L}_n(w)$ and $\mathrm{Proj}_{\mathcal{W}}(w)$. First we construct a attention layer to approximate $w - \eta\nabla\mathcal{L}_n(w)$. For every $m \in [2], j \in [N], k \in [K]$, we consider matrices $Q_{m,j,k}^{2N+3}, j_{m,j,k}^{2N+3}, V_{m,j,k}^{2N+3} \in \mathbb{R}^{D \times D}$ such that

$$Q_{1,j,k}^{2N+3}h_i = \begin{bmatrix} 1 \\ \mathbf{0} \end{bmatrix}, \quad K_{1,j,k}^{2N+3}h_i = \begin{bmatrix} \bar{s}_i(j)[k] \\ \mathbf{0} \end{bmatrix}, \quad V_{1,j,k}^{2N+3}h_i = -\frac{\eta(n+1)}{2n}\begin{bmatrix} \mathbf{0} \\ \bar{p}_i(j-1) \\ \mathbf{0} \end{bmatrix},$$

$$Q_{2,j,k}^{2N+3}h_i = \begin{bmatrix} -1 \\ \mathbf{0} \end{bmatrix}, \quad K_{2,j,k}^{2N+3}h_i = \begin{bmatrix} \bar{s}_i(j)[k] \\ \mathbf{0} \end{bmatrix}, \quad V_{2,j,k}^{2N+3}h_i = -\frac{\eta(n+1)}{2n}\begin{bmatrix} \mathbf{0} \\ -\bar{p}_i(j-1) \\ \mathbf{0} \end{bmatrix}.$$

$$\text{(D.16)}$$

Furthermore, we define approximation gradient $\nabla_w\overline{\mathcal{L}}_n(w)$ as follows,

$$\nabla_w\overline{\mathcal{L}}_n(w) := -\frac{1}{\eta(n+1)}\sum_{t=1}^{n+1}\sum_{m\in[2],j\in[N],k\in[K]}\sigma(\langle Q_{m,j,k}^{2N+3}h_i, K_{m,j,k}^{2N+3}h_t\rangle)V_{m,j,k}^{2N+3}h_t$$

$$= \frac{1}{2n}\sum_{t=1}^{n+1}\sum_{k=1}^{K}\sum_{j=1}^{N}(\sigma(\bar{s}_t(j)[k]) - \sigma(-\bar{s}_t(j)[k]))\begin{bmatrix} \mathbf{0} \\ \bar{p}_t(j-1) \\ \mathbf{0} \end{bmatrix} \qquad \text{(By our construction (D.16))}$$

$$= \frac{1}{2n}\sum_{t=1}^{n+1}\sum_{k=1}^{K}\sum_{j=1}^{N}\bar{s}_t(j)[k] \cdot \begin{bmatrix} \mathbf{0} \\ \bar{p}_t(j-1) \\ \mathbf{0} \end{bmatrix} \qquad \text{(By $f(x) = \sigma(x) - \sigma(-x)$)}$$

$$= \frac{1}{2n}\sum_{t=1}^{n+1}\sum_{j=1}^{N}\begin{bmatrix} \mathbf{0} \\ \mathbf{I}_{K\times K} \otimes \bar{p}_t(j-1) \cdot \bar{s}_t(j) \\ \mathbf{0} \end{bmatrix} \qquad \text{(By definition of Kronecker product)}$$

$$= \frac{1}{2n} \sum_{t=1}^{n} \begin{bmatrix} \mathbf{0} \\ \bar{A}_t(1) \\ \vdots \\ \bar{A}_t(N) \\ \mathbf{0} \end{bmatrix}, \qquad \text{(By } \bar{s}_{n+1}(j) = \mathbf{0} \text{ follows Lemma 5)}$$

where $\bar{A}_t(j) := \mathbf{I}_{K \times K} \otimes \bar{p}_t(j-1) \cdot \bar{s}_t(j)$ denotes the approximation for $A_t(j)$. Therefore, by the definition of ReLU attention layer follows Definition 7, for any $i \in [n+1]$,

$$\widetilde{h}_i = [\text{Attn}_{\theta_{2N+3}}(h_i)]$$

$$= h_i + \frac{1}{n+1} \sum_{i=1}^{n+1} \sum_{m \in [2], j \in [N], k \in [K]} \sigma(\langle Q_{m,j,k}^{2N+3} h_s, K_{m,j,k}^{2N+3} h_i \rangle) V_{m,j,k}^{2N+3} h_i$$

$$= [x_i; y_i; w; \bar{p}_i; \bar{r}'_i; g_i; \bar{s}_i; \mathbf{0}; 1; t_i] - \frac{\eta}{2n} \sum_{t=1}^{n} \begin{bmatrix} \mathbf{0} \\ \bar{A}_t(1) \\ \vdots \\ \bar{A}_t(N) \\ \mathbf{0} \end{bmatrix}$$

$$= [x_i; y_i; w - \eta \nabla_w \bar{\mathcal{L}}_n(w); \bar{p}_i; \bar{r}'_i; g_i; \mathbf{0}; 1; t_i]. \qquad \text{(By definition of } \nabla_w \bar{\mathcal{L}}_n(w))$$

Since we do not use approximation technique like Definition 4, this step do not generate extra error. Besides,by (C.1), matrices have operator norm bounds

$$\max_{j,m,k} \|Q_{j,m,k}^{2N+3}\|_1 \leq 1, \quad \max_{j,m,k} \|K_{j,m,k}^{2N+3}\|_1 \leq 1, \quad \sum_{j,m,k} \|V_{j,m,k}^{2N+3}\|_1 \leq 2\eta NK.$$

Consequently, $B_{\theta_{2N+3}} \leq 1 + 2\eta NK$. Fix any $\epsilon > 0$, then we pick appropriate $\epsilon_r, \epsilon'_r, \epsilon_l$ such that

$$\|\epsilon^{(l-1)}\|_2 = \eta \|\nabla_w \bar{\mathcal{L}}_n(\overline{w}^{(l-1)}) - \nabla_w \mathcal{L}_n(\overline{w}^{(l-1)})\|_2 \leq \eta \epsilon.$$

By Definition 3 and Lemma 6, for any $j \in [N-1], i \in [n]$, it holds

$$\|\bar{A}_i(j) - A_i(j)\|_2$$

$$\leq \sum_{k=1}^{K} \|\bar{s}_i(j)[k] \bar{p}_i(j-1) - s_i(j)[k] p_i(j-1)\|_2 \qquad \text{(By Definition 2 and definition of } \bar{A}_i(j))$$

$$\leq \sum_{k=1}^{K} |\bar{s}_i(j)[k] - s_i(j)[k]| \cdot \|\bar{p}_i(j-1)\|_2 + |s_i(j)[k]| \cdot \|\bar{p}_i(j-1) - p_i(j-1)\|_2$$

$$\qquad \text{(By triangle inequality)}$$

$$\leq P[(E_s^r \epsilon_r + E_s^{r'} \epsilon_{r'} + E_s^l \epsilon_l) \sqrt{P} B_r + B_s E_r \epsilon_r], \qquad \text{(By (3.11) and Lemma 6 )}$$

where $B_s$ is the upper bound of $\bar{s}_i(j)[k]$ and $E_s^r, E_s^{r'}, E_s^l$ are the coefficients of $\epsilon_r, \epsilon'_r, \epsilon_l$ in the upper bounds of $|\bar{s}_i(j)[k] - s_i(j)[k]|$ follow Lemma 6, respectively. We can drive similar results as $j = N$. Actually, by $P = \max\{\sqrt{K}, \sqrt{d}\}$ follows Lemma 6, above inequality also holds for $j = N$. Therefore, the error in total such that for any $w$,

$$\|\nabla_w \bar{\mathcal{L}}_n(w) - \nabla_w \mathcal{L}_n(w)\|_2$$

$$= \|\frac{1}{2n} \sum_{t=1}^{n} \begin{bmatrix} \mathbf{0} \\ \bar{A}_t(1) \\ \vdots \\ \bar{A}_t(N) \\ \mathbf{0} \end{bmatrix} - \frac{1}{2n} \sum_{t=1}^{n} \begin{bmatrix} \mathbf{0} \\ A_t(1) \\ \vdots \\ A_t(N) \\ \mathbf{0} \end{bmatrix} \|_2 \qquad \text{(By definition of } \bar{\mathcal{L}}_n(w) \text{ and } \mathcal{L}_n(w))$$

$$\leq \frac{1}{2} \max_{1 \leq t \leq n} \{ \sum_{j=1}^{N} \|\bar{A}_t(j) - A_t(j)\|_2 \}$$

$$\leq \frac{N}{2} P[(E_s^r \epsilon_r + E_s^{r'} \epsilon_{r'} + E_s^l \epsilon_l)\sqrt{P} B_r + B_s E_r \epsilon_r].$$

(By the error accumulation results derived before)

Let $C_l, C_r, C_{r'}$ denotes coefficients in front of $\epsilon_l, \epsilon_r, \epsilon_{r'}$ respectively. Then it holds

$$C_l = N P^{\frac{3}{2}} B_r E_s^l,$$
$$C_r = N P^{\frac{3}{2}} B_r E_s^r + N P B_s E_r,$$
$$C_{r'} = N P^{\frac{3}{2}} B_r E_s^{r'}.$$

Thus, to ensure $\|\nabla_w \bar{\mathcal{L}}_n(w) - \nabla_w \mathcal{L}_n(w)\|_2 \leq \epsilon$, we only need to select $\epsilon_l, \epsilon_r, \epsilon_r'$ as

$$\epsilon_l = \frac{2\epsilon}{3C_l}, \quad \epsilon_r = \frac{2\epsilon}{3C_r}, \quad \epsilon_r' = \frac{2\epsilon}{3C_{r'}}.$$

Therefore, we only need to pick the last MLP layer $\text{MLP}_{2N+4}$ such that it maps

$$[x_i; y_i; w - \eta\nabla_w \mathcal{L}_n(w); \bar{p}_i; \bar{r}_i'; g_i; \bar{s}_i; \mathbf{0}; 1; t_i] \xrightarrow{\text{MLP}_{2N+4}} [x_i; y_i; \text{Proj}_{\mathcal{W}}(w - \eta\nabla_w \mathcal{L}_n(w)); \mathbf{0}; 1; t_i].$$

By our assumption on the map $\text{Proj}_{\mathcal{W}}$, this is easy.

Finally, we analyze how many embedding dimensions of Transformers are needed to implement the above ICGD. Recall that

$$x_i, y_i \in \mathbb{R}^d, w \in \mathbb{R}^{2dK+(N-2)K^2}, \bar{p}_i \in \mathbb{R}^{(N-1)K+d}, \bar{r}_i' \in \mathbb{R}^{(N-2)K+d}, g_i \in \mathbb{R}^d, \bar{s}_i \in \mathbb{R}^{(N-1)K+d}.$$

Therefore, $\max\{\Omega(NK^2), D_w\}$ embedding dimensions of Transformer are required to implement ICGD on deep models.

Combining the above, we complete the proof. □

## D.8 PROOF OF COROLLARY 1.1

**Corollary 4.1** (Corollary 1.1 Restated: Error for implementing ICGD on $N$-layer neural network). Fix $L \geq 1$, under the same setting as Theorem 1, $(2N+4)L$-layer neural networks $\text{NN}_\theta$ approximates the true gradient descent trajectory $\{w_{\text{GD}}^l\}_{l \geq 0} \in \mathbb{R}^{D_N}$ with the error accumulation

$$\|\bar{w}^l - w_{\text{GD}}^l\|_2 \leq L_f^{-1}(1 + nL_f)^l \epsilon,$$

where $L_f$ denotes the Lipschitz constant of $\mathcal{L}_N(w)$ within $\mathcal{W}$.

First we introduce a helper lemma.

**Lemma 14** (Error for Approximating GD, Lemma G.1 of (Bai et al., 2023)). Let $\mathcal{W} \subset \mathbb{R}^d$ is a convex bounded domain and $\text{Proj}_{\mathcal{W}}$ projects all vectors into $\mathcal{W}$. Suppose $f : \mathcal{W} \to R$ and $\nabla f$ is $L_f$-Lipschitz on $\mathcal{W}$. Fix any $\epsilon > 0$, let sequences $\{\bar{w}^l\}_{l \geq 0} \in \mathbb{R}^d$ and $\{w_{\text{GD}}^l\}_{l \geq 0} \in \mathbb{R}^d$ are given by $\bar{w}^0 = w_{\text{GD}}^0 = \mathbf{0}$, then for all $l \geq 0$,

$$\bar{w}^l = \text{Proj}_{\mathcal{W}}(\bar{w}^{l-1} - \eta\nabla\mathcal{L}_n(\bar{w}^{l-1}) + \epsilon^{l-1}), \quad \|\epsilon^{l-1}\|_2 \leq \eta\epsilon,$$
$$w_{\text{GD}}^l = \text{Proj}_{\mathcal{W}}(w_{\text{GD}}^{l-1} - \eta\nabla\mathcal{L}_n(w_{\text{GD}}^{l-1}))$$

To show the convergence, we define the gradient mapping at $w$ with step size $\eta$ as,

$$G^f_{\mathcal{W},\eta} := \frac{w - \text{Proj}_{\mathcal{W}}(w - \eta \nabla \mathcal{L}_n(w))}{\eta}.$$

Then if $\eta \leq L_f$, for all $L \geq 1$, convergence holds

$$\min_{l \in [L-1]} \|G^f_{\mathcal{W},\eta}(\overline{w}^l)\|^2_2 \leq \frac{1}{L} \sum_{l=1}^{L-1} \|G^f_{\mathcal{W},\eta}(\overline{w}^l)\|^2_2 \leq \frac{8(f(\mathbf{0}) - \inf_{w \in \mathcal{W}} f(w))}{\eta L} + 10\epsilon^2.$$

Moreover, for any $l \geq 0$, the error accumulation is

$$\|\overline{w}^l - w^l_{\text{GD}}\|_2 \leq L_f^{-1}(1 + nL_f)^l \epsilon.$$

Lemma 14 shows Theorem 1 leads to exponential error accumulation in the general case. Moreover, Lemma 14 also provides convergence of approximating GD. Then we proof Corollary 1.1.

*Proof.* For any small $\epsilon$, by Theorem 1, the neural network $\text{NN}_\theta$ implements each gradient descent step with error bounded by $\epsilon$. Then we simply apply Lemma 14 to complete the proof. □

## E    EXTENSION: DIFFERENT INPUT AND OUTPUT DIMENSIONS

In this section, we explore the ICGD on $N$-layer neural networks under the setting where the dimensions of input $x_i$ and label $y_i$ can be different. Specifically, we consider our prompt datasets $\{(x_i, y_i)\}_{i \in [n]}$ where $x_i \in \mathbb{R}^{d_x}$ and $y_i \in \mathbb{R}^{d_y}$. We start with our new $N$-layer neural network.

**Definition 10** ($N$-Layer Neural Network). An $N$-Layer Neural Network comprises $N - 1$ hidden layers and 1 output layer, all constructed similarly. Let $r : \mathbb{R} \to \mathbb{R}$ be the activation function. For the hidden layers: for any $i \in [n + 1], j \in [N - 1]$, and $k \in [K]$, the output for the first $j$ layers w.r.t. input $x_i \in \mathbb{R}^d$, denoted by $\mathrm{pred}_h(x_i; j) \in \mathbb{R}^K$, is defined as recursive form:

$$\mathrm{pred}_h(x_i; 1)[k] := r(v_{1_k}^\top x_i), \quad \text{and} \quad \mathrm{pred}_h(x_i; j)[k] := r(v_{j_k}^\top \mathrm{pred}_h(x_i; j - 1)),$$

where $v_{1_k} \in \mathbb{R}^d$ and $v_{j_k} \in \mathbb{R}^K$ for $j \in \{2, \ldots, N - 1\}$ are the $k$-th parameter vectors in the first layer and the $j$-th layer, respectively. For the output layer ($N$-th layer), the output for the first $N$ layers (i.e the entire neural network) w.r.t. input $x_i \in \mathbb{R}^{d_x}$, denoted by $\mathrm{pred}_o(x_i; w, N) \in \mathbb{R}^{d_y}$, is defined for any $k \in [d_y]$ as follows:

$$\mathrm{pred}_o(x_i; w, N)[k] := r(v_{N_k}^\top \mathrm{pred}_h(x_i; N - 1)),$$

where $v_{N_k} \in \mathbb{R}^K$ are the $k$-th parameter vectors in the $N$-th layer and $w \in \mathbb{R}^{(d_x + d_y)K + (N-2)K^2}$ denotes the vector containing all parameters in the neural network,

$$w := \left[ v_{1_1}^\top, \ldots, v_{1_K}^\top, \ldots, v_{j_k}^\top, \ldots v_{N-1_1}^\top, \ldots, v_{N-1_K}^\top, v_{N_1}^\top, \ldots, v_{N_{d_y}}^\top \right]^\top.$$

Notice that our new $N$-layer neural network only modify the output layer compared to Definition 1. Intuitively, this results in minimal change in output, which allows our framework in Section 3.3 to function across varying input/output dimensions. Theoretically, we derive the explicit form of gradient $\nabla \mathcal{L}_n(w)$.

**Lemma 15** (Decomposition of One Gradient Descent Step). Fix any $B_v, \eta > 0$. Suppose the empirical loss function $\mathcal{L}_n(w)$ on $n$ data points $\{(x_i, y_i)\}_{i \in [n]}$ is defined as

$$\mathcal{L}_n(w) := \frac{1}{2n} \sum_{i=1}^n \ell(f(w, x_i), y_i), \quad \text{where } \ell : \mathbb{R}^{d_y} \times \mathbb{R}^{d_y} \to \mathbb{R} \text{ is a loss function,}$$

where $f(w, x_i), y_i)$ is the output of $N$-layer neural networks (Definition 10) with modified output layer. Suppose closed domain $\mathcal{W}$ and projection function $\mathrm{Proj}_{\mathcal{W}}(w)$ follows (3.4). Let $A_i(j), r_i'(j), R_i(j), V_j$ be as defined in Definition 2 (with modified dimensions), then the explicit form of gradient $\nabla \mathcal{L}_n(w)$ becomes

$$\nabla \mathcal{L}_n(w) = \frac{1}{2n} \sum_{i=1}^n \begin{bmatrix} A_i(1) \\ \vdots \\ A_i(N) \end{bmatrix},$$

where $A_i(j)$ denote the derivative of $\ell(p_i(N), y_i)$ with respect to the parameters in the $j$-th layer,

$$A_i(j) = \begin{cases} (R_i(N-1) \cdot V_N \cdot \ldots \cdot R_i(j-1) \cdot \left[ \mathbf{I}_{K \times K} \otimes p_i(j-1)^\top \right])^\top \cdot (\frac{\partial \ell(p_i(N), y_i)}{\partial p_i(N)})^\top, & j \neq N \\ (R_i(N-1) \cdot \left[ \mathbf{I}_{d_y \times d_y} \otimes p_i(N-1)^\top \right])^\top \cdot (\frac{\partial \ell(p_i(N), y_i)}{\partial p_i(N)})^\top, & j = N. \end{cases}$$

*Proof.* Simply follow the proof of Lemma 1. We show the different terms compared to Definition 2:

- Let $D_j \in \mathbb{R}$ denote the total number of parameters in the first $j$ layers.

$$D_j = \begin{cases} 0, & j = 0 \\ d_x K, & j = 1 \\ (j-1)K^2 + d_x K, & 2 \leq j \leq N-1 \\ (N-2)K^2 + (d_x + d_y)K, & j = N, \end{cases}$$

- The intermediate term $R_i(N-1)$,

$$R_i(N-1) = \text{diag}\{r'(v_{j+1_1}^\top p_i(j)), \dots, r'(v_{j+1_{d_y}}^\top p_i(j))\} \in \mathbb{R}^{d_y \times d_y}.$$

- The parameters matrices of the first and the last layers:

$$V_j := \begin{cases} \left[v_{1_1}, \dots, v_{1_K}\right]^\top \in \mathbb{R}^{K \times d_x}, & j = 1 \\ \left[v_{N_1}, \dots, v_{N_{d_y}}\right]^\top \in \mathbb{R}^{d_y \times K}, & j = N. \end{cases}$$

Thus we complete the proof. $\square$

Lemma 15 shows that the explicit form of gradient $\nabla \mathcal{L}_n(w)$ holds the same structure as Lemma 1. Therefore, it is simple to follow our framework in Section 3.3 to approximate $\nabla \mathcal{L}_n(w)$ term by term. Finally, we introduce the generalized version of main result Theorem 1.

**Theorem 5** (In-Context Gradient Descent on $N$-layer NNs). Fix any $B_v, \eta, \epsilon > 0, L \geq 1$. For any input sequences takes from (2.1), where $\{(x_i, y_i)\}_{i \in [n]}$ and $x_i \in \mathbb{R}^{d_x}$ and $y_i \in \mathbb{R}^{d_y}$, their exist upper bounds $B_x, B_y$ such that for any $i \in [n]$, $\|y_i\|_2 \leq B_y, \|x_i\|_2 \leq B_x$. Assume functions $r(t), r'(t)$ and $u(t,y)[k]$ are $L_r, L_{r'}, L_l$-Lipschitz continuous. Suppose $\mathcal{W}$ is a closed domain such that for any $j \in [N-1]$ and $k \in [K]$,

$$\mathcal{W} \subset \left\{ w = [v_{j_k}] \in \mathbb{R}^{D_N} : \|v_{j_k}\|_2 \leq B_v \right\},$$

and $\text{Proj}_{\mathcal{W}}$ project $w$ into bounded domain $\mathcal{W}$. Assume $\text{Proj}_{\mathcal{W}} = \text{MLP}_\theta$ for some MLP layer with hidden dimension $D_w$ parameters $\|\theta\| \leq C_w$. If functions $r(t), r'(t)$ and $u(t,y)[k]$ are $C^4$-smoothness, then for any $\epsilon > 0$, there exists a transformer model $\text{NN}_\theta$ with $(2N+4)L$ hidden layers consists of $L$ neural network blocks $\text{TF}_\theta^{N+2} \circ \text{EWML}_\theta^N \circ \text{TF}_\theta^2$,

$$\text{NN}_\theta := \text{TF}_\theta^{N+2} \circ \text{EWML}_\theta^N \circ \text{TF}_\theta^2 \circ \dots \circ \text{TF}_\theta^{N+2} \circ \text{EWML}_\theta^N \circ \text{TF}_\theta^2,$$

such that the heads number $M^l$, parameter dimensions $D^l$, and the parameter norms $B_{\theta^l}$ suffice

$$\max_{l \in [(2N+4)L]} M^l \leq \widetilde{O}(\epsilon^{-2}), \quad \max_{l \in [(2N+4)L]} D^l \leq O(K^2 N) + D_w, \quad \max_{l \in [(2N+4)L]} B_{\theta^l} \leq O(\eta) + C_w + 1,$$

where $\widetilde{O}(\cdot)$ hides the constants that depend on $d, K, N$, the radius parameters $B_x, B_y, B_v$ and the smoothness of $r$ and $\ell$. And this neural network such that for any input sequences $H^{(0)}$, take from (2.1), $\text{NN}_\theta(H^{(0)})$ implements $L$ steps in-context gradient descent on risk $\mathcal{L}_n(w)$ follows Lemma 15: For every $l \in [L]$, the $(2N+4)l$-th layer outputs $h_i^{((2N+4)l)} = [x_i; y_i; \overline{w}^{(l)}; \mathbf{0}; 1; t_i]$ for every $i \in [n+1]$, and approximation gradients $\overline{w}^{(l)}$ such that

$$\overline{w}^{(l)} = \text{Proj}_{\mathcal{W}}(\overline{w}^{(l-1)} - \eta \nabla \mathcal{L}_n(\overline{w}^{(l-1)}) + \epsilon^{(l-1)}), \quad \overline{w}^{(0)} = \mathbf{0},$$

where $\|\epsilon^{(l-1)}\|_2 \leq \eta \epsilon$ is an error term.

# F  EXTENSION: SOFTMAX TRANSFORMER

In this part, we demonstrate the existence of pretrained $\mathrm{Softmax}$ transformers capable of implementing ICGD on an $N$-layer neural network. First, we introduce our main technique: the universal approximation property of softmax transformers in Appendix F.1. Then, we prove the existence of pretrained softmax transformers that implement ICGD on $N$-layer neural networks in Appendix F.2.

## F.1  UNIVERSAL APPROXIMATION OF SOFTMAX TRANSFORMER

**Softmax-Attention Layer.** We replace modified normalized ReLU activation $\sigma/n$ in ReLU attention layer (Definition 7) by standard softmax. Thus, for any input sequence $H \in \mathbb{R}^{D \times n}$, a single head attention layer outputs

$$\mathrm{Attn}\,(H) = H + W^{(O)}(VH)\,\mathrm{Softmax}\left[(KH)^{\top}(QH)\right], \tag{F.1}$$

where $W^{(O)}, Q, K, V \in \mathbb{R}^{D \times D} \in \mathbb{R}^{d \times d}$ are the weight matrices. Then we introduce the softmax transformer block, which consists of two feed-forward neural network layers and a single-head self-attention layer with the softmax function.

**Definition 11** (Transformer Block $\mathcal{T}_{\mathrm{Softmax}}$). For any input sequences $H \in \mathbb{R}^{D \times n}$, let $\mathrm{FF}(H) \coloneqq H + W_2 \cdot \mathrm{ReLU}(W_1 H + b_1 \mathbb{1}_L^{\top}) + b_2 \mathbb{1}_L^{\top}$ be the Feed-Forward layer, where $d'$ is hidden dimensions, $W_1 \in \mathbb{R}^{d' \times D}$, $W_2 \in \mathbb{R}^{D \times d'}$, $b_1 \in \mathbb{R}^l$, and $b_2 \in \mathbb{R}^d$. We configure a transformer block with Softmax-attention layer as $\mathcal{T}_{\mathrm{Softmax}} \coloneqq \{\mathrm{FF} \circ \mathrm{Attn} \circ \mathrm{FF} : \mathbb{R}^{d \times L} \to \mathbb{R}^{d \times L}\}$.

**Universal Approximation of Softmax-Transformer.** We show the universal approximation theorem for Transformer blocks (Definition 11). Specifically, Transformer blocks $\mathcal{T}_{\mathrm{Softmax}}$ are universal approximators for continuous permutation equivariant functions on bounded domain.

**Lemma 16** (Universal Approximation of $\mathcal{T}_{\mathrm{Softmax}}$). Let $f(\cdot) \coloneqq \mathbb{R}^{d \times n} \to \mathbb{R}^{d \times n}$ be any $L$-Lipschitz permutation equivariant function supported on $[0, B_x]^{d \times n}$. We denote the discrete input domain of $[0, B_x]^{d \times n}$ by a grid $\mathbb{G}_D$ with granularity $D \in \mathbb{N}$ defined as $\mathbb{G}_D = \{B_x/D, 2B_x/D, \ldots, B_x\}^{d \times n} \subset \mathbb{R}^{d \times n}$. For any $\kappa > 0$, there exists a transformer network $f_{\mathrm{Softmax}} \in \mathcal{T}_{\mathrm{Softmax}}$, such that for any $Z \in [0, B_x]^{d \times n}$, it approximate $f(Z)$ as: $\|f_{\mathrm{Softmax}}(Z) - f(Z)\|_2 \leq \frac{L(8\sqrt{dn}+\sqrt{n})B_x}{2D} = \kappa$.

*Proof Sketch.* First, we use a piece-wise constant function to approximate $f$ and derive an upper bound based on its $L$-Lipschitz property. Next, we demonstrate how the feed-forward neural network $\mathcal{F}_1^{(FF)}$ quantizes the continuous input domain into the discrete domain $\mathbb{G}_D$ through a multiple-step function, using ReLU functions to create a piece-wise linear approximation. Then, we apply the self-attention layer $\mathcal{F}^{(SA)}$ on $\mathcal{F}_1^{(FF)}$, establishing a bounded output region for $\mathcal{F}_S^{(SA)} \circ \mathcal{F}_1^{(FF)}$. Finally, we employ a second feed-forward network $\mathcal{F}_2^{(FF)}$ to predict $f_{\mathrm{Softmax}}(Z)$ and assess the approximation error relative to the actual output $f(Z)$. See Appendix F.4 for a detailed proof. $\square$

## F.2  IN-CONTEXT GRADIENT DESCENT WITH SOFTMAX TRANSFORMER

**In-Context Gradient Descent with Softmax Transformer.** By applying universal approximation theory (Lemma 16), we now illustrate how to use Transformer block $\mathcal{T}_{\mathrm{Softmax}}$ (Definition 11) and MLP layers (Definition 8) to implement ICGD on general risk function $\mathcal{L}_n(w)$.

**Theorem 6** (In-Context Gradient Descent on General Risk Function). Fix any $B_w, \eta, \epsilon > 0, L \geq 1$. For any input sequences takes from (2.1), their exist upper bounds $B_x, B_y$ such that for any $i \in [n]$, $\|y_i\|_{\max} \leq B_y$, $\|x_i\|_{\max} \leq B_x$. Suppose $\mathcal{W}$ is a closed domain such that $\|w\|_{\max} \leq B_w$ and $\mathrm{Proj}_{\mathcal{W}}$ project $w$ into bounded domain $\mathcal{W}$. Assume $\mathrm{Proj}_{\mathcal{W}} = \mathrm{MLP}_\theta$ for some MLP layer. Define $l(w, x_i, y_i)$ as a loss function with $L$-Lipschitz gradient. Let $\mathcal{L}_n(w) = \frac{1}{n}\sum_{i=1}^{n} \ell(w, x_i, y_i)$ denote the empirical loss function, then there exists a transformer $\mathrm{NN}_\theta$, such that for any input sequences $H^{(0)}$, take from (2.1), $\mathrm{NN}_\theta(H^{(0)})$ implements $L$ steps in-context gradient descent on $\mathcal{L}_n(w)$: For every $l \in [L]$, the $4l$-th layer outputs $h_i^{(4l)} = [x_i; y_i; \overline{w}^{(l)}; \mathbf{0}; 1; t_i]$ for every $i \in [n+1]$, and approximation gradients

$\overline{w}^{(l)}$ such that

$$\overline{w}^{(l)} = \text{Proj}_{\mathcal{W}}(\overline{w}^{(l-1)} - \eta\nabla\mathcal{L}_n(\overline{w}^{(l-1)}) + \epsilon^{(l-1)}), \quad \overline{w}^{(0)} = \mathbf{0},$$

where $\|\epsilon^{(l-1)}\|_2 \leq \eta\epsilon$ is an error term.

*Proof Sketch.* By our assumption $\text{Proj}_{\mathcal{W}} = \text{MLP}_\theta$, we only need to find a transformer to implement gradient descent $w^+ := w - \eta\nabla\mathcal{L}_n(w)$. For ant input takes from (F.2), let function $f : \mathbb{R}^{D\times n} \to \mathbb{R}^{D\times n}$ maps $w$ into $w - \eta\nabla\mathcal{L}_n(w)$ and preserve other elements. By Lemma 7, their exist a transformer block $f_{\text{Softmax}}$ capable of approximating $f$ with any desired small error. Therefore, $f_{\text{Softmax}} \circ \text{MLP}$ suffices our requirements. Please see Appendix F.3 for a detailed proof. $\qquad\square$

### F.3   PROOF OF THEOREM 6

*Proof of Theorem 6.* We only need to construct a 4 layers transformer capable of implementing single step gradient descent. With out loss of generality, we assume $w \in \mathbb{R}^{D_w}$. Recall that the input sequences $H \in \mathbb{R}^{D\times n}$ takes form

$$H := \begin{bmatrix} x_1 & x_2 & \cdots & x_n & x_{n+1} \\ y_1 & y_2 & \cdots & y_n & 0 \\ q_1 & q_2 & \cdots & q_n & q_{n+1} \end{bmatrix} \in \mathbb{R}^{D\times(n+1)}, \quad q_i := \begin{bmatrix} w \\ 0 \\ 1 \\ t_i \end{bmatrix} \in \mathbb{R}^{D-(d+1)}. \qquad \text{(F.2)}$$

Let function $f : \mathbb{R}^{D\times n} \to \mathbb{R}^{D\times n}$ output

$$f(H) = \begin{bmatrix} x_1 & x_2 & \cdots & x_n & x_{n+1} \\ y_1 & y_2 & \cdots & y_n & 0 \\ q_1 & q_2 & \cdots & q_n & q_{n+1} \end{bmatrix}, \quad q_i := \begin{bmatrix} w - \eta\nabla\mathcal{L}_n(w) \\ 0 \\ 1 \\ t_i \end{bmatrix} \in \mathbb{R}^{D-(d+1)}.$$

By Lemma 16, for any $\kappa > 0$, there exists a transformer network $f_{\text{Softmax}} \in \mathcal{T}_{\text{Softmax}}$, such that for any input $H \in [-B, B]^{d\times L}$, we have $\|f_{\text{Softmax}}(H) - f(H)\|_2 \leq \kappa$. Therefore, by the equivalence of matrix norms, $\|f_{\text{Softmax}}(H) - f(H)\|_{\max} \leq \kappa$ holds without loss of generality. Above $B := \max\{B_x, B_y, B_w, 1\}$ denotes the upper bound for every elements in $H$. Thus, we obtain $\overline{w}$ from the identical position of $w$ in $f_{\text{Softmax}}(H)$. Suppose we choose $\kappa = \frac{\epsilon}{\sqrt{D_w}}$, then it holds

$$\begin{aligned} \|\overline{w} - (w - \eta\nabla\mathcal{L}_n(w)\|_2 &\leq \sqrt{D_w}\|\overline{w} - (w - \eta\nabla\mathcal{L}_n(w)\|_{\max} \\ &\leq \|f_{\text{Softmax}} - f(H)\|_{\max} \\ &\leq \sqrt{D_w} \cdot \frac{\epsilon}{\sqrt{D_w}} \\ &\leq \epsilon. \end{aligned}$$

Finally, by our assumption, there exists an MLP layer such that for any $i \in [n+1]$, it maps

$$[x_i; y_i; w - \eta\nabla\mathcal{L}_n(w); \mathbf{0}; 1; t_i] \xrightarrow{\text{MLP}} [x_i; y_i; \text{Proj}_{\mathcal{W}}(w - \eta\nabla_w\mathcal{L}_n(w)); \mathbf{0}; 1; t_i].$$

Therefore, a four-layer transformer $f_{\text{Softmax}} \circ \text{MLP}$ is capable of implementing one-step gradient descent through ICL. As a direct corollary, there exist a $4L$-layer transformer consists of $L$ identical blocks $f_{\text{Softmax}} \circ \text{MLP}$ to approximate $L$ steps gradient descent algorithm. Each block approximates a one-step gradient descent algorithm on general risk function $\mathcal{L}_n(w)$. $\qquad\square$

### F.4   PROOF OF LEMMA 16

In this section, we introduce a helper lemma Lemma 17 to prove Lemma 16. At the beginning, we assume all input sequences are separated by a certain distance.

**Definition 12** (Token-wise Separateness, Definition 1 of (Kajitsuka and Sato, 2024)). Let $N \geq 1$ and $Z^{(1)}, \ldots, Z^{(N)} \in \mathbb{R}^{d \times n}$ be input sequences. Then, $Z^{(1)}, \ldots, Z^{(N)}$ are called token-wise $(r_{\min}, r_{\max}, \delta)$-separated if the following three conditions hold.

- For any $i \in [N]$ and $k \in [n]$, $\left\| Z_{:,k}^{(i)} \right\|_2 > r_{\min}$ holds.

- For any $i \in [N]$ and $k \in [n]$, $\left\| Z_{:,k}^{(i)} \right\|_2 < r_{\max}$ holds.

- For any $i, j \in [N]$ and $k, l \in [n]$ with $Z_{:,k}^{(i)} \neq Z_{:,l}^{(j)}$, $\left\| Z_{:,k}^{(i)} - Z_{:,l}^{(j)} \right\|_2 > \delta$ holds.

Note that we refer to $Z^{(1)}, \ldots, Z^{(N)}$ as token-wise $(r_{\max}, \epsilon)$-separated instead if the sequences satisfy the last two conditions.

Then we introduce the definition of contextual mapping. Intuitively, a contextual mapping can provide every input sequence with a unique id, which enables us to construct approximation for labels.

**Definition 13** (Contextual mapping, Definition 2 of (Kajitsuka and Sato, 2024)). Let input sequences $Z^{(1)}, \ldots, Z^{(N)} \in \mathbb{R}^{d \times n}$. Then, a map $q : \mathbb{R}^{d \times n} \to \mathbb{R}^{d \times n}$ is called an $(r, \delta)$-contextual mapping if the following two conditions hold:

- For any $i \in [N]$ and $k \in [n]$, $\left\| q\left( Z^{(i)} \right)_{:,k} \right\|_2 < r$ holds.

- For any $i, j \in [N]$ and $k, l \in [n]$, if $Z_{:,k}^{(i)} \neq Z_{:,l}^{(j)}$, then $\left\| q\left( Z^{(i)} \right)_{:,k} - q\left( Z^{(j)} \right)_{:,l} \right\|_2 > \delta$ holds.

In particular, $q\left( Z^{(i)} \right)$ for $i \in [N]$ is called a context id of $Z^{(i)}$.

Next, we show that a softmax-based 1-layer attention block with low-rank weight matrices is a contextual mapping for almost all input sequences.

**Lemma 17** (Softmax attention is contextual mapping, Theorem 2 of (Kajitsuka and Sato, 2024)). Let $Z^{(1)}, \ldots, Z^{(N)} \in \mathbb{R}^{d \times n}$ be input sequences with no duplicate word token in each sequence, that is,

$$Z_{:,k}^{(i)} \neq Z_{:,l}^{(i)}$$

for any $i \in [N]$ and $k, l \in [n]$. Also assume that $Z^{(1)}, \ldots, Z^{(N)}$ are token-wise $(r_{\min}, r_{\max}, \epsilon)$ separated. Then, there exist weight matrices $W^{(O)} \in \mathbb{R}^{d \times s}$ and $V, K, Q \in \mathbb{R}^{s \times d}$ such that the ranks of $V, K$ and $Q$ are all 1, and 1-layer single head attention with softmax, i.e., $\mathcal{F}_S^{(SA)}$ with $h = 1$ is an $(r, \delta)$-contextual mapping for the input sequences $Z^{(1)}, \ldots, Z^{(N)} \in \mathbb{R}^{d \times n}$ with $r$ and $\delta$ defined by

$$r = r_{\max} + \frac{\epsilon}{4}$$

$$\delta = \frac{2(\log n)^2 \epsilon^2 r_{\min}}{r_{\max}^2 (|\mathcal{V}| + 1)^4 (2 \log n + 3) \pi d} \exp\left( -(|\mathcal{V}| + 1)^4 \frac{(2 \log n + 3) \pi d r_{\max}^2}{4 \epsilon r_{\min}} \right)$$

Applying Lemma 17, we extends Proposition 1 of (Kajitsuka and Sato, 2024) to our Lemma 16.[1] We provide explicit upper bound of error $\| f_{\text{Softmax}}(Z) - f(Z) \|_2$ and analysis with function $f$ of a broader supported domain.

**Lemma 18** (Lemma 16 Restated: Universal Approximation of $\mathcal{T}_{\text{Softmax}}$). Let $f(\cdot) := \mathbb{R}^{d \times n} \to \mathbb{R}^{d \times n}$ be any $L$-Lipschitz permutation equivariant function supported on $[0, B_x]^{d \times n}$. We denote the discrete input domain of $[0, B_x]^{d \times n}$ by a grid $\mathbb{G}_D$ with granularity $D \in \mathbb{N}$ defined as $\mathbb{G}_D = \{B_x/D, 2B_x/D, \ldots, B_x\}^{d \times n} \subset \mathbb{R}^{d \times n}$. For any $\kappa > 0$, there exists a transformer network $f_{\text{Softmax}} \in \mathcal{T}_{\text{Softmax}}$ (Definition 11), such that for any $Z \in [0, B_x]^{d \times n}$, it approximate $f(Z)$ as: $\| f_{\text{Softmax}}(Z) - f(Z) \|_2 \leq \frac{L(8\sqrt{dn} + \sqrt{n})B_x}{2D} = \kappa$.

*Proof.* We begin our 3-step proof.

---

[1]This extension builds on the results of (Hu et al., 2024a), which extend the rank-1 requirement to any rank for attention weights. Additionally, Hu et al. (2024b) apply similar techniques to analyze the statistical rates of diffusion transformers (DiTs).

**Approximation of $f$ by piece-wise constant function.** Since $f$ is a continuous function on a compact set, $f$ has maximum and minimum values on the domain. By scaling with $\mathcal{F}_1^{(FF)}$ and $\mathcal{F}_2^{(FF)}$, $f$ is assumed to be normalized: for any $Z \in \mathbb{R}^{d \times n} \setminus [0, B_x]^{d \times n}$

$$f(Z) = 0,$$

and for any $Z \in [0, B_x]^{d \times n}$

$$-B_y \leq f(Z) \leq B_y.$$

Let $D \in \mathbb{N}$ be the granularity of a grid $\mathbb{G}_D$:

$$\mathbb{G}_D = \{\frac{B_x}{D}, \frac{2B_x}{D}, \ldots, B_x\}^{d \times n} \subset \mathbb{R}^{d \times n},$$

where each coordinate only take discrete value $B_x/D, 2B_x/D, ..., B_x$. Now with a continuous input $Z$, we approximate $f$ by using a piece-wise constant function $\bar{f}$ evaluating on the nearest grid point $L$ of $Z$ in the following way:

$$\bar{f}(Z) = \sum_{L \in \mathbb{G}_D} f(L) \, 1_{Z \in L + [-B_x/D, 0)^{d \times n}}. \tag{F.3}$$

Additionally if $Z \in L + [-1/D, 0)^{d \times n}$, denote it as $Q(Z) = L$.

Now we bound the piece-wise constant approximation error $\|f - \bar{f}\|$ as follows.

Define set $P_D = \{L + [-B_x/D, 0)^{d \times n} | L \in \mathbb{G}_D\}$. It is a set of regions of size $(\frac{B_x}{D})^{d \times n}$, whose vertexes are the points in $\mathbb{G}_D$.

For any subset $U \in P_D$, the maximal difference of $f$ and $\bar{f}$ in this region is:

$$\max_{Z \in U} \|f(Z) - \bar{f}(Z)\|_2 = \max_{Z \in U} \|f(Z) - f(Q(Z))\|_2$$

$$\leq \max_{Z, Z' \in U} \|f(Z) - f(Z')\|_2$$

$$\leq L \cdot \max_{Z, Z' \in U} \|Z - Z'\|_2 \qquad \text{(By } f \text{ is a } L\text{-Lipschitz function)}$$

$$= L \cdot \sqrt{dn \cdot (\frac{B_x}{D})^2} \qquad (Z, Z' \text{ are in the same } \frac{B_x}{D}\text{-wide } (d \cdot n)\text{-dimension } U.)$$

$$= \frac{L\sqrt{dn}B_x}{D}. \tag{F.4}$$

**Quantization of input using $\mathcal{F}_1^{(FF)}$.** In the second step, we use $\mathcal{F}_1^{(FF)}$ to quantize the continuous input domain into $\mathbb{G}_D$. This process is achieved by a multiple-step function, and we use ReLU functions to approximate this multiple-step functions. This ReLU function can be easily implemented by a one-layer feed-forward network.

First for any small $\delta > 0$ and $z \in \mathbb{R}$, we construct a $\delta$-approximated step function using ReLU functions:

$$\frac{\sigma_R \left[\frac{z}{\delta}\right] - \sigma_R \left[\frac{z}{\delta} - B_x\right]}{D} = \begin{cases} 0 & z < 0 \\ \frac{z}{\delta D} & 0 \leq z < \delta B_x \\ \frac{B_x}{D} & \delta B_x \leq z \end{cases}, \tag{F.5}$$

where a one-hidden-layer feed-forward neural network is able to implement this. By shifting (F.5) by $B_x$, for any $t \in [D-1]$, we have:

$$\frac{\sigma_R\left[\frac{z}{\delta} - \frac{tB_x}{\delta D}\right] - \sigma_R\left[\frac{z}{\delta} - B_x - \frac{tB_x}{\delta D}\right]}{D} = \begin{cases} 0 & z < \frac{tB_x}{D} \\ \frac{z}{\delta D} & \frac{tB_x}{D} \leq z < \delta B_x + \frac{tB_x}{D} \\ \frac{B_x}{D} & \delta B_x + \frac{tB_x}{D} \leq z \end{cases}, \qquad \text{(F.6)}$$

when $\delta$ is small the above function approximates to a step function:

$$\text{quant}_D^{(t)}(z) = \begin{cases} 0 & z \leq \frac{tB_x}{D} \\ \frac{B_x}{D} & \frac{tB_x}{D} \leq z \end{cases}.$$

By adding up (F.6) at every $t \in [D-1]$, we have an approximated multiple-step function

$$\sum_{t=0}^{D-1} \frac{\sigma_R\left[\frac{z}{\delta} - \frac{tB_x}{\delta D}\right] - \sigma_R\left[\frac{z}{\delta} - B_x - \frac{tB_x}{\delta D}\right]}{D} \qquad \text{(F.7)}$$

$$\approx \sum_{t=0}^{D-1} \text{quant}_D^{(t)}(z) \qquad \text{(when } \delta \text{ is small.)}$$

$$= \text{quant}_D(z)$$

$$= \begin{cases} 0 & z < 0 \\ \frac{B_x}{D} & 0 \leq z < \frac{B_x}{D} \\ \vdots & \vdots \\ B_x & B_x - \frac{B_x}{D} \leq z \end{cases}. \qquad \text{(F.8)}$$

Note that the error of approximation at $z$ here estimated as:

$$\left| \sum_{t=0}^{D-1} \frac{\sigma_R\left[\frac{z}{\delta} - \frac{tB_x}{\delta D}\right] - \sigma_R\left[\frac{z}{\delta} - B_x - \frac{tB_x}{\delta D}\right]}{D} - \text{quant}_D(z) \right| \leq \frac{B_x}{D}, \qquad \text{(F.9)}$$

and for matrix $Z \in \mathbb{R}^{d \times n}$:

$$\left\| \sum_{t=0}^{D-1} \frac{\sigma_R\left[\frac{Z}{\delta} - \frac{Q(Z)}{\delta D}\right] - \sigma_R\left[\frac{Z}{\delta} - B_x E - \frac{Q(Z)}{\delta D}\right]}{D} - \text{quant}_D(Z) \right\|_2$$

$$\leq \sqrt{d \times n \times \left(\frac{B_x}{D}\right)^2} \qquad (Z \in \mathbb{R}^{d \times n})$$

$$= \frac{B_x \sqrt{dn}}{D}.$$

Subtract the last step function from (F.7) we get the desired result:

$$\sum_{t=0}^{D-1} \frac{\sigma_R\left[\frac{z}{\delta} - \frac{tB_x}{\delta D}\right] - \sigma_R\left[\frac{z}{\delta} - B_x - \frac{tB_x}{\delta D}\right]}{D} - \left(\sigma_R\left[\frac{z}{\delta} - \frac{B_x}{\delta}\right] - \sigma_R\left[\frac{z}{\delta} - 1 - \frac{B_x}{\delta}\right]\right). \qquad \text{(F.10)}$$

This equation approximate the quantization of input domain $[0, B_x]$ into $\{B_x/D, \ldots, B_x\}$ and making $\mathbb{R} \setminus [0, B_x]$ to 0. In addition to the quantization of input domain $[0, B_x]$, we add a penalty

term for input out of $[0, B_x]$ in the following way:

$$- B_x \sigma_R \left[ \frac{(z - B_x)}{\delta} \right] + B_x \sigma_R \left[ \frac{(z - B_x)}{\delta} - 1 \right] - B_x \sigma_R \left[ \frac{-z}{\delta} \right] + B_x \sigma_R \left[ \frac{-z}{\delta} - 1 \right] \quad \text{(F.11)}$$

$$\approx \text{penalty}(z) = \begin{cases} -B_x & z \leq 0 \\ 0 & 0 < z \leq B_x \\ -B_x & B_x < z \end{cases}.$$

Both (F.10) and (F.11) can be realized by the one-layer feed-forward neural network. Also, it is straightforward to show that generate both of them to input $Z \in \mathbb{R}^{d \times n}$.

Combining both components together, the first feed-forward neural network layer $\mathcal{F}_1^{(FF)}$ approximates the following function $\overline{\mathcal{F}}_1^{(FF)}(Z)$:

$$\mathcal{F}_1^{(FF)} \approx \overline{\mathcal{F}}_1^{(FF)}(Z) = \text{quant}_D^{d \times n}(Z) + \sum_{t=1}^{d} \sum_{k=1}^{n} \text{penalty}(Z_{t,k}). \quad \text{(F.12)}$$

Note how we generalize $\text{penalty}(\cdot)$ to multi-dimensional occasions in the above equation. Whenever an input sequence $Z$ has one entry $Z_{t,k}$ out of $[0, B_x]^{d \times n}$, we penalize the whole input sequence by adding a $-B_x$ to all entries. This makes all entries of this quantization lower bounded by $-dnB_x$.

(F.12) quantizes inputs in $[0, B_x]^{d \times n}$ with granularity $D$, while every element of the output is non-positive for inputs outside $[0, B_x]^{d \times n}$. In particular, the norm of the output is upper-bounded when every entry in $Z$ is out of $[0, B_x]$, this adds $-dnB_x$ penalties to all entries:

$$\max_{Z \in \mathbb{R}^{d \times n}} \left\| \mathcal{F}_1^{(FF)}(Z)_{:,k} \right\|_2 = \sqrt{d \cdot (-dnB_x)^2} \qquad \text{(One column is } d-\text{dimension.)}$$

$$\leq dn \cdot \sqrt{d} B_x, \quad \text{(F.13)}$$

for any $k \in [n]$.

**Estimating the influence of self-attention $\mathcal{F}^{(SA)}$.** Define $\widetilde{\mathbb{G}}_D \subset \mathbb{G}_D$ as:

$$\widetilde{\mathbb{G}}_D = \{ L \in \mathbb{G}_D \mid \forall k, l \in [n], L_{:,k} \neq L_{:,l} \}. \quad \text{(F.14)}$$

It is a set of all the input sequences that don't have have identical tokens after quantization.

Within this set, the elements are at least $\frac{B_x}{D}$ separated by the quantization. Thus Lemma 17 allows us to construct a self-attention $\mathcal{F}^{(}SA)$ to be a contextual mapping for such input sequences.

Since when $D$ is sufficiently large, originally different tokens will still be different after quantization. In this context, we omit $\mathbb{G}_D / \widetilde{\mathbb{G}}_D$ for simplicity.

From the proof of Lemma 17 in (Kajitsuka and Sato, 2024), we follow their way to construct self-attention and have following equation:

$$\left\| \mathcal{F}_S^{(SA)}(Z)_{:,k} - Z_{:,k} \right\|_2 < \frac{1}{4\sqrt{d}D} \max_{k' \in [n]} \|Z_{:,k'}\|_2, \quad \text{(F.15)}$$

for any $k \in [n]$ and $Z \in \mathbb{R}^{d \times n}$.

Combining this upper-bound with (F.13) we have

$$\left\| \mathcal{F}_S^{(SA)} \circ \mathcal{F}_1^{(FF)}(Z)_{:,k} - \mathcal{F}^{(FF)}(Z)_{:,k} \right\|_2 < \frac{1}{4\sqrt{d}D} \max_{k' \in [n]} \|\mathcal{F}^{(FF)}(Z_{:,k})\|_2$$

$$< \frac{1}{4\sqrt{dD}} \times dn\sqrt{d}B_x \qquad \text{(By (F.13))}$$

$$= \frac{dnB_x}{4D}. \qquad \text{(F.16)}$$

We show that if we take large enough $D$, every element of the output for $Z \in \mathbb{R}^{d \times n} \backslash [0, B_x]^{d \times n}$ is upper-bounded by

$$\mathcal{F}_S^{(SA)} \circ \mathcal{F}_1^{(FF)}(Z)_{t,k} < \frac{B_x}{4D} \quad (\forall t \in [d], \ k \in [n]). \qquad \text{(F.17)}$$

To show (F.17) holds, we consider the opposite occasion that there exists a $\mathcal{F}_S^{(SA)} \circ \mathcal{F}_1^{(FF)}(Z)_{t_0,k_0} \geq B_x/4D$. Then we divide the case into two sub cases:

1. The whole $\mathcal{F}_1^{(FF)}(Z)$ receives no less than 2 penalties. In this occasion, since every entry consists of two counterparts in (F.12): the quantization part $\text{quant}_D^{d \times n}(Z) \in [0, B_x]$ and aggregated with a penalty part $\sum_{t=1}^{d} \sum_{k=1}^{n} \text{penalty}(Z_{t,k}) \leq -2B_x$, for every entry we have $\mathcal{F}^{(FF)}(Z)_{t,k} \leq -B_x$. This yields that:

$$\|\mathcal{F}_S^{(SA)} \circ \mathcal{F}_1^{(FF)}(Z)_{:,k_0} - \mathcal{F}^{(FF)}(Z)_{:,k_0}\|_2 \geq \|\mathcal{F}_S^{(SA)} \circ \mathcal{F}_1^{(FF)}(Z)_{t_0,k_0} - \mathcal{F}^{(FF)}(Z)_{t_0,k_0}\|_2$$

$$\geq |\frac{B_x}{4D} - (-B_x)|$$

$$\geq \frac{dn}{4D}B_x, \qquad \text{(for a large enough D)}$$

thus we derive a contradiction towards (F.16) from the assumption, proving it to be incorrect.

2. The whole $\mathcal{F}_1^{(FF)}(Z)$ receives only one penalty. In this case all entries in $Z$ is penalized by $-B_x$ and satisfies:

$$\mathcal{F}_1^{(FF)}(Z)_{t,k} \in [-B_x, 0]^{d \times n}. \qquad \text{(F.18)}$$

By (F.15), this further denotes:

$$\left\|\mathcal{F}_S^{(SA)} \circ \mathcal{F}_1^{(FF)}(Z)_{:,k} - \mathcal{F}_1^{(FF)}(Z)_{:,k}\right\|_2 < \frac{1}{4\sqrt{dD}} \max_{k' \in [n]} \|\mathcal{F}_1^{(FF)}(Z)_{:,k'}\|_2 \qquad \text{(By (F.15))}$$

$$\leq \frac{1}{4\sqrt{dD}} \sqrt{d \times B_x^2} \qquad \text{(By (F.18))}$$

$$= \frac{B_x}{4D}. \qquad \text{(F.19)}$$

Yet by our assumption, there exists such an entry $\mathcal{F}_S^{(SA)} \circ \mathcal{F}^{(FF)}(Z)_{t_0,k_0} \geq B_x/4D$, which since $\mathcal{F}_1^{(FF)}(Z)_{t_0,k_0} \leq 0$, yields:

$$\left\|\mathcal{F}_S^{(SA)} \circ \mathcal{F}_1^{(FF)}(Z)_{:,k_0} - \mathcal{F}_1^{(FF)}(Z)_{:,k_0}\right\|_2 \geq \left\|\mathcal{F}_S^{(SA)} \circ \mathcal{F}_1^{(FF)}(Z)_{t_0,k_0} - \mathcal{F}_1^{(FF)}(Z)_{t_0,k_0}\right\|_2$$

$$\geq |\frac{B_x}{4D} - 0|$$

$$= \frac{B_x}{4D}$$

The final conclusion contradict the former result, suggesting the prerequisite to be fallacious.

Joining the incorrectness of the two sub-cases of the opposite occasion, we confirm the upper bound when input $Z$ is outside $[0, B_x]^{d \times n}$ in (F.17).

For the input $Z$ inside $[0, B_x]^{d \times n}$, we now show it is lower-bounded by

$$\mathcal{F}_S^{(SA)} \circ \mathcal{F}_1^{(FF)}(Z)_{t,k} > \frac{3B_x}{4D} \quad (\forall t \in [d], \ k \in [n]). \tag{F.20}$$

By our construction, every entry $Z$ in $[0, B_x]^{d \times n}$ satisfies:

$$\mathcal{F}_1^{(FF)}(Z)_{t,k} \in [\frac{B_x}{D}, B_x]. \tag{F.21}$$

By (F.15):

$$\left\| \mathcal{F}_S^{(SA)} \circ \mathcal{F}_1^{(FF)}(Z)_{:,k} - \mathcal{F}_1^{(FF)}(Z)_{:,k} \right\|_2$$

$$< \frac{1}{4\sqrt{d}D} \max_{k' \in [n]} \| \mathcal{F}_1^{(FF)}(Z)_{:k'} \|_2 \qquad \text{(By (F.15))}$$

$$\leq \frac{1}{4\sqrt{d}D} \sqrt{d \times B_x^2} \qquad (d-\text{dimension vector with each entry has maximum value } B_x.)$$

$$= \frac{B_x}{4D}. \tag{F.22}$$

This yields:

$$|\mathcal{F}_S^{(SA)} \circ \mathcal{F}_1^{(FF)}(Z)_{t,k} - \mathcal{F}_1^{(FF)}(Z)_{t,k}| \leq \left\| \mathcal{F}_S^{(SA)} \circ \mathcal{F}_1^{(FF)}(Z)_{:,k} - \mathcal{F}_1^{(FF)}(Z)_{:,k} \right\|_2$$

$$< \frac{B_x}{4D}. \tag{F.23}$$

Finally, we have:

$$\mathcal{F}_S^{(SA)} \circ \mathcal{F}_1^{(FF)}(Z)_{t,k}$$

$$> \mathcal{F}_1^{(FF)}(Z)_{t,k} - \left\| \mathcal{F}_S^{(SA)} \circ \mathcal{F}_1^{(FF)}(Z)_{t,k} - \mathcal{F}_1^{(FF)}(Z)_{t,k} \right\|_2$$

$$> \frac{B_x}{D} - \| \mathcal{F}_S^{(SA)} \circ \mathcal{F}_1^{(FF)}(Z)_{t,k} - \mathcal{F}_1^{(FF)}(Z)_{t,k} \|_2 \qquad \text{(By (F.21).)}$$

$$> \frac{B_x}{D} - \frac{B_x}{4D} \qquad \text{(By (F.23))}$$

$$= \frac{3B_x}{4D}.$$

Hence we finally finish the proof for the upper bound of $\mathcal{F}_S^{(SA)} \circ \mathcal{F}_1^{(FF)}(Z)_{t,k}$ for $Z$ outside $[0, B_x]$ in (F.17) and lower bound for $Z$ inside $[0, B_x]$ in (F.20).

**Approximation error** Now, we can conclude our work by constructing the final feed-forward network $\mathcal{F}_2^{(FF)}$. It receives the output of the self-attention layer and maps the ones in $\widetilde{\mathbb{G}}_D \subset (3B_x/4D, \infty)^{d \times n}$ to the corresponding value of the target function, and the rest in $(-\infty, B_x/4D)^{d \times n}$ to 0.

In order to adapt to the $L_2$ norm, we use a continuous and Lipschitz function to map the input $Z$ to its targeted corresponding output $f(Q(Z))$.

According to piece-wise linear approximation, function $\mathcal{F}_2^{(FF)}$ exists such that for any input $L \in G_D$, it maps it to corresponding $f(L)$, and for an arbitrary input $Z$, its output suffices:

$$\mathcal{F}_2^{(FF)}(Z) \in [\min_{\|L-Z\|_{\max} \le \frac{B_x}{2D}} f(L), \max_{\|L-Z\|_{\max} \le \frac{B_x}{2D}} f(L)]. \tag{F.24}$$

Next we estimate the difference between $\mathcal{F}_2^{(FF)} \circ \mathcal{F}_S^{(SA)} \circ \mathcal{F}_1^{(FF)}$ and $\mathcal{F}_2^{(FF)} \circ \mathcal{F}_S^{(SA)} \circ \overline{\mathcal{F}}_1^{(FF)}$.

The difference is caused by the difference between $\overline{\mathcal{F}}_1^{(FF)}$ and $\mathcal{F}_1^{(FF)}$. By (F.9), this difference is bounded by $\frac{1}{D}$ in every dimension, for any input $Z \in \mathbb{R}^{d \times n}$:

$$\|\overline{\mathcal{F}}_1^{(FF)}(Z) - \mathcal{F}_1^{(FF)}(Z)\|_2 < \frac{\sqrt{dn}B_x}{D}.$$

By (F.19):

$$\begin{aligned}
&\|\mathcal{F}_S^{(SA)} \circ \overline{\mathcal{F}}_1^{(FF)}(Z) - \mathcal{F}_S^{(SA)} \circ \mathcal{F}_1^{(FF)}(Z)\|_2 \\
&\le \|\mathcal{F}_S^{(SA)} \circ \overline{\mathcal{F}}_1^{(FF)}(Z) - \overline{\mathcal{F}}_1^{(FF)}(Z)\|_2 + \|\overline{\mathcal{F}}_1^{(FF)}(Z) - \mathcal{F}_1^{(FF)}(Z)\|_2 \\
&\quad + \|\mathcal{F}_1^{(FF)}(Z) - \mathcal{F}_S^{(SA)} \circ \mathcal{F}_1^{(FF)}(Z)\|_2 && \text{(By triangle inequality)} \\
&\le \frac{\sqrt{dn}B_x}{D} + 2 \cdot \frac{\sqrt{n}B_x}{4D}. && \text{(By } \|A\|_2 \le \|A\|_F \text{ and (F.19))}
\end{aligned}$$

In the section on quantization of the input, we used piece-wise linear functions (F.7) to approximate piece-wise-constant functions (F.8), this creates a deviation for the inputs on the boundaries of the constant regions. Consider $Z$ as one of these inputs whose value deviated from $\mathcal{F}_2^{(FF)} \circ \mathcal{F}_S^{(SA)} \circ \overline{\mathcal{F}}_1^{(FF)}(Q(Z))$. Let $f(L_1)$ denote the value given to $\mathcal{F}_2^{(FF)} \circ \mathcal{F}_S^{(SA)} \circ \mathcal{F}_1^{(FF)}(Z)$. Because the deviation take the output to a grid at most $\sqrt{dn}B_x/D + \sqrt{n}B_x/2D$ away from its original grid, under the quantization of the output, $f(L_1)$ at most deviate from its original output $\mathcal{F}_2^{(FF)} \circ \mathcal{F}_S^{(SA)} \circ \overline{\mathcal{F}}_1^{(FF)}(Z)$ by the distance of $\sqrt{dn}B_x/D + \sqrt{n}B_x/2D$ aggregated with 2 times of the maximal distance within a grid. They sum up to be:

$$\begin{aligned}
\|\mathcal{F}_2^{(FF)} \circ \mathcal{F}_S^{(SA)} \circ \mathcal{F}_1^{(FF)} - \mathcal{F}_2^{(FF)} \circ \mathcal{F}_S^{(SA)} \circ \overline{\mathcal{F}}_1^{(FF)}\|_2 &\le L \times (\frac{2\sqrt{dn}B_x + \sqrt{n}B_x}{2D} + 2\frac{\sqrt{dn}B_x}{D}) \\
&< L\frac{6\sqrt{dn}B_x + \sqrt{n}B_x}{2D}.
\end{aligned}$$

Lastly, by condition we neglect the $\mathbb{G}_D \setminus \widetilde{\mathbb{G}}_D$ part. This yields:

$$\mathcal{F}_2^{(FF)} \circ \mathcal{F}_S^{(SA)} \circ \overline{\mathcal{F}}_1^{(FF)} = \overline{f}.$$

Thus, adding up the errors yields:

$$\begin{aligned}
&\|f - \mathcal{F}_2^{(FF)} \circ \mathcal{F}_S^{(SA)} \circ \mathcal{F}_1^{(FF)}\|_2 \\
&\le \|f - \overline{f}\|_2 + \|\overline{f} - \mathcal{F}_2^{(FF)} \circ \mathcal{F}_S^{(SA)} \circ \mathcal{F}_1^{(FF)}\|_2 && \text{(By triangle inequality)} \\
&= L\frac{6\sqrt{dn}B_x + \sqrt{n}B_x}{2D} + L\frac{\sqrt{dn}B_x}{D} && \text{(By (F.4))} \\
&= \frac{L(8\sqrt{dn} + \sqrt{n})B_x}{2D}.
\end{aligned}$$

This completes the proof. □

## G  EXPERIMENTAL DETAILS

In this section, we conduct experiments to verify the capability of ICL to learn deep feed-forward neural networks. We conduct the experiments based on 3-layer NN, 4-layer NN and 6-layer NN using both ReLU-Transformer and $\mathrm{Softmax}$-Transformer based on the GPT-2 backbone.

**Experimental Objectives.**    Our objectives include the following three parts:

- **Objective 1.** Validating the performance of ICL matches that of training $N$-layer networks, i.e., the results in Theorem 1, Theorem 5, and Theorem 6.
- **Objective 2.** Validating the ICL performance in scenarios where the testing distribution diverges from the pretraining one or where prompt lengths exceed those used in pretraining.
- **Objective 3.** Validating the ICL performance in scenarios where the distribution of parameters in the $N$-layer network diverges from that of the pretraining phase.
- **Objective 4.** Validating that a deeper transformer achieves better ICL performance, supporting the idea that scaling up the transformer enables it to perform more ICGD steps.

**Computational Resource.** We conduct all experiments using 1 NVIDIA A100 GPU with 80GB of memory. Our code is based on the PyTorch implementation of the in-context learning for the transformer (Garg et al., 2022) at `https://github.com/dtsip/in-context-learning`.

### G.1  EXPERIMENTS FOR OBJECTIVES 1 AND 2

In this section, we conduct experiments to validate Objectives 1 and 2. We sample the input of feed-forward network $x \in \mathbb{R}^d$ from the Gaussian mixture distribution: $w_1 N(-2, I_d) + w_2 N(2, I_d)$, where $w_1, w_2 \in \mathbb{R}$. We consider three kinds of network $f : \mathbb{R}^d \to \mathbb{R}$, (i) 3-layer NN, (ii) 4-layer NN, and (iii) 6-layer NN. We generate the true output by $y = f(x)$. In our setting, we use $d = 20$.

**Model Architecture.** The sole difference between ReLU-Transformer and $\mathrm{Softmax}$-Transformer is the activation function in the attention layer. Both models comprise 12 transformer blocks, each with 8 attention heads, and share the same hidden and MLP dimensions of 256.

**Transformer Pretraining.** We pretrain the ReLU-Transformer and $\mathrm{Softmax}$-Transformer based on the GPT-2 backbone. In our setting, we sample the pertaining data from $N(-2, I_d)$, i.e., $w_1 = 1$ and $w_2 = 0$. Following the pre-training method in (Garg et al., 2022), we use the batch size as 64. To construct each sample in a batch, we use the following steps (take the generation for the $i$-th sample as an example):

1. Initialize the parameters in $f_i$ with a standard Gaussian distribution, i.e., $N(0, I)$.
2. Generate $n$ queries $\{x_{i,j}\}_{j=1}^n$ (i.e., input of $f_i$) from the Gaussian mixture model $\omega_1 N(-2, I_d) + \omega_2 N(2, I_d)$. Here we take $n = 51$.
3. For each query $x_{i,j}$, use $y_{i,j} = f_i(x_{i,j})$ to calculate the true output.

This generates a training sample for the transformer model with inputs

$$[x_{i,1}, y_{i,1}, \cdots, x_{i,50}, y_{i,50}, x_{i,51}],$$

and training target

$$o_i = [y_{i,1}, \cdots, y_{i,50}, y_{i,51}].$$

We use the MSE loss between prediction and true value of $o_i$. The pretraining process iterates for 500k steps.

**Testing Method.** We generate samples similar to the pretraining process. The batch size is 64, and the number of batch is 100, i.e., we have 6400 samples totally. For each sample, we extend the value $n$ from 51 to 76 to learn the performance of in-context learning when the prompt length is longer than we used in pretraining. The input to the model becomes

$$[x_{i,1}, y_{i,1}, \cdots, x_{i,75}, y_{i,75}, x_{i,76}].$$

We assess performance using the mean R-squared value for all 6400 samples.

**Baseline.** We use the 3-layer, 4-layer, and 6-layer feed-forward neural networks with 200 hidden dimensions as baselines by training them with in-context examples. Specially, given a testing sample (take the $i$-th sample as an example), which includes prompts $\{x_{i,j}, y_{i,j}\}_{j=1}^{k-1}$ and a test query $x_{i,k}$. We use $\{x_{i,j}, y_{i,j}\}_{j=1}^{k-1}$ to train the network with MSE loss for 100 epochs. We select the highest R-squared value from each epoch as the testing measure and calculate the average across all 6400 samples.

### G.1.1 PERFORMANCE OF RELU TRANSFORMER.

We use three different Gaussian mixture distribution $\omega_1 N(-2, I_d) + \omega_2 N(2, I_d)$ for the testing data: (i) $\omega_1 = 1, \omega_2 = 0$, (ii) $\omega_1 = 0.9, \omega_2 = 0.1$, (iii) $\omega_1 = 0.7, \omega_2 = 0.3$, (iv) $\omega_1 = 0.5, \omega_2 = 0.5$. Here the distribution in the first setting matches the distribution in pretraining. We show the results in Figure 1.

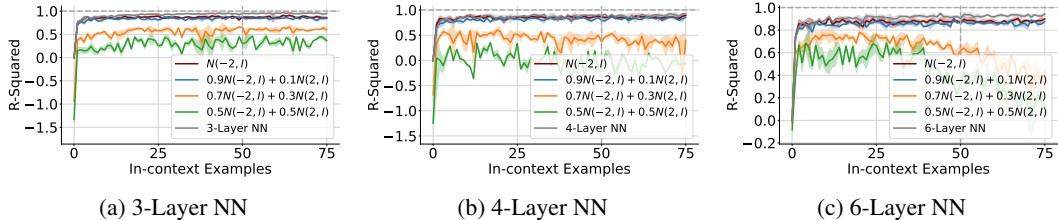

(a) 3-Layer NN       (b) 4-Layer NN       (c) 6-Layer NN

Figure 1: **Performance of ICL in ReLU-Transformer:** ICL learns 3-layer, 4-layer, and 6-layer NN and achieves R-squared values comparable to those from training with prompt samples. The results also show the ICL performance declines as the testing distribution diverges from the pretraining one.

### G.1.2 PERFORMANCE OF SOFTMAX TRANSFORMER.

We use three different Gaussian mixture distribution $\omega_1 N(-2, I_d) + \omega_2 N(2, I_d)$ for the testing data: (i) $\omega_1 = 1, \omega_2 = 0$, (ii) $\omega_1 = 0.9, \omega_2 = 0.1$, (iii) $\omega_1 = 0.7, \omega_2 = 0.3$, (iv) $\omega_1 = 0.5, \omega_2 = 0.5$. Here the distribution in the first setting matches the distribution in pretraining. We show the results in Figure 2.

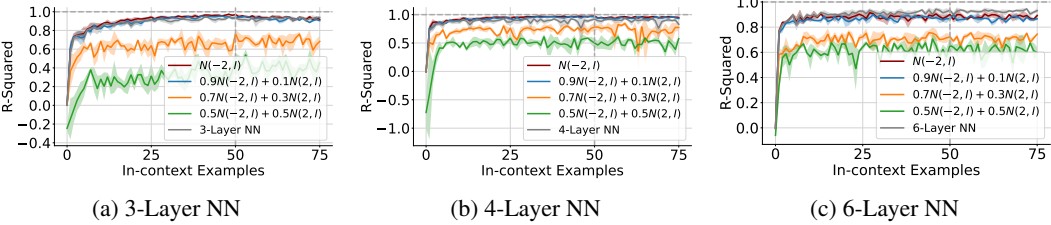

(a) 3-Layer NN       (b) 4-Layer NN       (c) 6-Layer NN

Figure 2: **Performance of ICL in** Softmax**-Transformer:** ICL learns 3-layer, 4-layer, and 6-layer NN and achieves R-squared values comparable to those from training with prompt samples. The results also show the ICL performance declines as the testing distribution diverges from the pretraining one. Note that performance decreases when the prompt length exceeds the pretraining length (i.e., 50), a well-known issue (Dai et al., 2019; Anil et al., 2022). We believe this is due to the absolute positional encodings in GPT-2, as noted in (Zhang et al., 2023)

The results in Appendix G.1.1 and Appendix G.1.2 show that the performance of ICL in the transformer matches that of training $N$-layer networks, regardless of whether the prompt lengths are within or exceed those used in pretraining. Furthermore, the ICL performance declines as the testing distribution diverges from the pretraining one.

### G.2 EXPERIMENTS FOR OBJECTIVE 3

In this section, we conduct experiments to validate Objective 3. For these experiments, we use testing data that is identical to the training data, which follows a distribution of $N(-2, I_d)$. We vary the

distribution of parameters in the $N$-layer network. During the training process, we set the distribution as $N(0, I)$. In the testing process, we examine different distributions, including $N(0, I)$, $N(-0.5, I)$, and $N(0.5, I)$. All other model hyperparameters and experimental details remain consistent with those described in Appendix G.1. We evaluate the ICL performance of both the ReLU-Transformer and the Softmax-Transformer for 4-layer networks, as shown in Figure 3 and Figure 4. The results demonstrate that the ICL performance in the transformer matches that of training $N$-layer networks, regardless of whether the parameter distribution in the $N$-layer network diverges from that of the pretraining phase.

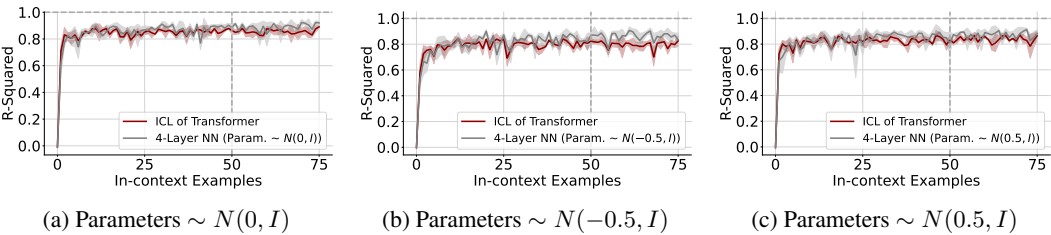

(a) Parameters $\sim N(0, I)$       (b) Parameters $\sim N(-0.5, I)$       (c) Parameters $\sim N(0.5, I)$

Figure 3: **Performance of ICL Across Various $N$-layer Network Parameter Distributions for the ReLU-Transformer:** ICL learns 4-layer NN and achieves R-squared values comparable to those from training with prompt samples, even when the parameter distribution in the $N$-layer network during testing diverges from that in the pretraining phase ($N(0, I)$).

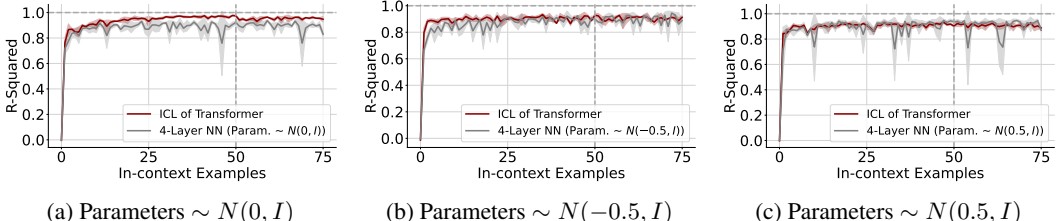

(a) Parameters $\sim N(0, I)$       (b) Parameters $\sim N(-0.5, I)$       (c) Parameters $\sim N(0.5, I)$

Figure 4: **Performance of ICL Across Various $N$-layer Network Parameter Distributions for the Softmax-Transformer:** ICL learns 4-layer NN and achieves R-squared values comparable to those from training with prompt samples, even when the parameter distribution in the $N$-layer network during testing diverges from that in the pretraining phase ($N(0, I)$).

### G.3 EXPERIMENTS FOR OBJECTIVE 4

In this section, we conduct experiments to validate Objective 4. For these experiments, we use testing data identical to the pertaining data from $N(-2, I_d)$. We vary the number of layers in the transformer architecture, testing configurations with 4, 6, 8 and 10 layers. All other model hyperparameters and experimental details remain consistent with those described in Appendix G.1. We evaluate the ICL performance of both the ReLU-Transformer and the Softmax-Transformer with 15, 30, and 45 in-context examples, as shown in Figure 5. The results show that a deeper transformer achieves better ICL performance, supporting the idea that scaling up the transformer enables it to perform more ICGD steps.

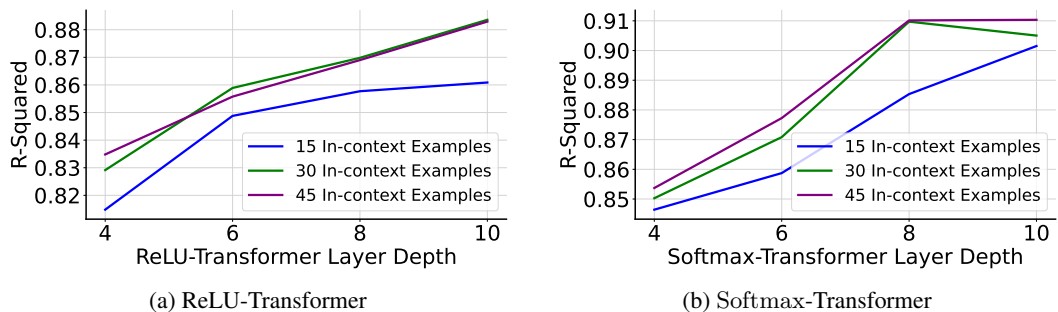

(a) ReLU-Transformer                    (b) Softmax-Transformer

Figure 5: **Performance of ICL Across Varying Transformer Depths**: We use the number of in-context examples as 15, 30, or 45 for both the ReLU-Transformer and the Softmax-Transformer. The results show that a deeper transformer achieves better ICL performance, supporting the idea that scaling up the transformer enables it to perform more ICGD steps.

# H APPLICATION: ICL FOR DIFFUSION SCORE APPROXIMATION

In this part, we give an important application of our work, i.e., learn the score function of diffusion models by the in-context learning of transformer models. We give the preliminaries about score matching generative diffusion models in Appendix H.1. Then, we give the analysis for ICL to approximate the diffusion score function in Appendix H.2.

## H.1 SCORE MATCHING GENERATIVE DIFFUSION MODELS

**Diffusion Model.** Let $x_0 \in \mathbb{R}^d$ be initial data following target data distribution $x_0 \sim P_0$. In essence, a diffusion generative model consists of two stochastic process in $\mathbb{R}^d$:

- A forward process gradually add noise to the initial data (e.g., images): $x_0 \to x_1 \to \cdots \to x_T$.
- A backward process gradually remove noise from pure noise: $y_T \to y_{T-1} \to \cdots \to y_0$.

Importantly, the backward process is the reversed forward process, i.e., $y_t \overset{\mathrm{d}}{\approx} x_{T-t}$ for $i \in 0, \ldots, T$.[2] This allows the backward process to reconstruct the initial data from noise, and hence generative. To achieve this time-reversal, a diffusion model learns the reverse process by ensuring the backward conditional distributions mirror the forward ones. The most prevalent technique for aligning these conditional dynamics is through "score matching" — a strategy training a model to match score function, i.e., the gradients of the log marginal density of the forward process (Song et al., 2020b;a; Vincent, 2011). To be precise, let $P_t, p_t(\cdot)$ denote the distribution function and destiny function of $x_t$. The score function is given by $\nabla \log p_t(\cdot)$. In this work, we focus on leveraging the in-context learning (ICL) capability of transformers to emulate the score-matching training process.

**Score Matching Loss.** We introduce the basic setting of score-matching as follows[3]. To estimate the score function, we use the following loss to train a score network $s_W(\cdot, t)$ with parameters $W$:

$$\min_W \int_{T_0}^T \gamma(t) \mathbb{E}_{x_t \sim P_t} \left[ \|s_W(x_t, t) - \nabla \log p_t(x_t)\|_2^2 \right] \mathrm{d}t, \quad \text{where } \gamma(t) \text{ is a weight function,} \quad \text{(H.1)}$$

and $T_0$ is a small value for stabilizing training and preventing the score function from diverging. In practice, as $\nabla \log p_t(\cdot)$ is unknown, we minimize the following equivalent loss (Vincent, 2011).

$$\min_W \int_{T_0}^T \gamma(t) \mathbb{E}_{x_0 \sim P_0} \left[ \mathbb{E}_{x_t \mid x_0} \left[ \|s_W(x_t, t) - \nabla \log p(x_t \mid x_0)\|_2^2 \right] \right] \mathrm{d}t, \quad \text{(H.2)}$$

where $p(x_t \mid x_0)$ is distribution of $x_t$ conditioned on $x_0$.

## H.2 ICL FOR SCORE APPROXIMATION

We first give the problem setup about the ICL for score approximation as the following:

**Problem 3** (In-Context Learning (ICL) for Score Function $\nabla \log p_t(\cdot)$)**.** Consider the score function $\nabla \log p_t(\cdot)$ for any $t \geq 0$. Given a dataset $\mathcal{D}_n := \{(x_i, y_i)\}_{i \in [n]}$, where $\{x_i\}_{i \in [n]} \subseteq \mathbb{R}^d$ and $y_i = \nabla \log p_{t_i}(x_i) \subseteq \mathbb{R}^d$ ($t_i \geq 0$), and a test input $x_{n+1}$, the goal of "ICL for Score Function" is to find a transformer $\mathcal{T}$ to predict $y_{n+1}$ based on $x_{n+1}$ and the in-context dataset $\mathcal{D}_n$. In essence, the desired transformer $\mathcal{T}$ serves as the trained score network $s_W(\cdot, t)$.

To solve Problem 3, we follow two steps: (i) Approximate the diffusion score function $\nabla \log p_t(\cdot)$ with a multi-layer feed-forward network with ReLU activation functions under the given training dataset $\mathcal{D}_n$. (ii) Approximate the gradient descent used to train this network by the in-context learning of the Transformer until convergence, using the same training set $\mathcal{D}_n$ as the prompts of ICL.

For the first step, we follow the score approximation results based on a multi-layer feed-forward network with ReLU activation in (Chen et al., 2023), stated as next lemma.

---

[2] $\overset{\mathrm{d}}{\approx}$ denotes distributional equivalence.
[3] Please also see Appendix B.1 and (Chen et al., 2024; Chan, 2024; Yang et al., 2023) for overviews.

**Lemma 19** (Score Approximation by Feed-Forward Networks, Theorem 1 of (Chen et al., 2023)). Given an approximation error $\epsilon > 0$, for any initial data distribution $P_0$, there exist a multi-layer feed-forward network with ReLU activation, $f(w, x, t) : \mathbb{R}^{D_w} \times \mathbb{R}^d \times \mathbb{R} \to \mathbb{R}^d$. Then for any $t \in [T_0, T]$, we have $\|f(w, \cdot, t) - \nabla \log p_t(\cdot)\|_{L^2(P_t)} \leq \mathcal{O}(\epsilon)$.

With the approximation result, we reduce the Problem 3 to Problem 2, where the loss function is (H.1). Following Theorem 1, we show that the in-context learning of transformer models can approximate the score function of diffusion model.

