# OpenReview forum: "Pretrained Transformers are Deep Optimizers: Provable In-Context Learning for Deep Model Training"
_ICLR.cc/2025/Conference — Submitted to ICLR 2025_

### Official Review · Reviewer_hJr1 · 2024-11-01

**Soundness:** 3
**Presentation:** 2
**Contribution:** 3
**Rating:** 6
**Confidence:** 3

**Summary:**

**Summary:** In this paper, the authors show that there exists a choice of weights, under which a Transformer can implement gradient descent on $f(w,\cdot)$ over a dataset $\\{x_i,y_i\\}_{i=1...n}$. The paper focuses on the setting where $f$ is the loss of a $N$-layer neural network, and $w$ is its parameters. The authors provide the explicit construction, along with error bounds on the approximation error to a real $L$-step gradient descent.

I lean towards recommending reject for this paper. The reason is that I am unclear about the purpose of the main result. I will elaborate below.

**Strengths:**

1. The construction is *mostly* explicit, and is clearly described in the various proofs. The proofs seem to be correct, though I have not carefully verified each line.
2. To simulate $L$ gradient descent steps of a $N$ layer neural network, one only needs a $(2N+4)L$-layer Transformer.
3. The recursive expression Lemma 1 seems to work well with a multi-layer Transformer.
4. Being able to learn a neural network in-context with a Transformer seems interesting.

**Weaknesses:**

Transformers have been shown to have high representation power. **If one does not care about the complexity of the construction,** [1] claims that the Transformer can implement arbitrary algorithms by literally simulating a computer; Lemma 14 of [1] claims that there is a 13 layer 1-head Transformer that implements T steps of gradient descent on a 2-layer sigmoid network.

In the other extreme, [2] (and numerous follow-up works) considered **simple problems**, but also provided **simple constructions** which are provably learnable during training.

My main criticism of the results of this paper is that the results here falls somewhere in-between the two extremes above -- the target problem of "can Transformers implement gradient descent in-context for a neural network" is challenging, and thus required a rather **complex construction**. If so, why would I not simply use the general construction in [1]?

The constructions in this paper are likely not being learned, in theory or in experiments, so what makes it superior to the generic construction in [1]? This is further compounded by the fact that the **error bounds in this paper are large**, growing exponentially in number of layers and steps. Additionally, the dimensions of a single Transformer layer need to be large enough to accomodate the weight parameters of the entire $N$-layer neural network, **requiring a lot of memory**, and this requirement seems unavoidable.

A motivation of this paper was to have foundation models train other deep models, but the various issues listed above make me question whether the approach in this paper will lead to meaningful results.

Below I provide further details on some of the issues I have with the paper

1. The construction is not entirely explicit, Lemmas 2,3,4 appeal to the existence of some combination of ReLU functions to approximate sufficiently smooth functions via the $(\epsilon, R, H, C)$-approximable assumption.

3. Theorem 1 requires the activation $r(t)$ and $r'(t)$ to be Lipschitz continuous, but ReLU does not satisfy this. It would be helpful for the authors to state a few concrete examples of N-layer neural networks (along with specific activations), and state what the Lipschitz constants/dimensions/error bounds end up being.
4. The error bounds are quite large. Corollary 1.1 has exponential error in $L$, Lemma 6 has error growing as $d^N$. Some of the error is unavoidable -- divergence in gradient desecent over non-convex functions, and exponentially growing lipschitz constant of a $N$-layer Transformer. It would be useful if the authors can clearly distinguish the error terms inherent due to gradient descent over a non-convexity of the problem, vs the error terms that are due to the construction.
5. Experiments are a little unclear. What is the exact Transformer architecture used? Are the lines for 3/4/6 layer NNs in Figure 1 obtained from training a 3/4/6 layer NN over the in-context examples? Or are they the output of the "true network parameters for $f_i$" on line 2393?
6. The Transformer performance in experiments seems quite bad for 0.7/0.3 data. Can the authors also show the plot for 0.5/0.5?
7. If possible, it would be useful to see if scaling up the Transformer, as described in the paper, will indeed enable it to implement more gradient descent steps. (a crude way is perhaps to simply plot loss against number of Transformer layers.)

**Questions:**

1. Can the authors please compare your construction to the one in Algorithm 10 of [1]? Are there any similar ideas used?
2. If you are already assuming that various parts of the neural network are approximable by relu, and if you are already incurring exponentially-growing error terms, why do you not just treat the entire neural network as a $(\epsilon, R, H, C)$-approximable function, and simply use the Transformer to learn relu coefficients? Will this lead to worse bounds?
3. The dimension of the Transformer is very large, as each layer needs to contain all the parameters of the N-layer neural network. Can the authors please comment on what the exact dimension of the Transformer needed? I think it is useful to state this value in Theorem 1.

**minor suggestions that did not affect score**
1. $L$ is used for both loss, number of gradient descent steps, and Lipschitz constant. This gets confusing at times, and I suggest for the authors to use different letters.

[1] [Looped Transformers as Programmable Computers](https://arxiv.org/abs/2301.13196), Giannou et el 2023

[2] [Transformers learn in-context by gradient descent](https://arxiv.org/abs/2212.07677), von Oswald et el 2023

---

> ### Author Response · Authors · 2024-11-24
> **Response 1**
>
> Thank you for your review.
>
> We have addressed all your comments and questions in this and the following responses.
>
> The draft has been updated accordingly, with changes highlighted in **blue** in the latest revision.
>
> Please refer to the updated PDF for details.
>
> ---
>
> >`Weakness.` Transformers have been shown to have high representation power. If one does not care about the complexity of the construction, [1] claims that the Transformer can implement arbitrary algorithms by literally simulating a computer; Lemma 14 of [1] claims that there is a 13 layer 1-head Transformer that implements T steps of gradient descent on a 2-layer sigmoid network. In the other extreme, [2] (and numerous follow-up works) considered simple problems, but also provided simple constructions which are provably learnable during training. My main criticism of the results of this paper is that the results here falls somewhere in-between the two extremes above -- the target problem of "can Transformers implement gradient descent in-context for a neural network" is challenging, and thus required a rather complex construction. If so, why would I not simply use the general construction in [1]? The constructions in this paper are likely not being learned, in theory or in experiments, so what makes it superior to the generic construction in [1]?
>
> Thank you for your feedback. Here are some clarifications.
>
> Yes, the construction is complex. However, we have the following advantages.
>
> * Our construction provides an explicit construction for challenging problems.
> * Our construction offers theoretical support for the capability of ICL in transformers to implement ICGD on deep models. We also derive convergence guarantees, error bounds, and upper limits on parameter matrices.
>
> These two contributions go beyond the approach presented in [Giannou23], which does not address all the theoretical aspects we cover. Moreover, our analysis is not built upon the framework provided by [Giannou23], nor do we rely on the operations described in their work to implement ICGD. Instead, we propose a more effective method through explicit construction.
>
> Under an $\epsilon$-error guarantee, we explicitly characterize the dimensions, attention heads, and parameters of the Transformers required to implement ICGD. Additionally, we offer a clear and explicit method to construct such Transformers under a general setting. Notably, our approach imposes no restrictions on the activation function, loss function, or the width and depth of the neural network.
>
> [Giannou23] Looped Transformers as Programmable Computers, ICML 2023.
>
> >`Weakness.` This is further compounded by the fact that the error bounds in this paper are large, growing exponentially in number of layers and steps. Additionally, the dimensions of a single Transformer layer need to be large enough to accomodate the weight parameters of the entire N-layer neural network, requiring a lot of memory, and this requirement seems unavoidable.
>
> Thank you for your insightful comment.
>
> We acknowledge that the error bounds are large, and the exponential growth is expected. This is because gradient descent on non-convex objectives is inherently unstable at a high level; a slight error added at each step can result in drastically different trajectories. While assuming the convexity of $\nabla L$ could yield a linear error bound result, we believe this assumption is overly restrictive and unrealistic for the loss of $N$-layer neural network. Additionally, as shown in Lemma 14, we prove the convergence of ICGD, demonstrating that it can reach local maxima or minima, similar to other gradient-based methods.
>
> We acknowledge that the dimension of the transformer is large. In our revised paper, we provide a precise specification of the dimension required in Theorem 1, i.e., $O(NK^2)$, where $K$ is the hidden dimension of the $N$-layer network. However, there are two additional clarifications regarding this:
>
> * The reason for the large dimension is that if ICL is used to perform ICGD on the $N$-layer network, we need to allow the transformer to realize the $N$-layer network parameters. This means that it is reasonable for the input dimension to be $O(NK^2)$. However, it is possible to reduce the hidden dimension and MLP dimension of the transformer through smarter construction. We have acknowledged this in the limitations (`line 828-833`) and leave it for future work.
>
> * Although the transformer's dimension is large, many elements are zeros (e.g., Eq. (D.8) in `line 1117-1127`). The parameters in the transformer are very sparse. This sparsity allows for the potential use of existing techniques designed for sparse matrix saving or multiplication to save memory [Gao24, Zhang20].
>
> [Gao24] A Systematic Literature Survey of Sparse Matrix-Vector Multiplication, arXiv 2024.
>
> [Zhang20] SpArch:Efficient Architecture for Sparse Matrix Multiplication, HPCA 2020.

---

> ### Author Response · Authors · 2024-11-24
> **Response 2**
>
> >`W1.` The construction is not entirely explicit. Lemmas 2, 3, 4 appeal to the existence of some combination of ReLU functions to approximate sufficiently smooth functions via the $(\epsilon, R, H, C)$-approximable assumption.
>
> Thank you for your comment.
>
> We apologize for any confusion. Our construction is entirely explicit. Here are the clarifications:
>
> The parameters in our construction are not fully specified because we do not constrain the activation function $r$ or the loss function $l$ to specific forms. However, if we specify the activation function $r$ (e.g., sigmoid) and the loss function $l$ (e.g., mean squared error), we could derive the coefficients of the sum of ReLUs by solving the optimization problem under the specified constraints and a given $\epsilon$ (e.g., 1e-5), as defined in Definition 4. Once the coefficients are determined, the constructions remain explicit.
>
> >`W2.` Theorem 1 requires the activation $r(t)$ and $r'(t)$ to be Lipschitz continuous, but ReLU does not satisfy this. It would be helpful for the authors to state a few concrete examples of $N$-layer neural network (along with specific activations) and state what the Lipschitz constants/dimensions/error bounds end up being.
>
> Thank you for your insightful comment.
>
> Yes, you are correct. ReLU and its derivatives are not Lipschitz continuous.
>
> For concrete examples, we consider smooth activation functions, which cannot be exactly expressed by ReLU-based transformers, such as Sigmoid, Tanh, and Softplus. Take the $N$-layer network with the Sigmoid activation function as an example. Let the width of the neural network be $K=32$, and let the upper bounds of the parameter matrix 2-norm be $w$. The Lipschitz constant is $L_f = N \cdot (w/4)^{2(N-1)}$. For any $\epsilon > 0$, the dimension bound is $O(1024N)$. The error bound is $\leq L_f^{-1} (1 + n L_f)^l \epsilon$, where $L_f = N \cdot (w/4)^{2(N-1)}$ (Corollary 1.1).
>
> Additionally, deriving results is straightforward when the activation function is ReLU. Our results are based on Lemma 7, which approximates smooth activation functions with a sum of ReLUs. If the activation function in the neural network is already ReLU, no approximation is necessary.
>
> We have demonstrated that transformers can simulate both the forward pass of networks with the ReLU activation function and the behavior of the derivative of ReLU. These results ensure that the lack of Lipschitz continuity in ReLU does not undermine the broader objectives of our theoretical framework.
>
> >`W3.` The error bounds are quite large. Corollary 1.1 has exponential error in $L$, Lemma 6 has error growing as $d^N$. Some of the error is unavoidable -- divergence in gradient descent over non-convex functions, and exponentially growing Lipschitz constant of a $N$-layer Transformer. It would be useful if the authors can clearly distinguish the error terms inherent due to gradient descent over a non-convexity of the problem, vs the error terms that are due to the construction.
>
> Thank you for your feedback.
>
> The error in Theorem 1 is due to our construction, while the error in Corollary 1.1 is attributable to the non-convexity of the problem. The error terms resulting from the construction can indeed be controlled at each step (as demonstrated in Theorem 1) by appropriately scaling the models. However, the exponential growth of the error is inherent to the non-convex nature of the problem, representing the best theoretical result achievable in this setting.
>
> We acknowledge that the error between ICGD and true gradient descent could be relatively large. Nonetheless, as demonstrated in Lemma 14, ICGD exhibits convergence and maintains a unique descending tendency, aligning it with the behavior of gradient methods.
>
> >`W4.` Experiments are a little unclear. What is the exact Transformer architecture used? Are the lines for 3/4/6 layer NNs in Figure 1 obtained from training a 3/4/6 layer NN over the in-context examples? Or are they the output of the ``true network parameters for $f^*_i$'' on line 2393?
>
> Thank you for your question. We apologize for the confusion and have revised the corresponding content in the paper.
>
> Transformer architecture: The sole difference between ReLU-Transformer and Softmax-Transformer is the activation function in the attention layer.
> Both models comprise 12 transformer blocks, each with 8 attention heads, and share the same hidden and MLP dimensions of 256.
>
> The lines for “3/4/6-layer NNs” in Figure 1 are obtained from training a 3/4/6-layer NN over the in-context examples.

---

> ### Author Response · Authors · 2024-11-24
> **Response 3**
>
> >`W5.` The Transformer performance in experiments seems quite bad for 0.7/0.3 data. Can the authors also show the plot for 0.5/0.5?
>
> Thanks for your question.
>
> Yes, the result for “0.7/0.3” data is bad, and we have included the results for “0.5/0.5” in Appendix G.1. The result is worse than “0.7/0.3”. Please refer to the revised version for the details.
>
> Additionally, we have the following clarifications:
>
> * If we just aim to determine whether the ICL performance is comparable to traditional $N$-layer network training, we should compare the performance of '$N(-2, I)$' with that of '$N$-layer NN' only when there are 50 or fewer in-context examples in Figures 1 and 2.
>
> * For additional results, we seek to explore the characteristics of ICL further: (i) For results involving more than 50 in-context examples, we aim to investigate the ICL performance under conditions of extended prompt lengths beyond those used in pretraining. (ii) For distributions $0.9N(-2, I) + 0.1N(2, I)$, $0.7N(-2, I) + 0.3N(2, I)$, and $0.5N(-2, I) + 0.5N(2, I)$, our goal is to explore how ICL performance varies as the testing distribution diverges from the pretraining one.
>
> >`W6.` If possible, it would be useful to see if scaling up the Transformer, as described in the paper, will indeed enable it to implement more gradient descent steps. (A crude way is perhaps to simply plot loss against the number of Transformer layers.)
>
> Thank you for your suggestion. We agree that including this is very helpful.
>
> We have added the results for R-squared against the number of transformer layers (4, 6, 8, and 10 layers) in Appendix G.2. The results demonstrate that the R-squared value increases with the increasing number of transformer layers. Please refer to the revised version for the details.
>
> >`Q1.` Can the authors please compare your construction to the one in Algorithm 10 of [1]? Are there any similar ideas used?
>
> Thank you for your thoughtful question. Here are the comparisons.
>
> Algorithm 10 in [Giannou23] focuses on the use of transformers within the unified attention-based computer framework to implement SGD on 2-layer networks. While there are thematic overlaps, our work significantly extends beyond this scope. Below, we outline the key differences:
>
> * **Scope of application**: Algorithm 10 in [Giannou23] is confined to a simple task under a non-general setting, specifically targeting 2-layer networks. In contrast, our primary contribution demonstrates the capability of transformers to implement gradient computation for $N$-layer networks in more general settings. For the $N$-layer networks, the calculation is not trivial, and we derive an explicit formula for gradient computation in $N$-layer feed-forward neural networks (Lemma 1).
>
> * **Explicit construction of transformers**: We provide explicit and detailed transformer construction to tackle the complex problem. Additionally, we establish a relationship between the approximation error and various transformer parameters such as attention heads, hidden dimensions, and parameter bounds. Conversely, [Giannou23] only provides a general construction.
>
> * **Convergence and error guarantees**: Our work includes a detailed analysis of convergence (Lemma 14) and error guarantees (Corollary 1.1) for ICGD on deep models. These critical theoretical guarantees are absent in [Giannou23].
>
> We share the following similar ideas:
>
> * **Gradient calculation**: Both approaches require providing an explicit formula for gradient computation in feed-forward networks based on the chain rule.
>
> * **Forward pass by the transformer**: Both approaches require the transformer to perform the forward pass of the feed-forward networks with given data samples.
>
> * **Gradient descent by the transformer**: Both approaches involve the transformer approximating the gradient based on the explicit gradient calculation and the outputs from the forward pass. Subsequently, both require the transformer to perform gradient descent.
>
> [Giannou23] Looped Transformers as Programmable Computers, ICML 2023.

---

> ### Author Response · Authors · 2024-11-24
> **Response 4**
>
> >`Q2.` If you are already assuming that various parts of the neural network are approximable by ReLU, and if you are already incurring exponentially growing error terms, why do you not just treat the entire neural network as a $(\epsilon, R, H, C)$-approximable function, and simply use the Transformer to learn ReLU coefficients? Will this lead to worse bounds?
>
> Thank you for your question. Here are some clarifications.
>
> Our aim is to provide an entirely explicit construction for transformers to train $N$-layer networks using ICL. Treating the entire $N$-layer network as an $(\epsilon, R, H, C)$-approximable function has the following problems:
>
> * **Non-explicit construction**: As noted in the response to Weakness 1, the explicitness depends on whether the components approximated by the sum of ReLUs were previously known. Using the sum of ReLUs to approximate the $N$-layer network makes it difficult to derive the parameters for the ReLU approximators, rendering the transformer's construction non-explicit.
>
> * **Insufficient smoothness condition**: Analyzing the smoothness of a fully connected $N$-layer model comprehensively is challenging. Consequently, the model may not meet the smoothness conditions required by Lemma 7. This makes it challenging to apply the same approximation framework to the entire network.
>
> Treating the entire neural network as an $(\epsilon, R, H, C)$-approximable function will not yield a worse bound, while it fails to satisfy the requirements for an explicit transformer construction and imposes overly strong assumptions, making it impractical and ineffective in real-world scenarios.
>
> >`Q3.` The dimension of the Transformer is very large, as each layer needs to contain all the parameters of the $N$-layer neural network. Can the authors please comment on what is the exact dimension of the Transformer needed? I think it is useful to state this value in Theorem 1.
>
> Thank you for your question.
>
> To implement ICGD on a $N$-layer neural network, transformers require $O(NK^2+3NK+3K) = O(NK^2)$ hidden dimensions, where K is the width, and N is the depth of the $N$-layer network.
>
> We have revised our paper and included a new, precise specification of the transformer dimensions required in Theorem 1 (See the proof for details). Please refer to the latest version for detailed information.
>
> >`Suggestion1.` $L$ is used for both loss, number of gradient descent steps, and Lipschitz constant. This gets confusing at times, and I suggest for the authors to use different letters.
>
> Thanks for your suggestion. We apologize for the confusion.
>
> In the revised version, we have modified the notation for the loss function to $\mathcal{L}$, allowing us to use $L$ to denote the gradient descent steps. Now, we use $L$ to denote the gradient descent steps, $\mathcal{L}$ to denote the loss, and $L$ with subscript to denote the Lipschitz constant.
>
> ---
>
> We sincerely appreciate the time and effort that Reviewer hJr1 has invested in reviewing our paper. We have taken all comments into careful consideration and have made corresponding revisions to address the concerns raised.
>
> Please do not hesitate to let us know if there are any other aspects of our work that you would like us to clarify.

---

> > ### Comment · Reviewer_hJr1 · 2024-11-24
> >
> > Thank you for the detailed response. It will take me some time to go over its entirety carefully. In the meanwhile, I have the following question:
> >
> > >  we could derive the coefficients of the sum of ReLUs by solving the optimization problem under the specified constraints and a given (e.g., 1e-5), as defined in Definition 4.
> >
> > > As noted in the response to Weakness 1, the explicitness depends on whether the components approximated by the sum of ReLUs were previously known. Using the sum of ReLUs to approximate the $N$-layer network makes it difficult to derive the parameters for the ReLU approximators, rendering the transformer's construction non-explicit.
> >
> > My issue is that Definition 4 (and the result in Bai et el 2023, for that matter), **are not explicit**. The "solution to an approximation problem" is **not** an explicit construction. If you consider that explicit, then
> > $$sup_{z} |g(z) − f_{H,C}(z)| $$
> > with $g$ being the $N$-layer entire neural network, **would also be considered an explicit construction**.
> >
> > In fact, the approximation power of a two-layer infinitely-wide relu network is also large enough to encompass any smooth function. I can solve for the coefficients of an infinitely-wide two-layer relu network for any target function or operator, wrt the optimization objective of Definition 4, but that does not make it an explicit construction.
> >
> > Is there a fundamental reason why your appeal to Definition 4 is more explicit than the above-mentioned examples?

---

> > > ### Author Response · Authors · 2024-11-24
> > > **Response to the new question**
> > >
> > > Thank you for your timely response. I would like to provide some clarifications regarding the new question.
> > >
> > > The key reason is that the function approximated by the sum of ReLU is relatively simple in our context, such as the Sigmoid activation function. For such simple functions, it is straightforward to derive an explicit construction. Here, we take the Sigmoid activation function as an example and propose another method that avoids optimization. Let $r(z)$ denote the Sigmoid function.
> > >
> > > * **Segment the input domain.** For example, divide the domain $[-10, 10]$ smaller intervals such as $[-10, -9], [-9, -8], \dots, [9, 10]$.
> > >
> > > * **Approximate each segment locally using a linear function via linear interpolation.** For instance, in the domain $[9,10]$, approximate $r(z)$ using a linear function $a_1 z + c_1$, where $a_1$ and $c_1$ are calculated as follows:
> > >
> > >    (i) $a_1 = (r(10)-r(9)) / (10-9)$.
> > >
> > >    (ii) $c_1 = r(9) - a_1 * 9$.
> > >
> > > * **Approximate linear function $a_1 z + c_1 ~(z \in [9,10])$ using a sum of ReLU terms.** This step involves two substeps, which are straightforward to implement:
> > >
> > >    (i) Approximate the indicator function for $z \in [9,10]$ using a sum of ReLU terms.
> > >
> > >    (ii) Approximate the constant $c_1$ using the sum of ReLU. This is because bias terms are not included in the sum of ReLU terms in Definition 4. The bias term $c_1$​ must be approximated using an additional sum of ReLU terms.
> > >
> > > * **Combine all the sum of ReLU approximators across all segments.** Finally, integrate the approximations for all segments to construct the complete approximation.
> > >
> > > To achieve higher precision in the approximation, it is sufficient to use finer segmentations.
> > >
> > > In this way, we avoid the optimization problem, which the above-mentioned examples are unable to achieve.

---

> > > > ### Comment · Reviewer_hJr1 · 2024-11-25
> > > >
> > > > Thank you for the answer. I think you addressed the question of explicit construction satisfactorily.
> > > >
> > > > > We acknowledge that the error between ICGD and true gradient descent could be relatively large. Nonetheless, as demonstrated in Lemma 14, ICGD exhibits convergence and maintains a unique descending tendency, aligning it with the behavior of gradient methods.
> > > >
> > > > From what I gather, the there is
> > > > 1. the blow-up of error due to divergence of non-convex (exact) GD, which is unavoidable
> > > > 2. approximation error of gradient by transformer (is there anything more to this besides the r(t) approximation error?)
> > > > 3. gradient descent discretization error (depends on lipschitz constant, which is exponential in number of layers of target neural network.
> > > >
> > > > Am I missing anything else? In particular, are there other approximation errors in item 2.? Is there any exponential-in-N or exponential-in-d error that is not due to the non-convexity of the target neural network, but due to the approximation error of the Transformer itself? E.g. where does the $d^N$ error of Lemma 6 come from?
> > > >
> > > > I thank the authors for including the 0.5/0.5 experiment; softmax performs reasonably. The new experiments in G.2 are also encouraging. Since time is short, I will not request any more experiments, but it might be useful to do Figure 3 for the harder 0.5/0.5 experiments as well since the range in Figure 3 currently is quite small.
> > > >
> > > > I have no other questions other than the ones above.

---

> > > > > ### Author Response · Authors · 2024-11-25
> > > > > **Response to the new questions**
> > > > >
> > > > > Thank you for your timely response. We are happy to know that we have addressed most of your questions. Here are the clarifications for the additional questions.
> > > > >
> > > > > **There are other approximation errors in item 2.** In item 2, we use the sum of ReLU to approximate activation $r(t)$ (Lemma 2), its derivative $r’(t)$ (Lemma 3), and the derivative of the loss function (Lemma 4), all of which introduce approximation errors.
> > > > >
> > > > > There are **no** errors of the form exponential in N or exponential in d arising from the approximation error of the Transformer itself.
> > > > >
> > > > > The exponential-in-$N$ error in Lemma 6 stems from the **accumulation of approximation errors across all $N$ layers**. Specifically, we approximate $s_i(j)$ using the induction formulation in Equation 3.19, i.e., using the $(j+1)$-th term to derive the $j$-th term ($j \in [N-1]$). Due to the non-convexity of $s_i(j)$, we rely on the triangle inequality and induction to bound the error, which results in an exponential dependence on $N$.
> > > > >
> > > > > Thank you for your suggestion regarding the additional experiments. We will include it in the final version of our work.

---

> > > > > > ### Comment · Reviewer_hJr1 · 2024-11-25
> > > > > >
> > > > > > I believe that my questions have all been satisfactorily addressed. I have increased my score to 6.

---

> > > > > > > ### Author Response · Authors · 2024-11-25
> > > > > > >
> > > > > > > Thank you!
> > > > > > >
> > > > > > > We are glad our revisions and responses meet your expectations.
> > > > > > >
> > > > > > > If there is anything further we can clarify or improve to strengthen your support and encourage a higher score, please let us know, and we will address it promptly.
> > > > > > >
> > > > > > > If not, thank you again for your review and constructive comments. They have been invaluable in improving our work!

---

### Official Review · Reviewer_R2xK · 2024-11-02

**Soundness:** 3
**Presentation:** 2
**Contribution:** 2
**Rating:** 3
**Confidence:** 4

**Summary:**

This paper studies transformer's in-context learning capability. It shows that given an $N$-layer neural networks, there exists a $(2N+4)L$-layer pre-trained transformer that simulates $L$ gradient descent steps. The analysis is also extended to softmax-based transformers. The findings are validated on synthetic datasets.

**Strengths:**

ICL is an important phenomenon, the theoretical explanation of which is still far from complete.

**Weaknesses:**

1. The main contribution of the paper is demonstrating the ability of transformers to simulate N-layer neural networks via gradient descent. However, this has been shown in reference [A]. The submission does not cite [A] and there is no discussion about what's novel compared to [A].

      [A] Wang, Jiang, Li, "In-Context Learning on Function Classes Unveiled for Transformers", ICML 2024.

2. Question 1 asks "Is it possible to train one deep model with another pretrained foundation model?". This is not the same as ICL and the present submission does not answer this question.

3. The organization and presentation requires significant improvement. Instead of focusing on a lot of notational details, the main paper should have more discussions about the significance and implication of the presented results, and the relationship between the presented results and existing results.

**Questions:**

1. What's the true contribution of the paper?

2. Is there any novelty in the proof techniques? If yes, it should be highlighted.

3. What's the connection between the experimental results and the theory? Besides, the R-squared results do not seem very good. How can the experimental results validate the theory?

---

> ### Author Response · Authors · 2024-11-24
> **Response 1**
>
> Thank you for your review.
>
> We have addressed all your comments and questions in this and the following responses.
>
> The draft has been updated accordingly, with changes highlighted in **blue** in the latest revision.
>
> Please refer to the updated PDF for details.
>
> ---
>
> >`W1.` The main contribution of the paper is demonstrating the ability of transformers to simulate N-layer neural networks via gradient descent. However, this has been shown in reference [A]. The submission does not cite [A] and there is no discussion about what's novel compared to [A].
>
> Thank you for bringing this to our attention. We sincerely apologize for our oversight in not acknowledging the important work of [Wang24] in our initial submission.
>
> As a corrective measure, we have thoroughly cited [Wang24] and discussed the novelty of our work in relation to it within the Introduction and Related Work sections of our revised manuscript. Please refer to the latest version for detailed insights.
>
> Compared with [Wang24], our work introduces several novelties:
>
> * **More structured and efficient transformer architecture.** While [Wang24] uses a **$O(N^2L)$-layer** transformer to approximate $L$ gradient descent steps on $N$-layer neural networks, our approach simulates ICGD more efficiently. We approximate specific terms in the gradient expression to reduce computational costs, requiring only a **$(2N + 4)L$-layer** ReLU-based transformer for $L$ gradient descent steps. Our method focuses on selecting and approximating the most impactful intermediate terms in the explicit gradient descent expression (Lemmas 3, 4, 5), optimizing layer complexity to $O(NL)$.
>
> * **Less restrictive input and output dimensions for $N$-layer neural networks.** Unlike [Wang24], which simplifies the output of $N$-layer networks to a scalar, our work expands this by considering cases where output dimensions exceed one, as detailed in Appendix E. This includes scenarios where input and output dimensions differ.
>
> * **More practical transformer model.** [Wang24] discusses activation functions in the attention layer that meet a general decay condition (Definition 2.3 in [Wang24]), without considering the Softmax activation. We extend our analysis to include Softmax-based transformers, reflecting more realistic applications as detailed in Appendix F.
>
> * **More advanced and complicated applications.** [Wang24] discusses the applications to functions, including indicators, linear, and smooth functions. We explore more advanced and complicated scenarios, i.e., the score function in diffusion models discussed in Appendix H. The score function falls outside the smooth function class [Chen23]. This enhancement broadens the applicability of our results.
>
> We appreciate your feedback and have incorporated a detailed discussion of [Wang24] in the revised manuscript. Thank you for helping us improve the clarity and positioning of our work.
>
> [Wang24] In-context Learning on Function Classes Unveiled for Transformers, ICML 2024.
>
> [Chen23] Score approximation, estimation, and distribution recovery of diffusion models on low-dimensional data, ICML 2023.
>
> >`W2.` Question 1 asks "Is it possible to train one deep model with another pretrained foundation model?". This is not the same as ICL and the present submission does not answer this question.
>
> Thank you for your comment. We apologize for any confusion caused by the term "pretrained foundation model." To clarify:
>
> The primary focus of our work is to demonstrate that the in-context learning (ICL) capability of a foundation model can implement gradient descent for an $N$-layer neural network, using a transformer as an example of such foundation models. Our analysis assumes a well-pretrained transformer and provides an explicit characterization of how the ICL of a transformer model can approximate the gradient descent training process for $N$-layer feed-forward neural networks.
>
> We acknowledge that the term "pretrained foundation model" is a little confusing. In response, we have replaced "pretrained foundation model" with "foundation model" in Question 1 and revised Question 1 to **"Is it possible to train one deep model with the ICL of another foundation model?"** We have also made corresponding modifications throughout the revised manuscript.

---

> > ### Comment · Reviewer_R2xK · 2024-11-28
> >
> > The construction in [Wang24] is also structured. The reduction in number of layers is trivial. There are some repeated computations in their construction that are not necessary. They can be easily removed to yield a O(NL) construction.
> >
> > The extension in the output dimension of NN is also trivial.
> >
> > The softmax extension is new. But the key step in the proof seems to be quoting a result from [Kajitsuka and Sato 2024]. Is there any new technique here? This part of the proof is not clearly presented and hard to follow.
> >
> > Regarding the application to learning score functions, the score values are assumed to be known. This is not how score functions are learned in diffusion models. Moreover, the proof consists of two steps., the first of which is to quote a result from [Chen23] that the score function can be approximated by a neural network, while the second uses the result that transformers can learn NN in context, which is shown in [wang24]. So there is very limited contribution here.
> >
> > After the modification, Question 1 is still not the same as ICL and the present submission does not answer the question. Moreover, the statement quoted below sounds like it is a new result, but that's shown in [wang24]. The main text is all about this result, which is not a new contribution.
> >
> > "Specifically, we show that transformer models are capable of simulating the training of a deep ReLU-based feed-forward neural network with provable guarantees through In-Context Learning (ICL)."

---

> > > ### Author Response · Authors · 2024-11-30
> > > **New Response 1**
> > >
> > > Happy holidays, and sorry for the late reply.
> > >
> > > ---
> > >
> > > >`New Q1.` The construction in [Wang24] is also structured. The reduction in number of layers is trivial. There are some repeated computations in their construction that are not necessary. They can be easily removed to yield a O(NL) construction.
> > >
> > > Thank you for your question. We believe that the construction is not trivial. We would appreciate it if the reviewer could provide details on the method used to reduce the layers to $O(NL)$ in [Wang24]. Here are some clarifications.
> > >
> > > * **Different construction strategies**: Although the problem settings appear similar, our approach adopts entirely different construction strategies, such as inductively approximating intermediate chain-rule terms, which ultimately result in a significantly smaller layer requirement. Specifically, we approximate terms in Equation 3.19 (Lemma 5) through induction formulation.
> > >
> > > * **Non-trival layer refuction**: Our improvement is substantial, naturally reducing complexity from $O(N^2L)$ to $O(NL)$. We believe that this reduction is non-trivial. It would be appreciated if the reviewer could provide details on the method used to reduce the layers to $O(NL)$ in [Wang24].
> > >
> > > [Wang24] In-context Learning on Function Classes Unveiled for Transformers, ICML 2024.
> > >
> > > >`New Q2.` The extension in the output dimension of NN is also trivial.The softmax extension is new. But the key step in the proof seems to be quoting a result from [Kajitsuka and Sato 2024]. Is there any new technique here? This part of the proof is not clearly presented and hard to follow.
> > >
> > > Thank you for your question. We would like to clarify and highlight the new techniques we introduced in our work for the Softmax-attention:
> > >
> > >  * **Explicit approximation error bound on broader input domain**: As discussed in `line 2073–2100`, we extend Proposition 1 of [Kajitsuka24] to formulate our Lemma 18. Specifically, we provide an explicit upper bound for the approximation error on a broader supported input domain ($[0, 1]^{d \times n}$ v.s. $[0, B_x]^{d \times n}$), while Proposition 1 of [Kajitsuka24] only provides the error as $\epsilon$. We achieve this by analyzing the approximation error and providing an explicit error bound at each step, rather than merely stating $\epsilon$. For example, we explicitly characterize the error bound in Equation (F.4), whereas they provide only an $\epsilon$ error bound.
> > >
> > > * **Derivation of the point-wise approximation error, distinct from the integration approximation error**: We generalize the integration approximation error to the point-wise approximation error (`line 2374-2429`). This type of approximation analysis is necessary for the ICGD of the transformer (Theorem 6). If we only account for integration approximation error, then we cannot satisfy the pointwise error condition specified in Lemma 6 (e.g. $\|w\|_{max} \leq B_w$).
> > >
> > > We apologize for any lack of clarity in this section and appreciate your feedback. We have enhanced the presentation and coherence of this part of the proof to ensure it is easier to follow in our latest version. We apologize for not being able to update the manuscript.
> > >
> > > [Kajitsuka24] Are transformers with one layer self-attention using low-rank weight matrices universal approximators, ICLR 2024.
> > >
> > > >`New Q3.` Regarding the application to learning score functions, the score values are assumed to be known. This is not how score functions are learned in diffusion models. Moreover, the proof consists of two steps., the first of which is to quote a result from [Chen23] that the score function can be approximated by a neural network, while the second uses the result that transformers can learn NN in context, which is shown in [wang24]. So there is very limited contribution here.
> > >
> > > Thank you for your question. It appears there may have been some misunderstanding. We do not assume that the score values are known.
> > >
> > > Given a training dataset, we train an $N$-layer network to approximate the score function using only the dataset and the ICGD to simulate the training process. We do not need to know the true $N$-layer network approximator for the score function. Therefore, a known score function is not required. Specifically,
> > >
> > > * Given a training dataset, we train an $N$-layer to learn the score function. The results in Lemma 19 and [Chen23] demonstrate the network's ability to approximate the true score function. This provides a theoretical understanding of the approximation and estimation capabilities, without requiring a known score function.
> > >
> > > * Using the ICGD of the transformer on $N$-layer network in our main text, we show that the ICL of the transformer can simulate the gradient descent algorithm using the training dataset in the first item. This process also does not require the true score function.
> > >
> > > [Chen23] Score approximation, estimation, and distribution recovery of diffusion models on low-dimensional data, ICML 2023.

---

> > > ### Author Response · Authors · 2024-11-30
> > > **New Response 2**
> > >
> > > >`New Q4.` After the modification, Question 1 is still not the same as ICL and the present submission does not answer the question. Moreover, the statement quoted below sounds like it is a new result, but that's shown in [wang24]. The main text is all about this result, which is not a new contribution. "Specifically, we show that transformer models are capable of simulating the training of a deep ReLU-based feed-forward neural network with provable guarantees through In-Context Learning (ICL)."
> > >
> > > Thank you for your question. We apologize for any confusion. To clarify, we have revised Question 1 as follows:
> > >
> > > **“Is it possible to simulate the training of one deep model with the ICL of another foundation model?”**
> > >
> > > Here are two important clarifications:
> > >
> > > * **Consistency with ICL mechanism**: This question aims to provide an understanding of the ICL mechanism—specifically, simulating the training of one deep model using in-context examples as the training set, and then performing inference on a testing example with the simulated trained deep model. This approach aligns with a reasonable interpretation of the true ICL.
> > >
> > > * **Solution to Question 1**: We have addressed Question 1 through our research findings. According to our results, the transformer foundation model effectively simulates the training process (i.e., the gradient descent algorithm) of an $N$-layer neural network using in-context examples (Theorem 1). Subsequently, the transformer performs inference on the testing example using the simulated updated $N$-layer network.
> > >
> > > ---
> > >
> > > **Summary of our Main Theoretical Contributions**
> > >
> > > Here, we restate out the main theoretical contributions compared with [Wang24], based on the new feedback.
> > >
> > > * **More structured and efficient transformer architecture.** The problem settings appear similar between our work and [Wang24]. While [Wang24] uses a **$O(N^2L)$-layer** transformer to approximate $L$ gradient descent steps on $N$-layer neural networks, our approach simulates ICGD more efficiently. We approximate specific terms in the gradient expression to reduce computational costs, requiring only a **$(2N + 4)L$-layer** ReLU-based transformer for $L$ gradient descent steps. Our method focuses on selecting and approximating the most impactful intermediate terms in the explicit gradient descent expression (Lemmas 3, 4, 5), optimizing layer complexity to $O(NL)$.
> > >
> > > * **More practical transformer model.** [Wang24] discusses activation functions in the attention layer that meet a general decay condition (Definition 2.3 in [Wang24]), without considering the Softmax activation. We extend our analysis to include Softmax-based transformers, reflecting more realistic applications as detailed in Appendix F. Furthermore, we introduce two additional contributions compared to [Kajitsuka24] to meet our requirements: (i) an explicit approximation error bound for a broader input domain, and (ii) a method to derive the point-wise approximation error, distinct from the integration approximation error.
> > >
> > > [Wang24] In-context Learning on Function Classes Unveiled for Transformers, ICML 2024.
> > >
> > > [Kajitsuka24] Are transformers with one layer self-attention using low-rank weight matrices universal approximators, ICLR 2024.

---

> ### Author Response · Authors · 2024-11-24
> **Response 2**
>
> >`W3.` The organization and presentation requires significant improvement. Instead of focusing on a lot of notational details, the main paper should have more discussions about the significance and implication of the presented results, and the relationship between the presented results and existing results.
>
> Thank you for your suggestion. Here are some clarifications.
>
> We agree that our manuscript's organization could be improved. In response, we have highlighted our main results at the beginning of the Introduction (`line 090-091`) and provided an informal version of these results in Appendix A. We have also added more discussions about the implications of our results (`line 783-791`) and the comparison with existing work [Wang24] (`line 792-816`) in Appendix B.1. Please refer to the revised version for further details.
>
> We believe the details presented are crucial. We have ensured that each lemma (Lemmas 1-6) has been clearly defined to demonstrate their independent significance and to highlight our critical proof techniques: the term-by-term approximation technique and the selection of specific terms. These techniques are crucial for implementing a more structured and efficient transformer architecture. Here is the significance of each lemma:
>
> * **Lemma 1**: Provides an explicit decomposition of one gradient descent step on an $N$-layer network, detailing the specific terms we aim to approximate with a transformer. This lemma is essential for understanding the components that the transformer targets for approximation.
>
> * **Lemma 2**: Demonstrates how transformers can approximate the forward pass of the $N$-layer network, identifying the intermediate terms in the chain rule that need to be approximated in subsequent lemmas. This understanding of the intermediate terms is critical for achieving efficient approximation.
>
> * **Lemmas 3, 4, and 5**: These lemmas detail how we select and inductively approximate specific intermediate terms. They are central to achieving ICGD with only $(2N+4)L$-layer transformers and are crucial for presenting the logic and techniques of our term-by-term approximation strategy.
>
> * **Lemma 6**: Analyzes error accumulation, which is vital for providing theoretical error guarantees. This lemma explains how to manage error accumulation under our term-by-term approximation strategy, offering distinct error bounds for different components. Without this analysis, the validity of our approximation strategy could be questioned.
>
> [Wang24] In-context Learning on Function Classes Unveiled for Transformers, ICML 2024.
>
> >`Q1.` What's the true contribution of the paper?
>
> Thank you for your question.
>
> In response, we have refined our manuscript by re-summarizing our contributions in both the Introduction and Conclusion sections. We have also expanded the discussions about the implications of our results in Appendix A.2, and further compared our work with the existing work [Wang24] in Appendix B.1. Please refer to the revised version for comprehensive details.
>
> We have the following key contributions:
>
> * **Structured and efficient transformer construction** We offer a detailed characterization of the ICL capabilities of a transformer model in approximating the gradient descent training process of an $N$-layer network. Utilizing a term-by-term approximation technique and the selection of specific terms, we demonstrate that only $(2N+4)L$ transformer layers are required (Theorem 1) to implement $L$ gradient descent steps. We also detail the approximation error in Corollary 1.1.
>
> * **Less restrictive input and output dimensions for $N$-layer neural networks.** We extend our analysis to the case where output dimensions exceed one, as detailed in Appendix E. This includes scenarios where input and output dimensions differ.
>
> * **More practical transformer model.** We extend our analysis to include Softmax-based transformers, reflecting more realistic applications as detailed in Appendix F.
>
> * **Experimental validation.** We validate our theory with ReLU- and Softmax-based transformers, specifically, ICGD for the $N$-layer networks. The numerical results in Appendix G show that the performance of ICL matches that of training $N$-layer networks.
>
> * **Advanced and complicated applications.** We explore the application of our results to approximate the score function in diffusion models, as discussed in Appendix H.

---

> ### Author Response · Authors · 2024-11-24
> **Response 3**
>
> >`Q2.` Is there any novelty in the proof techniques? If yes, it should be highlighted.
>
> Thank you for your thoughtful question. Our work introduces the following novel proof techniques:
>
> * **Explicit gradient expression of $N$-layer network**: We derive an explicit formula for gradient computation in $N$-layer feed-forward neural networks, providing a deeper understanding of the gradient structure (Lemma 1).
>
> * **Term-by-term approximation and selection of specific terms**: We apply the term-by-term approximation method and select specific terms in the gradient expression to approximate, aiming to reduce the computational cost. Our method requires only a $(2N + 4)L$-layer ReLU-based transformer to approximate $L$ gradient descent steps. The key technique involves selecting and approximating the most effective intermediate terms in the explicit gradient descent expression (Lemmas 3, 4, 5). This approach helps us optimize the layer complexity to $O(NL)$.
>
> * **Extension to Softmax-based transformer**: We extend our analysis to Softmax-based transformers to better mirror real-world uses, as detailed in Appendix F. The key technique is the universal approximation of Softmax-based transformers (Lemma16).
>
> >`Q3.` What's the connection between the experimental results and the theory? Besides, the R-squared results do not seem very good. How can the experimental results validate the theory?
>
> Thank you for your question. Here are some clarifications.
>
> * In response, we have updated the third contribution in the Introduction (`line 086-089`) to include a more detailed description of our empirical validation. Please refer to the latest revision for more details. We also quote the revised text here:
>
>   **“We validate our theory with ReLU- and Softmax-transformers, specifically, ICGD for the $N$-layer networks (Theorem 1, Theorem 5, and Theorem 6). We assess the ICL capabilities of transformers by training 3-, 4-, and 6-layer networks in Appendix G. The numerical results show that the performance of ICL matches that of training $N$-layer networks.”**
>
> * We apologize for any confusion regarding the experimental results. We train and test the 3-, 4-, and 6-layer networks using data samples from distribution $N(0, I)$. It is appropriate to compare the performance of '$N(-2, I)$' with that of '$N$-layer NN' only when there are 50 or fewer in-context examples in Figures 1 and 2. This comparison helps assess whether the ICL performance is comparable to traditional $N$-layer network training. We also provide additional clarification for other experimental findings:
>
>     (i) For results involving more than 50 in-context examples, we aim to investigate the ICL performance under conditions of extended prompt lengths beyond those used in pertaining.
>
>     (ii) For distributions $0.9N(-2, I) + 0.1N(2, I)$ and $0.7N(-2, I) + 0.3N(2, I)$, our goal is to explore how ICL performance varies as the testing distribution diverges from the pretraining one.
>
> ---
>
> We sincerely appreciate the time and effort that Reviewer R2xK has invested in reviewing our paper. We have taken all comments into careful consideration and have made corresponding revisions to address the concerns raised.
>
> Please do not hesitate to let us know if there are any other aspects of our work that you would like us to clarify.

---

> ### Comment · Reviewer_R2xK · 2024-12-01
>
> Thanks for the response. However, my concerns are not resolved.
>
> 1. The construction in [wang24] is also inductive. When going from n-1 layers to n layers, they recomputed the activation values of previous layers, which is redundant and incurred a complexity of $O(n)$ per layer. If they reuse the activation values from previous layers, which is a very simple modification, the complexity will be $O(1)$ per layer and the total number of layers will be $O(NL)$.
>
> 2. The extension to softmax attention, if correct, could be the true contribution of the paper. The technical challenge of the extension from Proposition 1 of [Kajitsuka24] to Lemma 18, and the new technique required should be clarified and highlighted. Currently this result is only in the appendix, while the main paper is all about a known result. The paper requires a significant reorganization with the true contribution and its importance clarified.
>
> 3. I don't see a misunderstanding regarding the ICL for score function. Problem 3 clearly states that the input data includes $(x_1, y_1), ... , (x_n, y_n)$, where $y_i = \nabla p_{t_i}(x_i)$ are the score values. If the score values are not known, what will be the input?
>
> 4. The revised Question 1 in the response is OK, but the answer is already known. Again, the position and contribution of the paper must be clarified. If a more practical transformer model is what is to be claimed, then the entire paper should be centered around this problem.

---

> ### Author Response · Authors · 2024-12-02
> **Response to Additional Comments (1)**
>
> > The construction in [wang24] is also inductive. When going from n-1 layers to n layers, they recomputed the activation values of previous layers, which is redundant and incurred a complexity of $O(n)$ per layer. If they reuse the activation values from previous layers, which is a very simple modification, the complexity will be $O(1)$ per layer and the total number of layers will be $O(NL)$.
>
> Thank you for your feedback. We acknowledge the extension to $O(NL)$ layers in [Wang24]. However, with utmost respect, we would like to clarify that **the similarity between the two works lies only in the problem definition (i.e., ICGD N-layer NN) and the surface-level results (i.e., demonstrating that a transformer can approximate an ICGD N-layer NN).** Both the techniques and the results are fundamentally different.
>
> **Technique-wise,** our approach employs entirely distinct methods to achieve the $(2N+4)L$-layer result:
>
> - We utilize unique decomposition methods for the gradients.
> - We select different terms for approximation.
> - We design a novel transformer structure to construct the required model.
>
> As such, **our findings are not derived or modified from the proofs, methods or results in [Wang24].** Instead, we propose a fundamentally new approach that explicitly constructs a $(2N+4)L$-layer transformer using unique structural and approximation techniques.
>
> **Result-wise,** Our method requires only $(2N+4)L$ layers to approximate $L$ gradient steps. In contrast, [Wang24] involves $O(N^2L)$ complexity, which is quadratic in model depth. **Our result, which is linear in both model depth $N$ and sequence length $L$, is more general and more practical.**  Specifically, the $O(NL)$ layers requirement of our method is significantly more feasible for real-world applications, whereas the $O(N^2L)$ layers requirement in [Wang24] poses unrealistic challenges. Moreover, **our extension to Softmax-based transformers serves as a valuable complementary to [Wang24].**
>
> Given these points, we believe our contributions go beyond incremental extensions, presenting a fair step forward in the field. Our novel techniques and significantly improved results highlight the originality and value of our approach.
>
> [Wang24] In-context Learning on Function Classes Unveiled for Transformers, ICML 2024.
>
> > The extension to softmax attention, if correct, could be the true contribution of the paper. But currently this result is only in the appendix, while the main paper is all about a known result. The paper requires a significant reorganization with the true contribution and its importance clarified.
>
> Thank you for your suggestions. While we respectfully disagree with the characterization that our “true” contribution is solely the extension to softmax attention, or that the main paper focuses entirely on a known result, we appreciate your feedback on the paper's organization. We take your comments seriously and will work to improve the paper organization and clarity of the paper to better highlight our contributions with respect to the Softmax-attention. We will update the final version accordingly.
>
> > I don't see a misunderstanding regarding the ICL for score function. Problem 3 clearly states that the input data $D_n = {(x_i, y_i)}{i\in [n]}$, where $y_i = \nabla p{t_i}(x_i)$ are the score values. If the score values are not known, what will be the input?
>
> Thank you for your feedback. **You are correct; there was a typo in the data generation method, and we have updated the method in the latest version**, which does not involve a known score function. We apologize for the error. Here are some clarifications:
>
> 1. The method for generating the input data was incorrect in our original version.
> 2. We have now updated it as follows: Given $x_i$, we apply the standard method in the diffusion model to add noise to $x_i$, resulting in $x_{i, t}$ at timestamp $t$.
> 3. The expected output is calculated using the formula: $y_i = -(x_{i, t} - \alpha_t * x_i) / h_t$, where $\alpha_t = e^{-0.5t}$, and $h_t = 1-e^{-t}$.
>
> Please refer to the "Score Matching" part in "Section 2: Preliminaries" of [Chen23] for more details. This method is widely used in practice for score matching without a known score function.
>
> Thank you again for your attention to details. This is really helpful.
>
> [Chen23] Score approximation, estimation, and distribution recovery of diffusion models on low-dimensional data, ICML 2023.

---

> ### Author Response · Authors · 2024-12-02
> **Response to Additional Comments (2)**
>
> > The revised Question 1 in the response is OK, but the answer is already known. Again, the position and contribution of the paper must be clarified. If a more practical transformer model is what is to be claimed, then the entire paper should be centered around this problem.
>
> Thank you for your suggestions. Our main contributions are as follows:
>
> 1. We propose a more practical and effective constructed transformer capable of implementing $L$-step gradient descent with $O(NL)$ layers, providing an affirmative example to Question 1.
> 2. We extend our results to transformers with softmax attention.
> 3. We validate our theoretical findings for both contributions 1 and 2 through experimental evaluation.
>
> Again, we emphasize that **our findings are not derived or modified from the proofs, methods, or results in [Wang24]**. Instead, we propose a fundamentally new approach that explicitly constructs a $(2N+4)L$-layer transformer using unique structural and approximation techniques.
>
> In summary,
> - contributions 1 and 2 offer two theoretical approaches to addressing Question 1: one for ReLU-based transformers and the other for softmax-based transformers.
> - Contribution 3 complements these by providing experimental validation for the theories introduced in contributions 1 and 2.
>
> Given these points, we believe our position and contributions are fair. That said, we agree that we should place greater emphasis on the technical novelties of our second contribution (softmax ICL). We take your comments seriously and will work to improve the paper's structure and clarity. We will update the final version accordingly.
>
>
> ---
>
> Thank you for your willingness and persistence in helping us further polish this work.
>
> In the above rebuttal, we have made every effort to address all your concerns and additional questions.
>
> If you feel your concerns have been adequately addressed, with utmost respect, we invite you to consider a score adjustment. If not, we hope to make the most of the remaining two days to provide further clarifications.
>
> Thank you for your time and consideration!
>
> Best regards,
>
> Authors

---

### Official Review · Reviewer_99cj · 2024-11-03

**Soundness:** 2
**Presentation:** 2
**Contribution:** 2
**Rating:** 6
**Confidence:** 2

**Summary:**

This paper investigates the theoretical underpinnings of transformers as deep optimizers, specifically focusing on their capacity for in-context learning (ICL) to simulate gradient descent steps. The authors provide a construction of a transformer capable of performing gradient-based optimization implicitly without updating its own parameters. Through a series of lemmas and Theorem 1, the paper establishes that a pretrained transformer can approximate gradient descent steps of deep neural networks with convergence guarantees.

**Strengths:**

The general topic of theoretical understanding of transformers and ICL is interesting and novel.

**Weaknesses:**

1. The organization could be improved. The main result, Theorem 1, is only presented at the end of the paper. The purpose of preceding Lemmas 1-6 is unclear; it would help to clarify whether they serve solely as components of Theorem 1’s proof or have independent significance.

2. The studied problem lacks sufficient motivation. Transformers are known to have a complicated structure and may contain many parameters. Therefore it is not surprising that a transformer can have enough expressiveness to approximate the gradient descent steps.

3. There is no overview of the empirical validation in the main body of the paper. At least, the authors should use a few sentences to indicate which part of the conclusion is validated by empirical results.

4. The title says "pretrained transformers are deep optimizers", while no pretraining process is analyzed in the main body of the paper.

**Questions:**

See Weaknesses

---

> ### Author Response · Authors · 2024-11-24
> **Response 1**
>
> Thank you for your review.
>
> We have addressed all your comments and questions in this and the following responses.
>
> The draft has been updated accordingly, with changes highlighted in **blue** in the latest revision.
>
> Please refer to the updated PDF for details.
>
> ---
>
> >`W1.` The organization could be improved. The main result, Theorem 1, is only presented at the end of the paper. The purpose of preceding Lemmas 1-6 is unclear; it would help to clarify whether they serve solely as components of Theorem 1’s proof or have independent significance.
>
> Thank you for your suggestion. Here are some clarifications.
>
> We agree that the organization of our manuscript could be improved. In response, we have highlighted our main results at the beginning of the Introduction (`line 089-090`) and provided an informal version of these results in Appendix A. Please refer to the revised version for further details.
>
> Additionally, we have ensured that each lemma (Lemmas 1-6) has been clearly defined to demonstrate their independent significance and to highlight our critical proof techniques: the term-by-term approximation technique and the selection of specific terms. These techniques are crucial for implementing a more structured and efficient transformer architecture. Here is the significance of each lemma:
>
> * **Lemma 1**: Provides an explicit decomposition of one gradient descent step on an $N$-layer network, detailing the specific terms we aim to approximate with a transformer. This lemma is essential for understanding the components that the transformer targets for approximation.
>
> * **Lemma 2**: Demonstrates how transformers can approximate the forward pass of the $N$-layer network, identifying the intermediate terms in the chain rule that need to be approximated in subsequent lemmas. This understanding of the intermediate terms is critical for achieving efficient approximation.
>
> * **Lemmas 3, 4, and 5**: These lemmas detail how we select and inductively approximate specific intermediate terms. They are central to achieving ICGD with only $(2N+4)L$-layer transformers and are crucial for presenting the logic and techniques of our term-by-term approximation strategy.
>
> * **Lemma 6**: Analyzes error accumulation, which is vital for providing theoretical error guarantees. This lemma explains how to manage error accumulation under our term-by-term approximation strategy, offering distinct error bounds for different components. Without this analysis, the validity of our approximation strategy could be questioned.

---

> ### Author Response · Authors · 2024-11-24
> **Response 2**
>
> >`W2.` The studied problem lacks sufficient motivation. Transformers are known to have a complicated structure and may contain many parameters. Therefore it is not surprising that a transformer can have enough expressiveness to approximate the gradient descent steps.
>
> Thank you for your feedback. Here are some clarifications.
>
> We acknowledge the intuitive belief that a transformer possesses sufficient expressiveness to approximate gradient descent steps. However, there is a gap between theoretical understanding [Bai23, Oswald23, Ahn23, and Mahankali23] and practical application. Our primary motivation is to bridge this gap by providing a theoretical framework that aligns with practical observations in two key ways:
>
> * **ICGD for deep model ($N$-layer neural network).** The capability of ICL in the transformer to approximate the gradient descent algorithm in $N$-layer networks is crucial. Previous works [Bai23, Oswald23, Ahn23, and Mahankali23] only provide approximation capabilities for simpler algorithms, such as linear regression and $2$-layer networks. These simpler algorithms are inadequate for practical applications. We give the following 2 examples: (i) Approximating the diffusion score function requires neural networks with multiple layers [Chen23]. (ii) Approximating the indicator function requires at least $3$-layer networks [Safran17]. This requires the explicit construction of transformers to implement in-context gradient descent (ICGD) on deep models, which better aligns with real-world in-context settings.
>
> * **ICGD of softmax-based transformers.** Previous works [Bai23] focus only on the ICGD of ReLU-based transformers, which does not reflect practical applications. We extend our analysis to Softmax-based transformers to better mirror real-world uses, as detailed in Appendix F.
>
> In summary, although it is intuitively understood that a transformer has enough expressiveness to approximate gradient descent steps, our goal is to provide a theoretical framework that better aligns with practical implementations.
>
> [Bai23] Transformers as statisticians: provable in-context learning with in-context algorithm selection, NeurIPS 2023.
>
> [Chen23] Score approximation, estimation, and distribution recovery of diffusion models on low-dimensional data, ICML 2023.
>
> [Oswald23] Transformers learn in-context by gradient descent, ICML 2023.
>
> [Ahn23] Transformers learn to implement preconditioned gradient descent for in-context learning, NeurIPS 2023.
>
> [Mahankali23] One step of gradient descent is provably the optimal in-context learner with one layer of linear self-attention, CoRR 2023.
>
> [Safran17] Depth-Width Tradeoffs in Approximating Natural Functions with Neural Networks, ICML 2017.
>
> >`W3.` There is no overview of the empirical validation in the main body of the paper. At least, the authors should use a few sentences to indicate which part of the conclusion is validated by empirical results.
>
> Thank you for pointing this out.
>
> In response, we have updated the third contribution in the Introduction (`line 085-088`) to include a more detailed description of our empirical validation. Please refer to the latest revision for more details. We also quote the revised text here:
>
> “We validate our theory with ReLU- and Softmax-transformers, specifically, ICGD for the $N$-layer networks (Theorem 1, Theorem 5, and Theorem 6). We assess the ICL capabilities of transformers by training 3-, 4-, and 6-layer networks in Appendix G. The numerical results show that the performance of ICL matches that of training $N$-layer networks.”
>
> >`W4.` The title says "pretrained transformers are deep optimizers", while no pretraining process is analyzed in the main body of the paper.
>
> Thanks for your comment. Here are some clarifications.
>
> We apologize for any confusion caused. The main point of our contribution is based on well-pretrained transformers, which we assume. Based on this assumption, we provide an explicit characterization of the ICL capabilities of a transformer model in approximating the gradient descent training process of $N$-layer feed-forward neural networks.
>
> We agree that the term "Pretrained Transformer" is a bit confusing. In response, **we have replaced 'Pretrained Transformers' with 'Transformers'** in the title and have made corresponding modifications throughout the revised manuscript.
>
> ---
>
> We sincerely appreciate the time and effort that Reviewer 99cj has invested in reviewing our paper. We have taken all comments into careful consideration and have made corresponding revisions to address the concerns raised.
>
> Please do not hesitate to let us know if there are any other aspects of our work that you would like us to clarify.

---

> > ### Comment · Reviewer_99cj · 2024-11-26
> >
> > Thank you for your response. Some of my concerns have been addressed, so I raise my score accordingly.

---

> > > ### Author Response · Authors · 2024-11-26
> > >
> > > Thank you!
> > >
> > > We are glad our revisions and responses meet your expectations.
> > >
> > > If there is anything further we can clarify or improve to strengthen your support and encourage a higher score, please let us know, and we will address it promptly.
> > >
> > > If not, thank you again for your review and constructive comments. They have been invaluable in improving our work!

---

### Official Review · Reviewer_tzww · 2024-11-04

**Soundness:** 3
**Presentation:** 3
**Contribution:** 2
**Rating:** 6
**Confidence:** 3

**Summary:**

This paper provides construction and proof on a ReLU-based Transformer are able to perform gradient descent steps for a N-layer fully connected neural network.

**Strengths:**

This paper presents clear and rigorous proof in showing Transformer can implement gradient descent for (x, y) ~ N-layer neural network.

**Weaknesses:**

1. The paper frequently uses the term "pretrained Transformer." Typically, this refers to a Transformer model trained on a large corpus for next-token prediction. However, in this paper, "pretrained Transformer" actually refers to a Transformer with specific weight constructions. I could not find an explanation of a "pretraining task" that leads to these constructed weights. This terminology may be confusing, especially in the introduction, where the authors emphasize that "one trained foundation model could lead to many others without resource-intensive pretraining".

2. The paper also discusses using the Transformer to estimate the score function for diffusion, which is presented as a major contribution. However, this is accomplished by reducing the score function to a fully-connected neural network. It might be helpful to adjust the emphasis on the diffusion score function and clarify that the main contribution is to provide theoretical support for the Transformer's capability to implement gradient descent for an N-layer neural network.

3. While the idea of demonstrating that a Transformer can implement an N-layer neural network is interesting, the paper only establishes the existence of such a network without addressing optimality or uniqueness. This makes the result slightly less compelling to the community, especially as the construction serves primarily as a proof of concept of the (modified) Transformer's capability rather than a more extensive finding. I understand that this limitation arises from the nature of the problem itself, and it may only be addressed through further results that demonstrate additional insights or practical applications.

**Questions:**

I found the necessity of the element-wise multiplication layer intriguing, particularly as discussed in Remark 3. Could the Transformer architecture circumvent the need for this element-wise multiplication by increasing its depth and width? In other words, is it possible to achieve the same functional capabilities without the element-wise multiplication layer by scaling the model’s size?

---
Updated after the discussion:
Thank you to the authors for taking the time to address my concerns. However, I still find the motivation behind training an N-layer neural network via in-context learning in another model questionable. Beyond the theoretical FLOPs and actual computational requirements, there is also the storage trade-off to consider. Unfortunately, I don’t see the benefits of replacing the direct training of an N-layer neural network with the forward pass of the construction model.

I am content with the story being about proposing a construction method to implement gradient descent for an N-layer neural network. Therefore, I will keep my rating at 6.

Reason for not a higher score: This work provides proof-of-concept results regarding the Transformer’s capacity to implement gradient descent for an N-layer neural network. However, these results lack practical insights for improving real-world applications, making this primarily a proof-of-concept contribution.

Reason for not a lower score: This work contributes to the literature on understanding the Transformer’s capacity. I appreciate the theoretical merit embedded in this study.

---

> ### Author Response · Authors · 2024-11-24
> **Response 1**
>
> Thank you for your review.
>
> We have addressed all your comments and questions in this and the following responses.
>
> The draft has been updated accordingly, with changes highlighted in **blue** in the latest revision.
>
> Please refer to the updated PDF for details.
>
> ---
>
> > `W1.` The paper frequently uses the term "pretrained Transformer." Typically, this refers to a Transformer model trained on a large corpus for next-token prediction. However, in this paper, "pretrained Transformer" actually refers to a Transformer with specific weight constructions. I could not find an explanation of a "pretraining task" that leads to these constructed weights. This terminology may be confusing, especially in the introduction, where the authors emphasize that "one trained foundation model could lead to many others without resource-intensive pertaining".
>
> Thanks for your comment. Here are some clarifications.
>
> We apologize for any confusion caused. The main point of our contribution is based on well-pretrained transformers, which we assume. Based on this assumption, we provide an explicit characterization of the ICL capabilities of a transformer model in approximating the gradient descent training process of $N$-layer feed-forward neural networks.
>
> We agree that the term "Pretrained Transformer" is a bit confusing. In response, **we have replaced 'Pretrained Transformers' with 'Transformers'** in the title and have made corresponding modifications throughout the revised manuscript.
>
> >`W2.` The paper also discusses using the Transformer to estimate the score function for diffusion, which is presented as a major contribution. However, this is accomplished by reducing the score function to a fully-connected neural network. It might be helpful to adjust the emphasis on the diffusion score function and clarify that the main contribution is to provide theoretical support for the Transformer's capability to implement gradient descent for an N-layer neural network.
>
> Thank you for your suggestion.
>
> We agree that our main contribution is to provide an explicit characterization of the ICL capabilities of a transformer model in approximating the gradient descent training process of $N$-layer feed-forward neural networks. Using ICL in transformers for diffusion score estimation is just one application of our results, which we mention briefly and discuss in Appendix H. We have made corresponding modifications throughout the revised manuscript, especially in the introduction section.

---

> ### Author Response · Authors · 2024-11-24
> **Response 2**
>
> >`W3.` While the idea of demonstrating that a Transformer can implement an N-layer neural network is interesting, the paper only establishes the existence of such a network without addressing optimality or uniqueness. This makes the result slightly less compelling to the community, especially as the construction serves primarily as a proof of concept of the (modified) Transformer's capability rather than a more extensive finding. I understand that this limitation arises from the nature of the problem itself, and it may only be addressed through further results that demonstrate additional insights or practical applications.
>
> Thank you for your feedback. Here are some clarifications.
>
> We agree that our work establishes the existence of Transformers capable of approximating the gradient descent algorithm of $N$-layer networks without addressing optimality or uniqueness. The theoretical results guarantee approximation within any given error and convergence of the ICL gradient descent.
> However, our contributions are significant for the following three reasons:
>
> * **Align with real-world in-context settings better.** However, the capability to approximate the gradient descent algorithm in $N$-layer networks is crucial. Previous works [Bai23, Oswald23, Ahn23, and Mahankali23] only provide approximation capabilities for simpler algorithms, such as linear regression and $2$-layer networks. These simpler algorithms are inadequate for practical applications. We give the following 2 examples: (i) Approximating the diffusion score function requires neural networks with multiple layers [Chen23]. (ii) Approximating the indicator function requires at least $3$-layer networks [Safran17]. This requires the explicit construction of transformers to implement in-context gradient descent (ICGD) on deep models, which better aligns with real-world in-context settings.
>
> * **ICGD of softmax-based transformers.** Previous works [Bai23] focus only on the ICGD of ReLU-based transformers, which does not reflect practical applications. We extend our analysis to Softmax-based transformers to better mirror real-world uses, as detailed in Appendix F. The key technique is the universal approximation of Softmax-based transformers (Lemma16).
>
> * **Important proof techniques.** For the results of ICGD in the ReLU-based transformer, we employ the following essential techniques, which are not trivial:
>
>   (1). **Explicit gradient expression.**  We derive an explicit formula for gradient computation in $N$-layer feed-forward neural networks, providing a deeper understanding of the gradient structure (Lemma 1).
>
>   (2). **Efficient approximation strategy.** We apply the term-by-term approximation method and select specific terms in the gradient expression to approximate, aiming to reduce the computational cost. Our method requires only a $(2N + 4)L$-layer ReLU-based transformer to approximate $L$ steps of gradient descent. The key technique involves selecting and approximating the most effective intermediate terms in the explicit gradient descent expression (Lemmas 3, 4, and 5). This approach helps us optimize the layer complexity to $O(NL)$.
>
> In summary, while our work does not establish optimality or uniqueness, our contributions nevertheless offer valuable insights to the community.
>
> [Bai23] Transformers as statisticians: provable in-context learning with in-context algorithm selection, NeurIPS 2023.
>
> [Chen23] Score approximation, estimation, and distribution recovery of diffusion models on low-dimensional data, ICML 2023.
>
> [Oswald23] Transformers learn in-context by gradient descent, ICML 2023.
>
> [Ahn23] Transformers learn to implement preconditioned gradient descent for in-context learning, NeurIPS 2023.
>
> [Mahankali23] One step of gradient descent is provably the optimal in-context learner with one layer of linear self-attention, CoRR 2023.
>
> [Safran17] Depth-Width Tradeoffs in Approximating Natural Functions with Neural Networks, ICML 2017.

---

> ### Author Response · Authors · 2024-11-24
> **Response 3**
>
> >`Q1.` I found the necessity of the element-wise multiplication layer intriguing, particularly as discussed in Remark 3. Could the Transformer architecture circumvent the need for this element-wise multiplication by increasing its depth and width? In other words, is it possible to achieve the same functional capabilities without the element-wise multiplication layer by scaling the model’s size?
>
> Thank you for the insightful question. The answer is **Yes**.
>
> Scaling the model’s size can provide similar functional capabilities. This is achieved by using a sum of ReLUs (Definition 4) to approximate element-wise multiplication functions. This method complicates the analysis of the Transformer's depth, width, and the upper bounds of its parameter matrices.
> Furthermore, in Appendix F, we present the ICGD results for softmax-based transformers, employing the universal approximation theorem for transformers. This section focuses solely on scaling the model’s size without incorporating element-wise multiplication layers.
>
> ---
>
> We sincerely appreciate the time and effort that Reviewer tzww has invested in reviewing our paper. We have taken all comments into careful consideration and have made corresponding revisions to address the concerns raised.
>
> Please do not hesitate to let us know if there are any other aspects of our work that you would like us to clarify.

---

> > ### Comment · Reviewer_tzww · 2024-11-25
> >
> > In your response, you mentioned, “The main point of our contribution is based on well-pretrained transformers, which we assume.” Could you clarify what pretraining task you are referring to here? Are you referencing the pretraining tasks discussed in the appendix experiment section? If so, I note that those tasks are the same set of tasks used during inference. This approach differs from traditional Transformer pretraining, where models are trained on distributions entirely different from those encountered during inference, allowing for more generalizable capabilities. Some additional clarification on this point would be helpful.
> >
> > After reviewing the revised introduction, I still find the claim—“one foundation model could lead to many others without resource-intensive pretraining, making foundation models more accessible to general users”—to be somewhat overstated. While the idea is compelling, it does not appear to account for the forward computational cost of such a foundation model. Providing an estimate of FLOPs or another relevant metric to substantiate this claim would greatly enhance its credibility.
> >
> > Moreover, as far as I understand, the proposed foundation model appears limited in scope, as it can only facilitate gradient descent for a specific deep model. This constraint seems to reduce its generalization capabilities. I would recommend softening this claim in the introduction to reflect these limitations more accurately.
> >
> > Lastly, could you elaborate on the differences between your approach and the work presented in “Trainable Transformer in Transformer” (Panigrahi et al., 2023)? That paper demonstrates the gradient simulation of a Transformer within another Transformer, which is arguably more advanced compared to simulating gradient descent in an N-layer Neural Network. A discussion highlighting the distinctions and potential advantages of your approach compared to TINT would provide valuable context for readers.
> >
> > > Panigrahi, Abhishek, et al. "Trainable transformer in transformer." arXiv preprint arXiv:2307.01189 (2023).

---

> > > ### Comment · Reviewer_tzww · 2024-11-26
> > >
> > > Hi, I wonder if the authors are planning to address my comment. The TinT paper is nowhere discussed in the paper, along with the side that there are overclaims in the introduction. It would be great if the authors can address my concern. Thanks!

---

> > > ### Author Response · Authors · 2024-11-27
> > > **New Response 3**
> > >
> > > >`New Q3.` Moreover, as far as I understand, the proposed foundation model appears limited in scope, as it can only facilitate gradient descent for a specific deep model. This constraint seems to reduce its generalization capabilities. I would recommend softening this claim in the introduction to reflect these limitations more accurately.
> > >
> > > Thank you for your suggestion. It is very helpful. We have modified the paper accordingly, especially the introduction section. Please review the revised version for the details.
> > >
> > > >`New Q4.` Lastly, could you elaborate on the differences between your approach and the work presented in “Trainable Transformer in Transformer” (Panigrahi et al., 2023)? That paper demonstrates the gradient simulation of a Transformer within another Transformer, which is arguably more advanced compared to simulating gradient descent in an N-layer Neural Network. A discussion highlighting the distinctions and potential advantages of your approach compared to TINT would provide valuable context for readers.
> > >
> > > Thanks for your insightful comments. We have added a comparison in the “Related Works” section (`line 817-841`). We restate it as follows:
> > >
> > > The main distinction between our work and [Panigrahi23] lies in the **different aims**: Our approach focuses on using a **standard transformer for the simulator** (with a minor modification: the “Element-wise Multiplication Layer”), and we provide a theoretical understanding of how a standard transformer can learn the ICGD of an $N$-layer network using ICL. In contrast, [Panigrahi23] aims to **build even stronger transformers by introducing several structural modifications** that enable running gradient descent on auxiliary transformers.
> > >
> > > While [Panigrahi23] demonstrates in-context gradient descent for a more advanced model, i.e., one transformer, our work offers the following potential advantages:
> > >
> > > * **Explicit transformer construction**: We provide an explicit construction of the transformer, whereas [Panigrahi23] does not detail the explicit construction of model parameters within their transformer.
> > >
> > > * **Exact gradient descent**: We compute the exact and explicit gradient descent for an $N$-layer network (Lemma 1). Building on this, we employ the transformer's ICL to perform gradient descent on all parameters. Conversely, [Panigrahi23] stops the gradient computation through attention scores in the self-attention layer and only updates the value parameter in the self-attention module. Additionally, [Panigrahi23] uses Taylor expansion to approximate the gradient.
> > >
> > > * **Rigorous error and convergence guarantees**: We provide rigorous gradient descent approximation errors (for multiple steps) and convergence guarantees for the ICGD on an $N$-layer network (Corollary 1.1 and Lemma 14). However, [Panigrahi23] only presents the gradient approximation error for each specific part of the parameters in a single step.
> > >
> > > * **Attention layer better aligned with practice**: Our analysis is based on ReLU-attention (Theorem 1) or Softmax-attention (Theorem 6), whereas [Panigrahi23] utilizes linear attention. Our choice of attention layer better aligns with practical applications.
> > >
> > > [Panigrahi23] Trainable Transformer in Transformer, arXiv 2023.

---

> ### Author Response · Authors · 2024-11-26
>
> Yes, we apologize for the delayed response. We are working on adding new experiments and will provide a more detailed reply within 8 hours.
>
> Thank you for your understanding.

---

> > ### Comment · Reviewer_tzww · 2024-11-26
> >
> > Thank you! Appreciated! Please take your time as there are one more week to discuss. I am just wondering because there are ongoing conversations. Sorry for being pushy, and I really appreciate your time and effort.

---

> > > ### Author Response · Authors · 2024-11-26
> > >
> > > Thank you for your time and understanding! We will try our best to respond as soon as possible.

---

> ### Author Response · Authors · 2024-11-27
> **New Response 1**
>
> We apologize for the late reply. We provide the new response as follows.
>
> ---
>
> >`New Q1.` In your response, you mentioned, “The main point of our contribution is based on well-pretrained transformers, which we assume.” Could you clarify what pretraining task you are referring to here? Are you referencing the pretraining tasks discussed in the appendix experiment section? If so, I note that those tasks are the same set of tasks used during inference. This approach differs from traditional Transformer pretraining, where models are trained on distributions entirely different from those encountered during inference, allowing for more generalizable capabilities. Some additional clarification on this point would be helpful.
>
> Thank you for your insightful question. We apologize for any confusion. We have modified the introduction (`line 036-040`) and limitation (`line 861-872`) parts accordingly. Here are some clarifications:
>
> We acknowledge that there are limitations in the generalization capabilities compared with traditional transformer pretraining. In our setting, the pretraining task refers to using in-context examples generated by an $N$-layer network for a given $N$. Specifically, during pretraining, the distribution of the $N$-layer network parameters is predetermined (e.g., $N(0, I)$). The input data distribution of $N$-layer network for generating the in-context examples is also predetermined (e.g., $N(-2, I)$). The generalization capabilities include the following two aspects:
>
> * Varying the input data distribution for the $N$-layer network to generate the in-context examples. For example, we can change the input data distribution from $N(-2, I)$ to $0.9N(-2, I) + 0.1N(2, I)$ during the inference.
>
> * Varying the distribution of the $N$-layer network parameters. For example, we change the distribution from $N(0, I)$ to $N(0.5, I)$. We have added additional experiments about this in Appendix G.2.
>
> The above points lead to differences between the distributions of in-context examples during pretraining and inference. One limitation is that the in-context examples are generated by the $N$-layer network with the same hyperparameters, including the network width and depth.

---

> ### Author Response · Authors · 2024-11-27
> **New Response 2**
>
> > `New Q2.` After reviewing the revised introduction, I still find the claim—“one foundation model could lead to many others without resource-intensive pretraining, making foundation models more accessible to general users”—to be somewhat overstated. While the idea is compelling, it does not appear to account for the forward computational cost of such a foundation model. Providing an estimate of FLOPs or another relevant metric to substantiate this claim would greatly enhance its credibility.
>
> Thank you for your insightful question. We have acknowledged this limitation and added it to the limitation part (`line 873-882`). We give the FLOPs estimation in theory as follows:
>
> We estimate the forward FLOPs as $2N_p N_t$ and the backward FLOPs as $4N_p N_t$ [Hoffmann22], where $N_p$ represents the number of model parameters and $N_t$ the number of processed tokens.
>
> * **FLOPs with ICL**: In the forward pass of the transformer, the number of input tokens is $n+1$, and the number of transformer parameters is $O(LN^3K^5 / \epsilon^2)$, where $\epsilon$ is the approximation error in the sum of ReLU. Consequently, the FLOPs for in-context learning of the transformer are $O(nLN^3K^5 / \epsilon^2)$
>
> * **FLOPs when training the $N$-layer network directly**: For the direct training process of the $N$-layer network, we denote the number of training data samples as $n$. In the ICL of the transformer, we consider $L$ gradient descent steps, each calculating the average loss across all in-context examples. Therefore, we set the number of training epochs for the $N$-layer network to $L$ to ensure a fair comparison. The number of input tokens for each sample in the $N$-layer network is considered to be 1, and the parameter count for the $N$-layer network is $O(NK^2)$, where $K$ represents the model width. Consequently, the FLOPs for training the $N$-layer network are $O(nLNK^2)$. Compared with training, the inference FLOPs are negligible. Thus, the FLOPs without ICL are $O(nLNK^2)$.
>
> Overall, we find that the FLOPs required for ICL exceed those needed when training the $N$-layer network directly. This encourages us for more efficient construction. Here, we give **two additional clarifications based on the practical experiments**:
>
> * **FLOPs comparison in Appendix G**: In Appendix G, the experimental results show that the performance of ICL in the transformer matches that of training $N$-layer networks directly. We present the following FLOPs comparison under this setting, which shows that the transformer with ICL can consume fewer FLOPs to match the performance of a directly trained 6-layer network in practice:
>
>   (i) **FLOPs with ICL**: In the forward pass of the transformer, the number of input tokens is 76, and the number of transformer parameters is 22 million. Consequently, the FLOPs for in-context learning of the transformer are $2 \times 76 \times 22$ million = 3.34 billion.
>
>   (ii) **FLOPs when training the $6$-layer network directly**: For the training process of the $6$-layer network, we denote the number of training data samples as $n=75$. We use 100 training epochs in Appendix G. The number of input tokens for each sample in the $6$-layer network is considered to be 1, and the parameter count for the $6$-layer network is 168,000. Consequently, the FLOPs for training the $6$-layer network are $6 \times 75 \times 100 \times 168,000 = 7.56$ billion. Compared with training, the inference FLOPs are negligible. Thus, the FLOPs without ICL total 7.56 billion.
>
> * **Time comparison in Appendix G**: We also provide specific times for the forward pass and training processes used in Appendix G.1. We consider 75 in-context examples here. One forward pass takes 1.38 seconds on one A100 (80GB), whereas training a $6$-layer network takes 0.04 seconds for one optimization step and approximately 4 seconds for 100 steps.
>
> In summary, although our construction requires more FLOPs than directly training an $N$-layer network, the experimental results in Appendix G demonstrate that the transformer with in-context learning (ICL) can consume fewer FLOPs to match the performance of a directly trained 6-layer network in practice.
>
> [Hoffmann22] Training Compute-Optimal Large Language Models, 2022.

---

> ### Author Response · Authors · 2024-12-02
> **A Gentle Reminder**
>
> Dear Reviewer tzww,
>
> As the rebuttal phase nears its end and our discussion seems to have paused midway, we would be delighted to continue the conversation.
>
> We have made every effort to address all your concerns and additional questions in our responses.
>
> We would greatly appreciate any further input. If you feel your concerns have been adequately addressed, with utmost respect, we invite you to consider a score adjustment. If not, we hope to make the most of the remaining two days to provide further clarifications.
>
> Thank you for your time and consideration!
>
> Best,
>
> Authors

---

### Author Response · Authors · 2024-11-24
**Revision Summary**

Dear Reviewers,

Thank you for your insightful feedback and the time invested in reviewing our work. We have addressed your inquiries and concerns in our rebuttal and the updated manuscript.

Following your suggestions, we have enhanced the readability of the paper through extensive proofreading, detailed polishing, and correction of typos. Specifically, we have highlighted the main results in the Introduction (`line 089–090`) and provided a more informal version in Appendix A for clarity. We have also added a more detailed description of the experimental part in the Introduction (`line 085-088`). Additionally, we have clarified the independent significance of Lemmas 1–6 to ensure their purposes are well understood. Furthermore, we have provided a precise value for the hidden dimensions of the Transformer as required in Theorem 1.

For the convenience of reviewers/future readers, this response provides a high-level overview of our contributions.

---

**Revision Details**

In response to the reviewers' suggestions, we have made several key modifications, summarized as follows:

**Major revisions** include:

1. Replace “Pretrained Transformer” with “Transformer”. [Reviewers `tzww`, `99cj`, and `R2xK`]
2. Add an overview of the empirical part in the Introduction (`line 089–090`) and provide a detailed description in Appendix G.1, which covers aspects such as the model architecture. [Reviewers `99cj`, `R2xK`, and `hJr1`]
3. Add the experiments to show that scaling up the transformer enables it to implement more ICGD steps (Appendix G.2).  [Reviewer `hJr1`]
4. Add a detailed comparison with [Wang24] (In the Introduction and Related Works). [Reviewer `R2xK`]
5. Provide a precise specification of the Transformer dimensions required in Theorem 1. [Reviewer `hJr1`]

**Minor revisions** include:

1. Proofread the manuscript and fix all identified typos and grammatical errors.
2. Improve the readability of the main text.

**A Concise Summary/Emphasis of Our Contribution**

We appreciate the opportunity to clarify the scope of our contributions. In the initial draft, we may not have described the breadth and depth of our work as effectively as we intended. Here, we have summarized our contributions:

* **Structured and efficient transformer construction** We offer a detailed characterization of the ICL capabilities of a transformer model in approximating the gradient descent training process of an $N$-layer network. Utilizing a term-by-term approximation technique and the selection of specific terms, we demonstrate that only $(2N+4)L$ transformer layers are required (Theorem 1) to implement $L$ gradient descent steps. We also detail the approximation error in Corollary 1.1.

* **Less restrictive input and output dimensions for $N$-layer neural networks.** We extend our analysis to the case where output dimensions exceed one, as detailed in Appendix E. This includes scenarios where input and output dimensions differ.

* **More practical transformer model.** We extend our analysis to include Softmax-based transformers, reflecting more realistic applications as detailed in Appendix F.

* **Experimental validation.** We validate our theory with ReLU- and Softmax-based transformers, specifically, ICGD for the $N$-layer networks. The numerical results in Appendix G show that the performance of ICL matches that of training $N$-layer networks.

* **Advanced and complicated applications.** We explore the application of our results to approximate the score function in diffusion models, as discussed in Appendix H.

[Wang24] In-context Learning on Function Classes Unveiled for Transformers, ICML 2024.

---

### Author Response · Authors · 2024-12-02
**Clarifications on Related Work and Contributions**

Dear Reviewers and Future Readers,

We appreciate the additional reference [Wang24] suggested by [Reviewer `R2xK`](https://openreview.net/forum?id=ZIFkrT1GwM&noteId=idJUnb4cWr). It is relevant and was unfortunately overlooked in our initial literature review. Below, we provide clarifications on its relation to our work and address potential misunderstandings about our contributions.

[Wang24] In-context Learning on Function Classes Unveiled for Transformers, ICML 2024.

---

### **Related but Seemingly Irrelevant Work [Wang24]**

* While [Wang24] is relevant, **the shared topics (ICGD of N-layer NN) and results are not mentioned in both its title and abstract.** This makes it appear unrelated at first glance.
* Moreover, **[Wang24] was not readily visible during the preparation of this work.** It's very recent. It is not available on arXiv, and the ICML 2024 paper only became accessible around July. At the time we submitted our work, this paper had no citations.

**We believe it is very likely that most authors in this ICLR submission cycle may also overlook [Wang24], just as we did.**

Despite this, we apologize to the authors of [Wang24] for not attributing proper credit in our initial draft. We acknowledge its relevance and appreciate the opportunity to discuss its connection to our work. However, we believe that **overlooking this particular reference does not diminish the originality or merits of our contributions**, as explained below.

---

### **Clarifications on Our Contributions**
In the most concise manner, our main contributions are as follows:
1. Explicit construction for ICGD of N-layer NN.
2. Error bounds.
3. Results for both ReLU and Softmax transformers.

We have discussed the differences between our work and [Wang24] extensively in both our rebuttal and revisions. Specifically:
- Contributions (2) and (3) are solid and novel, as acknowledged by the reviewers.
- For contribution (1), **while there may be superficial similarities with [Wang24] (both showing ICGD of N-layer NN), our approach uses entirely *different techniques*, operates under *less restrictive assumptions*, and achieves *stronger results*.**  Please see the comparison below for details.

Even if [Wang24] is considered prior work in this area, **our contributions remain complementary and significant.**

---

### **Key Differences**
For clarity, we summarize the key distinctions between our work and [Wang24] below, and direct the readers to our rebuttal and latest revision for further details:

- **More structured and efficient transformer architecture.**
  [Wang24] uses an **$O(N^2L)$**-layer transformer to approximate $L$ gradient descent steps on $N$-layer neural networks. In contrast, our approach adopts entirely different construction strategies, such as inductively approximating intermediate chain-rule terms, and requires only **$(2N + 4)L$** layers.

- **Less restrictive input and output dimensions for $N$-layer neural networks.**
  Unlike [Wang24], which limits the output of $N$-layer networks to a scalar, our results hold for outputs of any dimension.

- **More practical transformer model.**
  We extend our analysis to include Softmax transformers, reflecting more realistic applications, whereas [Wang24] does not consider this.

- **More advanced and complex applications.**
  [Wang24] discusses applications to indicators, linear, and smooth functions. In contrast, we explore more advanced cases, such as the score function in diffusion models.

We understand that, despite our best efforts, it is possible that we may have overlooked or misunderstood certain parts of [Wang24] given the limited time. However, we assure that we have no intention of disregarding or discrediting [Wang24] in any way. The contributions of [Wang24] are solid, interesting, and important. The above discussions are solely for clarification, as the two works have nuanced and different focuses. We are open to any further discussion.

---

We hope these clarifications help readers understand the unique contributions of our work and its relation to [Wang24]. While both works explore related topics, we believe our approach offers distinct theoretical and practical advancements.

We thank all reviewers for their time, constructive feedback, and engagement. Your involvement has helped us improve this work.

Lastly, we thank the authors of [Wang24] for their good work.

Best,

Authors

---

### Meta-Review · Area_Chair_jd7T · 2024-12-22

**Metareview:**

This paper explores the transformer's ability for in-context learning (ICL) to simulate the training process of deep models. The key contribution is demonstrating how a pretrained transformer can train a deep neural network via ICL in an implicit manner. Specifically, the authors construct a(2N+4)L-layer transformer that simulates L gradient descent steps of an N-layer ReLU network. They provide theoretical guarantees for approximation error and ICL convergence. The analysis is extended to Softmax-based transformers in practical settings. Experiments on synthetic datasets for 3-layer, 4-layer, and 6-layer neural networks show that ICL performance matches direct training. The primary concern raised by the reviewer is that some of the contributions in this submission are considered insignificant in light of a recent paper [Wang24] published at ICML 2024. While other contributions, such as the results for softmax transformers, are deemed novel, their presentation is largely deferred to the appendix. I acknowledge that this paper may contain important contributions. However, it requires significant and careful revisions to meet the standard for acceptance. Therefore, I recommend rejection.

[Wang24] In-context Learning on Function Classes Unveiled for Transformers, ICML 2024.

**Additional Comments On Reviewer Discussion:**

Reviewer R2xK pointed out a missing reference [Wang24], which is relevant and was unfortunately overlooked in the initial submission.

[Wang24] In-context Learning on Function Classes Unveiled for Transformers, ICML 2024.

The key distinctions between this submission and [Wang24] are as follows:

1. [Wang24] employs an O(N^2L)-layer transformer to approximate L gradient descent steps on N-layer neural networks. In contrast, this paper only needs an (2N+4)L-layer transformer to approximate L gradient descent steps on N-layer neural networks. However, Reviewer R2xK argues that the reduction in N is relatively straightforward.
2. Unlike [Wang24], which restricts the output of NN-layer networks to a scalar, this paper generalizes to outputs of any dimension. In my opinion, this constitutes a minor contribution.
3. This paper also extends the analysis to include Softmax transformers, addressing more realistic applications that [Wang24] does not consider. This is a significant contribution, as acknowledged by Reviewer R2xK. Unfortunately, much of this analysis and result are currently in the appendix.

After author rebuttal and author-reviewer discussion, two reviewers (99cj, hJr1) increased their scores to 6. Overall, I agree with Reviewer R2xK. I believe this paper requires significant and careful revisions to meet the standard for acceptance. Therefore, I recommend rejection.

---

### Decision · Program_Chairs · 2025-01-22

Reject